# Natural continual learning: success is a journey, not (just) a destination

**Ta-Chu Kao[1*]**   **Kristopher T. Jensen[1*]**   **Gido M. van de Ven[1,2]**

**Alberto Bernacchia[3]**   **Guillaume Hennequin[1]**

1. Department of Engineering, University of Cambridge
2. Center for Neuroscience and Artificial Intelligence, Baylor College of Medicine
3. MediaTek Research, Cambridge

{tck29, ktj21}@cam.ac.uk  ven@bcm.edu
alberto.bernacchia@mtkresearch.com  gjeh2@cam.ac.uk

## Abstract

Biological agents are known to learn many different tasks over the course of their lives, and to be able to revisit previous tasks and behaviors with little to no loss in performance. In contrast, artificial agents are prone to 'catastrophic forgetting' whereby performance on previous tasks deteriorates rapidly as new ones are acquired. This shortcoming has recently been addressed using methods that encourage parameters to stay close to those used for previous tasks. This can be done by (i) using specific parameter regularizers that map out suitable destinations in parameter space, or (ii) guiding the optimization journey by projecting gradients into subspaces that do not interfere with previous tasks. However, these methods often exhibit subpar performance in both feedforward and recurrent neural networks, with recurrent networks being of interest to the study of neural dynamics supporting biological continual learning. In this work, we propose Natural Continual Learning (NCL), a new method that unifies weight regularization and projected gradient descent. NCL uses Bayesian weight regularization to encourage good performance on all tasks at convergence and combines this with gradient projection using the prior precision, which prevents catastrophic forgetting during optimization. Our method outperforms both standard weight regularization techniques and projection based approaches when applied to continual learning problems in feedforward and recurrent networks. Finally, the trained networks evolve task-specific dynamics that are strongly preserved as new tasks are learned, similar to experimental findings in biological circuits.

## 1 Introduction

Catastrophic forgetting is a common feature of machine learning algorithms where training on a new task often leads to poor performance on previously learned tasks. This is in contrast to biological agents which are capable of learning many different behaviors over the course of their lives with little to no interference across tasks. The study of continual learning in biological networks may therefore help inspire novel approaches in machine learning, while the development and study of continual learning algorithms in artificial agents can help us better understand how this challenge is overcome in the biological domain. This is particularly true for more challenging continual learning settings where task identity is not provided at test time, and for continual learning in recurrent neural

35th Conference on Neural Information Processing Systems (NeurIPS 2021).

networks (RNNs), which is important due to the practical and biological relevance of RNNs. However, continual learning in these settings has recently proven challenging for many existing algorithms, particularly those that rely on parameter regularization to mitigate forgetting [12, 13, 47]. In this work, we address these shortcomings by developing a continual learning algorithm that not only encourages good performance across tasks at convergence but also regularizes the optimization path itself using trust region optimization. This leads to improved performance compared to existing methods.

Previous work has addressed the challenge of continual learning in artificial agents using weight regularization, where parameters important for previous tasks are regularized to stay close to their previous values [1, 17, 24, 32, 37, 54]. This approach can be motivated by findings in the neuroscience literature of increased stability for a subset of synapses after learning [49, 50]. More recently, approaches based on projecting gradients into subspaces orthogonal to those that are important for previous tasks have been developed in both feedforward [40, 53] and recurrent [12] neural networks. This is consistent with experimental findings that neural dynamics often occupy orthogonal subspaces across contexts in biological circuits [3, 14, 20, 22]. While these methods have been found to perform well in many continual learning settings, they also suffer from several shortcomings. In particular, while Bayesian weight regularization provides a natural way to weigh previous and current task information, this approach can fail in practice due to its approximate nature and often requires additional tuning of the importance of the prior beyond what would be expected in a rigorous Bayesian treatment [46]. In contrast, while projection-based methods have been found empirically to mitigate catastrophic forgetting, it is unclear how the 'important subspaces' should be selected and how such methods behave when task demands begin to saturate the network capacity.

In this work, we develop natural continual learning (NCL), a new method that combines (i) Bayesian continual learning using weight regularization with (ii) an optimization procedure that relies on a trust region constructed from an approximate posterior distribution over the parameters given previous tasks. This encourages parameter updates predominantly in the null-space of previously acquired tasks while maintaining convergence to a maximum of the Bayesian approximate posterior. We show that NCL outperforms previous continual learning algorithms in both feedforward and recurrent networks. We also show that the projection-based methods introduced by Duncker et al. [12] and Zeng et al. [53] can be viewed as approximations to such trust region optimization using the posterior from previous tasks. Finally, we use tools from the neuroscience literature to investigate how the learned networks overcome the challenge of continual learning. Here, we find that the networks learn latent task representations that are stable over time after initial task learning, consistent with results from biological circuits.

## 2 Method

**Notations** We use $\boldsymbol{X}^\top$, $\boldsymbol{X}^{-1}$, $\mathrm{Tr}(\boldsymbol{X})$ and $\mathrm{vec}(\boldsymbol{X})$ to denote the transpose, inverse, trace, and column-wise vectorization of a matrix $\boldsymbol{X}$. We use $\boldsymbol{X} \otimes \boldsymbol{Y}$ to represent the Kronecker product between matrices $\boldsymbol{X} \in \mathbb{R}^{n \times n}$ and $\boldsymbol{Y} \in \mathbb{R}^{m \times m}$ such that $(\boldsymbol{X} \otimes \boldsymbol{Y})_{mi+k,mj+l} = \boldsymbol{X}_{ij}\boldsymbol{Y}_{kl}$. We use bold lower-case letters $\boldsymbol{x}$ to denote column vectors. $\mathcal{D}_k$ refers to a 'dataset' corresponding to task $k$, which in this work generally consists of a set of input-output pairs $\{\boldsymbol{x}_k^{(i)}, \boldsymbol{y}_k^{(i)}\}$ such that $\ell_k(\boldsymbol{\theta}) := \log p(\mathcal{D}_k|\boldsymbol{\theta}) = \sum_i \log p_{\boldsymbol{\theta}}(\boldsymbol{y}_k^{(i)}|\boldsymbol{x}_k^{(i)})$ is the task-related performance on task $k$ for a model with parameters $\boldsymbol{\theta}$. Finally, we use $\hat{\mathcal{D}}_k$ to refer to a dataset generated by inputs from the $k^{th}$ task where $\{\hat{\boldsymbol{y}}_k^{(i)} \sim p_{\boldsymbol{\theta}}(\boldsymbol{y}|\boldsymbol{x}_k^{(i)})\}$ are drawn from the model distribution $\mathcal{M}$.

### 2.1 Bayesian continual learning

**Problem statement** In continual learning, we train a model on a set of $K$ tasks $\{\mathcal{D}_1, \ldots, \mathcal{D}_K\}$ that arrive sequentially, where the data distribution $\mathcal{D}_k$ for task $k$ in general differs from $\mathcal{D}_{\neq k}$. The aim is to learn a probabilistic model $p(\mathcal{D}|\boldsymbol{\theta})$ that performs well on all tasks. The challenge in the continual learning setting stems from the sequential nature of learning, and in particular from the common assumption that the learner does not have access to "past" tasks (i.e., $\mathcal{D}_j$ for $j < k$) when learning task $k$. While we enforce this stringent condition in this paper, our approach may be easily combined with memory-based techniques such as coresets or generative replay [8, 10, 13, 32, 34, 36, 38, 42, 43, 45, 48].

**Bayesian approach** The continual learning problem is naturally formalized in a Bayesian framework whereby the posterior after $k-1$ tasks is used as a prior for task $k$. More specifically, we choose a prior $p(\boldsymbol{\theta})$ on the model parameters and compute the posterior after observing $k$ tasks according to Bayes' rule:

$$p(\boldsymbol{\theta}|\mathcal{D}_{1:k}) \propto p(\boldsymbol{\theta}) \prod_{k'=1}^{k} p(\mathcal{D}_{k'}|\boldsymbol{\theta})$$

$$\propto p(\boldsymbol{\theta}|\mathcal{D}_{1:k-1})p(\mathcal{D}_k|\boldsymbol{\theta}), \tag{1}$$

where $\mathcal{D}_{1:k}$ is a concatenation of the first $k$ tasks $(\mathcal{D}_1, \ldots, \mathcal{D}_k)$. In theory, it is thus possible to compute the exact posterior $p(\boldsymbol{\theta}|\mathcal{D}_{1:k})$ after $k$ tasks, while only observing $\mathcal{D}_k$, by using the posterior $p(\boldsymbol{\theta}|\mathcal{D}_{1:k-1})$ after $k-1$ tasks as a prior. However, as is often the case in Bayesian inference, the difficulty here is that the posterior is typically intractable. To address this challenge, it is common to perform approximate online Bayesian inference. That is, the posterior $p(\boldsymbol{\theta}|\mathcal{D}_{1:k-1})$ is approximated by a parametric distribution with parameters $\boldsymbol{\phi}_{k-1}$. The approximate posterior $q(\boldsymbol{\theta}; \boldsymbol{\phi}_{k-1})$ is then used as a prior for task $k$.

**Online Laplace approximation** A common approach is to use the Laplace approximation whereby the posterior $p(\boldsymbol{\theta}|\mathcal{D}_{1:k-1})$ is approximated as a multivariate Gaussian $q$ using local gradient information [17, 24, 37]. This involves (i) finding a mode $\boldsymbol{\mu}_k$ of the posterior during task $k$, and (ii) performing a second-order Taylor expansion around $\boldsymbol{\mu}_k$ to construct an approximate Gaussian posterior $q(\boldsymbol{\theta}; \boldsymbol{\phi}_k) = \mathcal{N}(\boldsymbol{\theta}; \boldsymbol{\mu}_k, \boldsymbol{\Lambda}_k^{-1})$, where $\boldsymbol{\Lambda}_k$ is the precision matrix and $\boldsymbol{\phi}_k = (\boldsymbol{\mu}_k, \boldsymbol{\Lambda}_k)$. In this case, gradient-based optimization is used to find the posterior mode on task $k$ (c.f. Equation 1):

$$\boldsymbol{\mu}_k = \arg\max_{\boldsymbol{\theta}} \ \log p(\boldsymbol{\theta}|\mathcal{D}_k, \boldsymbol{\phi}_{k-1}) \tag{2}$$

$$= \arg\max_{\boldsymbol{\theta}} \ \log p(\mathcal{D}_k|\boldsymbol{\theta}) + \log q(\boldsymbol{\theta}; \boldsymbol{\phi}_{k-1}) \tag{3}$$

$$= \arg\max_{\boldsymbol{\theta}} \ \underbrace{\ell_k(\boldsymbol{\theta}) - \frac{1}{2}(\boldsymbol{\theta} - \boldsymbol{\mu}_{k-1})^\top \boldsymbol{\Lambda}_{k-1}(\boldsymbol{\theta} - \boldsymbol{\mu}_{k-1})}_{:= \mathcal{L}_k(\boldsymbol{\theta})} \tag{4}$$

The precision matrix $\boldsymbol{\Lambda}_k$ is given by the Hessian of the negative log posterior at $\boldsymbol{\mu}_k$:

$$\boldsymbol{\Lambda}_k = - \nabla_{\boldsymbol{\theta}}^2 \log p(\boldsymbol{\theta}|\mathcal{D}_k, \boldsymbol{\phi}_{k-1})\big|_{\boldsymbol{\theta}=\boldsymbol{\mu}_k} = H(\mathcal{D}_k, \boldsymbol{\mu}_k) + \boldsymbol{\Lambda}_{k-1}, \tag{5}$$

where $H(\mathcal{D}_k, \boldsymbol{\mu}_k) = - \nabla_{\boldsymbol{\theta}}^2 \log p(\mathcal{D}_k|\boldsymbol{\theta})\big|_{\boldsymbol{\theta}=\boldsymbol{\mu}_k}$ is the Hessian of the negative log likelihood of $\mathcal{D}_k$.

Continual learning with the online Laplace approximation thus involves two steps for each new task $\mathcal{D}_k$. First, given $\mathcal{D}_k$ and the previous posterior $q(\boldsymbol{\theta}; \boldsymbol{\mu}_{k-1}, \boldsymbol{\Lambda}_{k-1}^{-1})$ (i.e. the new prior), $\boldsymbol{\mu}_k$ is found using gradient-based optimization (Equation 4). This step can be interpreted as optimizing the likelihood of $\mathcal{D}_k$ while penalizing changes in the parameters $\boldsymbol{\theta}$ according to their importance for previous tasks, as determined by the prior precision matrix $\boldsymbol{\Lambda}_{k-1}$. Second, the new posterior precision matrix $\boldsymbol{\Lambda}_k$ is computed according to Equation 5.

**Approximating the Hessian** In practice, computing $\boldsymbol{\Lambda}_k$ presents two major difficulties. First, because $q(\boldsymbol{\theta}; \boldsymbol{\phi}_k)$ is a Gaussian distribution, $\boldsymbol{\Lambda}_k$ has to be positive semi-definite (PSD), which is not guaranteed for the Hessian $H(\mathcal{D}_k, \boldsymbol{\mu}_k)$. Second, if the number of model parameters $n_\theta$ is large, it may be prohibitive to compute a full $(n_\theta \times n_\theta)$ matrix. To address the first issue, it is common to approximate the Hessian with the Fisher information matrix (FIM; 17, 30, 37):

$$\boldsymbol{F}_k = \mathbb{E}_{p(\hat{\mathcal{D}}_k|\boldsymbol{\theta})} \left[ \nabla_{\boldsymbol{\theta}} \log p(\hat{\mathcal{D}}_k|\boldsymbol{\theta}) \nabla_{\boldsymbol{\theta}} \log p(\hat{\mathcal{D}}_k|\boldsymbol{\theta})^\top \right]\Big|_{\boldsymbol{\theta}=\boldsymbol{\mu}_k} \approx H(\mathcal{D}_k, \boldsymbol{\mu}_k) \tag{6}$$

The FIM is PSD, which ensures that $\boldsymbol{\Lambda}_k = \sum_{k'=1}^{k} \boldsymbol{F}_{k'}$ is also PSD. Computing $\boldsymbol{F}_k$ may still be impractical if there are many model parameters, and it is therefore common to further approximate the FIM using structured approximations with fewer parameters. In particular, a diagonal approximation to $\boldsymbol{F}_k$ recovers Elastic Weight Consolidation (EWC; 24), while a Kronecker-factored approximation [31] recovers the method proposed by Ritter et al. [37]. We denote this method 'KFAC' and use it in Section 3 as a comparison for our own Kronecker-factored method.

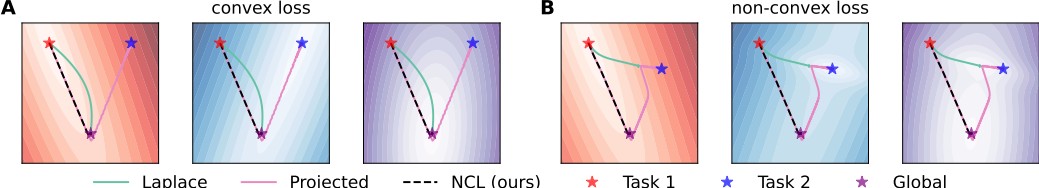

Figure 1: **Continual learning in a toy problem.** **(A)** Loss landscapes of task 1 ($\ell_1$; left), task 2 ($\ell_2$; middle) and the combined loss $\ell_{1+2} = \ell_1 + \ell_2$ (right). Stars indicate the global optima for $\ell_1$ (red), $\ell_2$ (blue), and $\ell_{1+2}$ (purple). We assume that $\boldsymbol{\theta}$ has been optimized for $\ell_1$ and consider how learning proceeds on task 2 using either the Laplace posterior ('Laplace', green), projected gradient descent on $\ell_2$ with preconditioning according to task 1 ('Projected', pink), or NCL (black dashed). Laplace follows the steepest gradient of $\ell_{1+2}$ and transiently forgets task 1. NCL follows a flat direction of $\ell_1$ and converges to the global optimum of $\ell_{1+2}$ with good performance on task 1 throughout. Projected gradient descent follows a similar optimization path to NCL but eventually diverges towards the optimum of $\ell_2$. **(B)** As in (A), now with non-convex $\ell_2$ (center), leading to a second local optimum of $\ell_{1+2}$ (right) while $\ell_1$ is unchanged (left). In this case, Laplace can converge to a local optimum which has 'catastrophically' forgotten task 1. Projected gradient descent moves only slowly in 'steep' directions of $\ell_1$ but eventually converges to a minimum of $\ell_2$. Finally, NCL finds a local optimum of $\ell_{1+2}$ which retains good performance on task 1. See Appendix K for further mathematical details.

## 2.2 Natural continual learning

While the online Laplace approximation has been applied successfully in several continual learning settings [24, 37], it has also been found to perform sub-optimally on a range of problems [12, 46]. Additionally, its Bayesian interpretation in theory prescribes a unique way of weighting the contributions of previous and current tasks to the loss. However, to perform well in practice, weight regularization approaches have been found to require ad-hoc re-weighting of the prior term by several orders of magnitude [24, 37, 46]. These shortcomings could be due to an inadequacy of the approximations used to construct the posterior (Section 2.1). However, we show in Figure 1 that standard gradient descent on the Laplace posterior has important drawbacks even in the exact case. First, we show that exact Bayesian inference on a simple continual regression problem can produce indirect optimization paths along which previous tasks are transiently forgotten as a new task is being learned (Figure 1A; green). Second, when the loss is non-convex, we show that exact Bayesian inference can still lead to catastrophic forgetting (Figure 1B; green).

An alternative approach that has found recent success in a continual learning setting involves projection based methods which restrict parameter updates to a subspace that does not interfere with previous tasks [12, 53]. However, it is not immediately obvious how this projected subspace should be selected in a way that appropriately balances learning on previous and current tasks. Additionally, such projection-based algorithms have fixed points that are minima of the current task, but not necessarily minima of the (negative) Bayesian posterior. This can lead to catastrophic forgetting in the limit of long training times (Figure 1; pink), unless the learning rate is exactly zero in directions that interfere with previous tasks.

To combine the desirable features of both classes of methods, we introduce "Natural Continual Learning" (NCL) – an extension of the online Laplace approximation that also restricts parameter updates to directions which do not interfere strongly with previous tasks. In a Bayesian setting, we can conveniently express what is meant by such directions in terms of the prior precision matrix $\boldsymbol{\Lambda}$. In particular, 'flat' directions of the prior (low precision) correspond to directions that will not significantly affect the performance on previous tasks. Formally, we derive NCL as the solution of a trust region optimization problem. This involves minimizing the posterior loss $\mathcal{L}_k(\boldsymbol{\theta})$ within a region of radius $r$ centered around $\boldsymbol{\theta}$ with a distance metric of the form $d(\boldsymbol{\theta}, \boldsymbol{\theta} + \boldsymbol{\delta}) = \sqrt{\boldsymbol{\delta}^\top \boldsymbol{\Lambda}_{k-1} \boldsymbol{\delta} / 2}$ that takes into account the curvature of the prior via its precision matrix $\boldsymbol{\Lambda}_{k-1}$:

$$\boldsymbol{\delta} = \arg\min_{\boldsymbol{\delta}} \mathcal{L}_k(\boldsymbol{\theta}) + \nabla_{\boldsymbol{\theta}} \mathcal{L}_k(\boldsymbol{\theta})^\top \boldsymbol{\delta} \quad \text{subject to} \quad \frac{1}{2} \boldsymbol{\delta}^\top \boldsymbol{\Lambda}_{k-1} \boldsymbol{\delta} \leq r^2, \tag{7}$$

where $\mathcal{L}_k(\boldsymbol{\theta} + \boldsymbol{\delta}) \approx \mathcal{L}_k(\boldsymbol{\theta}) + \nabla_{\boldsymbol{\theta}} \mathcal{L}_k(\boldsymbol{\theta})^\top \boldsymbol{\delta}$ is a first-order approximation to the updated Laplace objective. The solution to this subproblem is given by $\boldsymbol{\delta} \propto \boldsymbol{\Lambda}_{k-1}^{-1} \nabla_{\boldsymbol{\theta}} \ell_k(\boldsymbol{\theta}) - (\boldsymbol{\theta} - \boldsymbol{\mu}_{k-1})$ (see

Appendix A for a derivation), which gives rise to the NCL update rule

$$\boldsymbol{\theta} \leftarrow \boldsymbol{\theta} + \gamma \left[ \boldsymbol{\Lambda}_{k-1}^{-1} \nabla_{\boldsymbol{\theta}} \ell_k(\boldsymbol{\theta}) - (\boldsymbol{\theta} - \boldsymbol{\mu}_{k-1}) \right] \tag{8}$$

for a learning rate parameter $\gamma$ (which is implicitly a function of $r$ in Equation 7). To get some intuition for this learning rule, we note that $\boldsymbol{\Lambda}_{k-1}^{-1}$ acts as a preconditioner for the first (likelihood) term, which drives learning on the current task while encouraging parameter changes predominantly in directions that do not interfere with previous tasks. Meanwhile, the second term encourages $\boldsymbol{\theta}$ to stay close to $\boldsymbol{\mu}_{k-1}$, the optimal parameters for the previous task. As we illustrate in Figure 1, this combines the desirable features of both Bayesian weight regularization and projection-based methods. In particular, NCL shares the fixed points of the Bayesian posterior while also mitigating intermediate or complete forgetting of previous tasks by preconditioning with the prior covariance. Notably, if the loss landscape is non-convex (as it generally will be), NCL can converge to a different local optimum from standard weight regularization despite having the same fixed points (Figure 1B).

**Implementation**    The general NCL framework can be applied with different approximations to the Fisher matrix $\boldsymbol{F}_k$ in Equation 6 (see Section 2.1). In this work, we use a Kronecker-factored approximation [31, 37]. However, even after making a Kronecker-factored approximation to $\boldsymbol{F}_k$ for each task $k$, it remains difficult to compute the inverse of a sum of $k$ Kronecker products (c.f. Equation 5). To address this challenge, we derived an efficient algorithm for making a Kronecker-factored approximation to $\boldsymbol{\Lambda}_k = \boldsymbol{F}_k + \boldsymbol{\Lambda}_{k-1} \approx \boldsymbol{A}_k \otimes \boldsymbol{G}_k$ when $\boldsymbol{\Lambda}_{k-1} = \boldsymbol{A}_{k-1} \otimes \boldsymbol{G}_{k-1}$ and $\boldsymbol{F}_k$ are also Kronecker products. This approximation minimizes the KL-divergence between $\mathcal{N}(\boldsymbol{\mu}_k, (\boldsymbol{A}_k \otimes \boldsymbol{G}_k)^{-1})$ and $\mathcal{N}(\boldsymbol{\mu}_k, (\boldsymbol{\Lambda}_{k-1} + \boldsymbol{F}_k)^{-1})$ (see Appendix G for details). Before training on the first task, we assume a spherical Gaussian prior $\boldsymbol{\theta} \sim \mathcal{N}(\boldsymbol{0}, p_w^{-2}\boldsymbol{I})$. The scale parameter $p_w$ can either be set to a fixed value (e.g. 1) or treated as a hyperparameter, and we optimize $p_w$ explicitly for our experiments in feedforward networks. NCL also has a parameter $\alpha$ which is used to stabilize the matrix inversion $\boldsymbol{\Lambda}_{k-1}^{-1} \approx (\boldsymbol{A}_{k-1} \otimes \boldsymbol{G}_{k-1} + \alpha^2 \boldsymbol{I})^{-1}$ (Appendix E). This is equivalent to a hyperparameter used for such matrix inversions in OWM [53] and DOWM [12], and it is important for good performance with these methods. The $p_w$ and $\alpha$ are largely redundant for NCL, and we generally prefer to fix $\alpha$ to a small value ($10^{-10}$) and optimize the $p_w$ only. However, for our experiments in RNNs, we instead fix $p_w = 1$ and perform a hyperparameter optimization over $\alpha$ for a more direct comparison with OWM and DOWM. The NCL algorithm is described in pseudocode in Appendix E together with additional implementation and computational details. Finally, while we have derived NCL with a Laplace approximation in this section for simplicity, it can similarly be applied in the variational continual learning framework of Nguyen et al. [32] (Appendix J). Our code is available online[1].

## 2.3   Related work

As discussed in Section 2.1, our method is derived from prior work that relies on Bayesian inference to perform weight regularization for continual learning [17, 24, 32, 37]. However, we also take inspiration from the literature on natural gradient descent [2, 25] to introduce a preconditioner that encourages parameter updates primarily in flat directions of previously learned tasks (Appendix H).

Recent projection-based methods [12, 40, 53] have addressed the continual learning problem using an update rule of the form

$$\boldsymbol{\theta} \leftarrow \boldsymbol{\theta} + \gamma \boldsymbol{P}_L \nabla_{\boldsymbol{\theta}} \ell_k(\boldsymbol{\theta}) \boldsymbol{P}_R, \tag{9}$$

where $\boldsymbol{P}_L$ and $\boldsymbol{P}_R$ are projection matrices constructed from previous tasks which encourage parameter updates that do not interfere with performance on these tasks. Using Kronecker identities, we can rewrite Equation 9 as

$$\boldsymbol{\theta} \leftarrow \boldsymbol{\theta} + \gamma (\boldsymbol{P}_R \otimes \boldsymbol{P}_L) \nabla_{\boldsymbol{\theta}} \ell_k(\boldsymbol{\theta}). \tag{10}$$

This resembles the NCL update rule in Equation 8 where we identify $\boldsymbol{P}_R \otimes \boldsymbol{P}_L$ with the approximate inverse prior precision matrix used for gradient preconditioning in NCL, $\boldsymbol{\Lambda}_{k-1}^{-1} = \boldsymbol{A}_{k-1}^{-1} \otimes \boldsymbol{G}_{k-1}^{-1}$. Indeed, we note that for a Kronecker-structured approximation to $\boldsymbol{F}_k$, the matrix $\boldsymbol{A}_{k-1}$ approximates the empirical covariance matrix of the network activations experienced during all tasks up to $k-1$ (4, 31, Appendix D), which is exactly the inverse of the projection matrix $\boldsymbol{P}_R$ used in previous work [12, 53]. We thus see that NCL takes the form of recent projection-based continual learning algorithms with two notable differences:

---

[1]https://github.com//tachukao/ncl

(i) NCL uses a *left* projection matrix $\boldsymbol{P}_L$ designed to approximate the posterior covariance of previous tasks $\boldsymbol{\Lambda}_{k-1}^{-1} \approx \boldsymbol{P}_R \otimes \boldsymbol{P}_L$ (i.e., the prior covariance on task $k$; Appendix D), while Zeng et al. [53] use the identity matrix $\boldsymbol{I}$ and Duncker et al. [12] use the covariance of recurrent inputs (Appendix F). Notably, both of these choices of $\boldsymbol{P}_L$ still provide reasonable approximations to $\boldsymbol{\Lambda}_{k-1}^{-1}$, and thus the parameter updates of OWM and DOWM can also be viewed as projecting out steep directions of the prior on task $k$ (Appendix F).

(ii) NCL includes an additional regularization term $(\boldsymbol{\theta} - \boldsymbol{\mu}_{k-1})$ derived from the Bayesian posterior objective, while Duncker et al. [12] and Zeng et al. [53] do not use such regularization. Importantly, this means that while NCL has a similar preconditioner and optimization path to these projection based methods, NCL has stationary points at the modes of the approximate Bayesian posterior while the stationary points of OWM and DOWM do not incorporate prior information from previous tasks (c.f. Figure 1).

It is also interesting to note that previous Bayesian continual learning algorithms include a hyperparameter $\lambda$ that scales the prior compared to the likelihood term for the current task [27]:

$$\mathcal{L}_k^{(\lambda)}(\boldsymbol{\theta}) = \log p(\mathcal{D}_k|\boldsymbol{\theta}) - \lambda(\boldsymbol{\theta} - \boldsymbol{\mu}_{k-1})^\top \boldsymbol{\Lambda}_{k-1}(\boldsymbol{\theta} - \boldsymbol{\mu}_{k-1}). \tag{11}$$

To minimize this loss and thus find a mode of the approximate posterior, it is common to employ pseudo-second-order stochastic gradient-based optimization algorithms such as Adam [23] that use their own gradient preconditioner based on an approximation to the Hessian of Equation 11. Interestingly, this Hessian is given by $\boldsymbol{H}_k = -H(\mathcal{D}_k, \boldsymbol{\theta}) - \lambda\boldsymbol{\Lambda}_{k-1}$, which in the limit of large $\lambda$ becomes increasingly similar to preconditioning with the prior precision as in NCL. Consistent with this, previous work using the online Laplace approximation has found that large values of $\lambda$ are generally required for good performance [24, 37, 46]. Recent work has also combined Bayesian continual learning with natural gradient descent [33, 44], and in this case a relatively high value of $\lambda = 100$ was similarly found to maximize performance [33].

## 3 Experiments and results

### 3.1 NCL in feedforward networks

To verify the utility of NCL for continual learning, we first compared our algorithm to standard methods in feedforward networks across two continual learning benchmarks: split MNIST and split CIFAR-100 (see Appendix B for task details). For each benchmark, we considered three continual learning settings [47]. In the 'task-incremental' setting, task identity is available to the network at test time, in our case via a multi-head output layer [5]. In the 'domain-incremental' setting, task identity is unavailable at test time, and the output layer is shared between all tasks. Finally, in the 'class-incremental' setting, the network has to both infer task identity and solve the task, in our case by performing classification over all possible classes irrespective of which task the input in question is drawn from.

van de Ven and Tolias previously showed that parameter regularization methods such as EWC perform poorly in the domain- and class-incremental settings [47]. We therefore applied NCL as well as synaptic intelligence [SI; 54], online EWC [41], Kronecker factored EWC [KFAC; 37], and orthogonal weight modification [OWM; 53] to split MNIST and split CIFAR-100 in the task-, domain- and class-incremental learning settings. For these continual learning problems, we found that NCL outperformed all the baseline methods in the task- and domain-incremental learning settings (Figure 2). In the class-incremental settings, we found that NCL performed comparably to but slightly worse than OWM. However, both OWM and NCL comfortably outperformed the other compared methods in this setting. These results suggest that the subpar performance of parameter regularization methods can be alleviated by regularizing their optimization paths, particularly in the domain- and class-incremental learning settings.

For the split MNIST and split CIFAR-100 experiments, each baseline method had a single hyperparameter ($c$ for SI, $\lambda$ for EWC and KFAC, $\alpha$ for OWM, and $p_w$ for NCL; Appendix E) that was optimized on a held-out seed (see Appendix I.2). However, by setting the NCL prior to a unit Gaussian, we were also able to achieve good performance across task sets in a hyperparameter-free setting, further highlighting the robustness of the method (see "NCL (no opt)" in Figure 2).

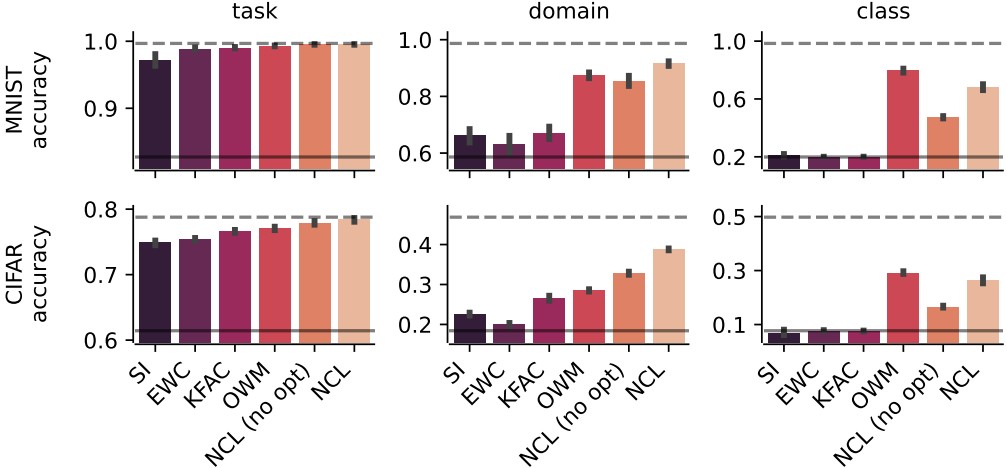

Figure 2: **NCL performance in feedforward networks.** Average test accuracy after learning all tasks on split MNIST (top row) and split CIFAR-100 (bottom-row) in the task-, domain- and class-incremental learning setting. Dashed horizontal lines denote average performance when networks are trained simultaneously on all tasks. Solid horizontal lines denote average performance when networks are trained sequentially on each task without applying any continual learning methods. Error bars denote standard error across 20 (MNIST) or 10 (CIFAR) random seeds. 'NCL' indicates natural continual learning where the initial prior has been optimized on a held-out random seed, and 'NCL (no-prior)' indicates NCL with a simple unit Gaussian prior and no hyperparameter optimization. Numerical results for these experiments are provided in Table 2 in Appendix I.3.

## 3.2 NCL in recurrent neural networks

We then proceeded to consider how NCL compares to previous methods in recurrent neural networks (RNNs), a setting that has recently proven challenging for continual learning [12, 13] and which is of interest to the study of continual learning in biological circuits [12, 51]. In these experiments, the task identity is available to the RNN (i.e., we consider the task-incremental learning setting).

**Stimulus-response tasks** In this section, we consider a set of neuroscience inspired 'stimulus-response' (SR) tasks (51; details in Appendix B). We first compared the performance and behavior of NCL to OWM, the top performing method in the feedforward setting (Figure 2), and to the projection-based DOWM method designed explicitly for RNNs [12]. For a more direct comparison with OWM and DOWM, we fixed the NCL prior to a unit Gaussian for all RNN experiments and instead performed a hyperparameter optimization over '$\alpha$' used to regularize the matrix inversions for all three methods (Section 2.2, Appendix E, Appendix I.2, 12, 53). Following previous work, we trained RNNs with 256 recurrent units to sequentially solve six stimulus-response tasks [12, 50]. While NCL, OWM and DOWM all managed to learn the six tasks without catastrophic forgetting, we found that NCL achieved superior average performance across tasks after training (Figure 3A).

We then compared NCL, OWM, and DOWM to KFAC, the top performing parameter regularization method in our feedforward experiments (Figure 2) which uses Adam [23] to optimize the objective in Equation 4 with a Kronecker-factored approximation to the posterior precision matrix (Section 2.1; 37). Consistent with the results shown in Duncker et al. [12], we found that NCL, OWM, and DOWM outperformed KFAC with $\lambda = 1$ (Figure 3A; see also 12 for a comparison of DOWM and EWC). We note that NCL and KFAC optimize the same objective function (Equation 4) and approximate the posterior precision matrix in the same way, but they differ in the way they precondition the gradient of the objective. These results thus demonstrate empirically that the choice of optimization algorithm is important to prevent forgetting, consistent with the intuition provided by Figure 1.

In feedforward networks, poor performance with weight regularization approaches such as EWC and KFAC has been mitigated by optimizing the hyperparameter $\lambda$, which increases the importance of the prior term compared to a standard Bayesian treatment (Equation 11; Section 3.1, 24, 27, 37). We confirmed this here by performing a grid search over $\lambda$, which showed that KFAC with $\lambda \in$

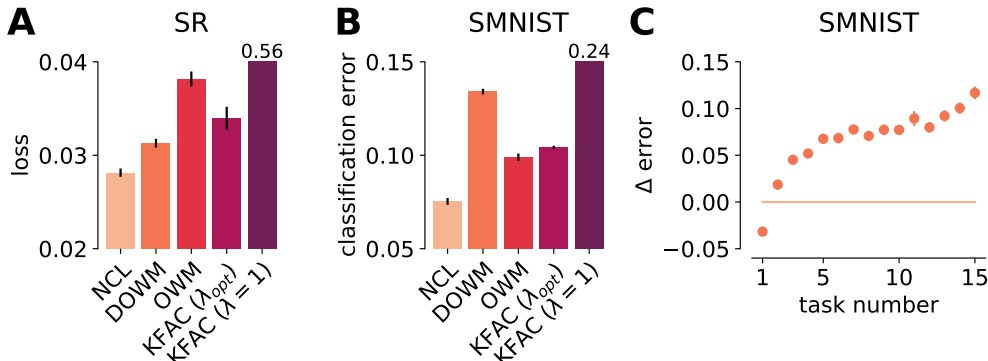

Figure 3: **Performance on SR and SMNIST tasks. (A)** Mean loss of NCL, DOWM, OWM, KFAC (optimal $\lambda$), and KFAC ($\lambda = 1$) across stimulus-response tasks after sequential training on all tasks. Error bars indicate standard error across 5 random seeds. Here and in (B), KFAC with $\lambda = 1$ failed catastrophically, and its performance is indicated in text as it does not fit on the axes. **(B)** Mean classification error across SMNIST tasks after sequential training. **(C)** Difference between the mean classification error of Laplace-DOWM and NCL as a function of task number. Error bars in (B) and (C) indicate standard error across 100 random task permutations.

$[100, 1000]$ could perform comparably to the projection-based methods (Appendix I.1; Figure 3A). We hypothesize that the good performance provided by high $\lambda$ is partly due to the approximate second order nature of Adam which, together with the relative increase in the prior term compared to the data term, leads to preconditioning with a matrix resembling the prior $\mathbf{\Lambda}_{k-1}$ (Section 2.3). In support of this hypothesis, we found that the KL divergence between the Adam preconditioner and the approximate prior precision $\mathbf{\Lambda}_{k-1}$ decreased with increasing $\lambda$, and that the performance of KFAC with Adam could also be rescued by increasing $\lambda$ only when computing the preconditioner while retaining $\lambda = 1$ when computing the gradients (Appendix I.1).

**Stroke MNIST**  One way to challenge the continual learning algorithms further is to increase the number of tasks. We thus considered an augmented version of the stroke MNIST dataset [SMNIST; 9]. The original dataset consists of the MNIST digits transformed into pen strokes with the direction of the stroke at each time point provided as an input to the network. Similar to Ehret et al. [13], we constructed a continual learning problem by considering consecutive binary classification tasks inspired by the split MNIST task set. We further increased the number of tasks by including a set of extra digits where the x and y dimensions have been swapped in the input stroke data, and another set where both the x and y dimensions have changed sign. We also added high-variance noise to the inputs to increase the task difficulty. This gave rise to a total of 15 binary classification tasks, each with unique digits not used in other tasks, which we sought to learn in a continual fashion using an RNN with 30 recurrent units (see Appendix B for details).

As for the SR task set in Section 3.2, we found that NCL outperformed previous projection-based methods (Figure 3B). We again found that weight regularization with a KFAC approximation performed poorly with $\lambda = 1$, and that this poor performance could be partially rescued by optimizing over $\lambda$ (Figure 3B). To investigate how the difference in performance between NCL and DOWM was affected by their different approximations to the Fisher matrix (Appendix F), we implemented NCL using the DOWM projection matrices as an alternative approximation to the inverse Fisher matrix. We refer to this method as Laplace-DOWM. We then considered how the performance on each task at the end of training depended on task number, averaged over different task permutations (Figure 3C). We found that while Laplace-DOWM outperformed NCL on the first task, this method generally performed worse on subsequent tasks. Notably, Laplace-DOWM exhibited a near-monotonic decrease in relative performance with task number, which is consistent with the intuition that DOWM overestimates the dimensionality of the parameter subspace that matters for previous tasks (Appendix F). In contrast, although neural circuits are known to use orthogonal subspaces in different contexts, there is no general sense that learning more tasks in the past should systematically hinder learning in future contexts for biological agents.

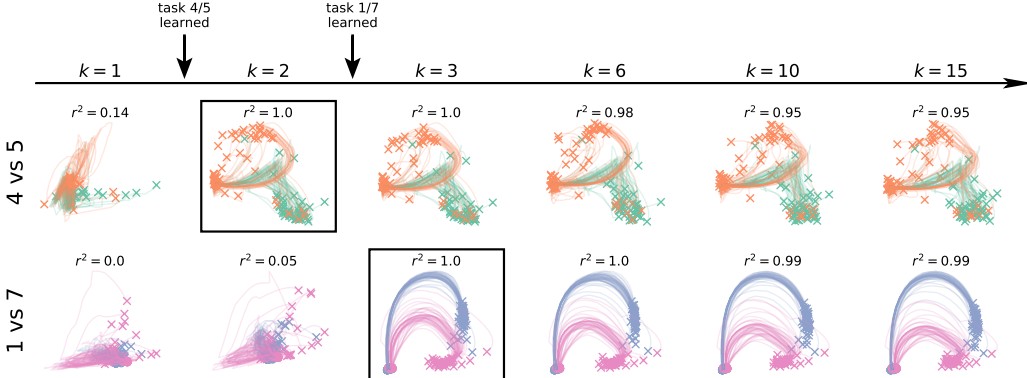

Figure 4: **Latent dynamics during SMNIST.** We considered two example tasks, 4 vs 5 (top) and 1 vs 7 (bottom). For each task, we simulated the response of a network trained by NCL to 100 digits drawn from that task distribution at different times during learning. We then fitted a factor analysis model for each example task to the response of the network right after the correponding task had been learned (squares; $k = 2$ and $k = 3$ respectively). We used this model to project the responses at different times during learning into a common latent space for each example task. For both example tasks, the network initially exhibited variable dynamics with no clear separation of inputs and subsequently acquired stable dynamics after learning to solve the task. The $r^2$ values above each plot indicate the similarity of neural population activity with that collected immediately after learning the corresponding task, quantified across all neurons (not just the 2D projection).

### 3.3 Dissecting the dynamics of networks trained on the SMNIST task set

To further investigate how the trained RNNs solve the continual learning problems and how this relates to the neuroscience literature, we dissected the dynamics of networks trained on the SMNIST task set using the NCL algorithm. To do this, we analyzed latent representations of the RNN activity trajectories, as is commonly done to study the collective dynamics of artificial and biological networks [15, 18, 20, 29, 52]. We considered two consecutive classification tasks, namely classifying 4's vs 5's ($k = 2$) and classifying 1's vs 7's ($k = 3$). For each of these tasks, we trained a factor analysis model right after the task was learned, using network activity collected while presenting 50 examples of each of the two input digits associated with the task. We then tracked the network responses to the same set of stimuli at various stages of learning, both before and after the task in question was acquired, using the trained factor analysis model to visualize low-dimensional summaries of the dynamics (Figure 4).

Consistent with the network having successfully learned to solve these two tasks, we found that latent trajectories diverged over time for the two types of inputs in each task. Critically, these diverging dynamics only emerged after the task was learned, and remained highly stable thereafter (Figure 4). The stability of the task-associated representations is consistent with recent work in the neuroscience literature showing that, in a primate reaching task, latent neural trajectories remain stable after learning [15]. Since here we have access to the activity of all neurons throughout the task, we proceeded to quantify the source of this stability at the level of single units. The stability of such single-neuron dynamics after learning has recently been a topic of much interest in biological circuits [7, 28, 39]. In the RNNs, we found that the single-unit representations of a given digit changed during learning of the task involving that digit but stabilized after learning, consistent with work in several distinct biological circuits [6, 11, 16, 19, 21, 35]. Similar results were found using the DOWM algorithm, which was explicitly designed to preserve network dynamics on previously learned tasks [12]. Interestingly, the stable task representations learned by NCL and DOWM differed markedly from a network trained with replay for continual learning, which instead led to task representations that continued to change after initial task acquisition (Appendix I.4). This illustrates how different approaches to continual learning can lead to qualitatively different circuit dynamics, and it suggests the use of continual learning in artificial networks as a model system for biological continual learning.

# 4 Discussion

In summary, we have developed a new framework for continual learning based on approximate Bayesian inference combined with trust-region optimization. We showed that this framework encompasses recent projection-based methods and found that it performs better than naive weight regularization. This was particularly evident when task identity was not provided at test time and in recurrent neural networks, settings which have previously been challenging for many continual learning algorithms [12, 13, 47]. Furthermore, we showed that our principled probabilistic approach outperforms previous projection-based methods [12, 53], in particular when the number of tasks and their complexity challenges the network's capacity. Finally, we analyzed the dynamics of the learned RNNs in a sequential binary classification problem, where we found that the latent dynamics adapt to each new task. We also found that the task-associated dynamics were subsequently conserved during further learning, consistent with experimental reports of stable neural representations [11, 15, 19]. Importantly, our results suggest that preconditioning with the prior covariance can lead to improved performance over existing continual learning algorithms. In future work, it will therefore be interesting to apply this idea to other weight regularization approaches such as EWC with a diagonal approximate posterior [24]. Finally, a separate branch of continual learning utilizes replay-like mechanisms to reduce catastrophic forgetting [8, 26, 34, 42, 43, 46]. While our work has focused on weight regularization, such regularization and replay are not mutually exclusive. Instead, these two approaches have been found to further improve robustness to catastrophic forgetting when combined [32, 45].

**Impact and limitations**   While we have shown that NCL represents an important conceptual and methodological advance for continual learning, it also comes with several limitations. One such limitation arises from the relative difficulty of computing the prior Fisher matrix which is needed for our projection step. Indeed the success of methods such as Adam [23] and EWC [24] is due in part to their ease of implementation which facilitates broad applicability. It will therefore be interesting to investigate how approximations such as a running average of a diagonal approximation to the empirical Fisher matrix as used in Adam could facilitate the development of simple yet powerful variants of NCL.

Furthermore, while NCL mitigates the need to overcount the prior from previous tasks via $\lambda$ as in KFAC, it does introduce two other (largely redundant) hyperparameters in the form of (i) the scale of the prior before the first task, and (ii) the parameter $\alpha$ used to regularize the inversion of the prior Fisher matrix, similar to OWM and DOWM [12, 53]. While $\alpha$ is an important hyperparameter for OWM and DOWM and we also optimize it in the RNN setting for a more direct comparison (Section 3.2), we find it more natural to set this parameter to a constant small value present only for numerical stability (Appendix E). This leaves the prior scale which we optimize explicitly in the feedforward setting (Section 3.1). However, in future work it would be interesting to consider whether a good prior can be determined in a data free manner to make NCL a hyperparameter-free method. Finally, computing the Fisher matrix used for pre-conditioning requires explicit knowledge of task boundaries. In future work, it will therefore be interesting to develop an algorithm similar to NCL which also works for online learning problems with continually changing task distributions.

Addressing these challenges is important since machine learning algorithms increasingly need to be robust to changing data distributions and dynamic task specifications as they become more prominent in our everyday lives. Much work has therefore gone into the development of methods for continual learning in the machine learning community. However, with the increasing prevalence of practical algorithms for continual learning, it also becomes increasingly important that we understand how and why these algorithms work – insights that can also help us understand when they might fail. In this work, we have therefore attempted to shed light on the relationship between recent methods for continual learning as well as developing a new algorithm with a principled probabilistic interpretation that makes its underlying assumptions more explicit. Taken together, we hope that this work will help improve our understanding of methods for continual learning while also providing an avenue for further research to increase the reliability and robustness of future continual learning algorithms.

## Acknowledgements

We are grateful to Siddharth Swaroop, Lea Duncker, Laura Driscoll, Naama Kadmon Harpaz, and Yashar Ahmadian for insightful discussions. We thank Siddharth Swaroop and Robert Pinsler for useful comments on the manuscript.

## Funding disclosure

KTJ was funded by a Gates Cambridge scholarship. GMV was supported by the Lifelong Learning Machines (L2M) program of the Defence Advanced Research Projects Agency (DARPA) via contract number HR0011-18-2-0025.

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
