# Appendix – Natural continual learning

## A  Derivation of the NCL learning rule

In this section, we provide further details of how the NCL learning rule in Section 2.2 is derived and also provide an alternative derivation of the algorithm.

**NCL learning rule**   As discussed in Section 2.2, we derive NCL as the solution of a trust region optimization problem. That is, we maximize the posterior loss $\mathcal{L}_k(\boldsymbol{\theta})$ within a region of radius $r$ centered around $\boldsymbol{\theta}$ with a distance metric of the form $d(\boldsymbol{\theta}, \boldsymbol{\theta} + \boldsymbol{\delta}) = \sqrt{\boldsymbol{\delta}^\top \boldsymbol{\Lambda}_{k-1} \boldsymbol{\delta}/2}$. This distance metric was chosen to take into account the curvature of the prior via its precision matrix $\boldsymbol{\Lambda}_{k-1}$ and encourage parameter updates that do not affect performance on previous tasks. Formally, we solve the optimization problem

$$\boldsymbol{\delta} = \arg\min_{\boldsymbol{\delta}} \mathcal{L}_k(\boldsymbol{\theta}) + \nabla_{\boldsymbol{\theta}} \mathcal{L}_k(\boldsymbol{\theta})^\top \boldsymbol{\delta} \quad \text{subject to} \quad \frac{1}{2} \boldsymbol{\delta}^\top \boldsymbol{\Lambda}_{k-1} \boldsymbol{\delta} \le r^2, \tag{12}$$

where $\mathcal{L}_k(\boldsymbol{\theta} + \boldsymbol{\delta}) \approx \mathcal{L}_k(\boldsymbol{\theta}) + \nabla_{\boldsymbol{\theta}} \mathcal{L}_k(\boldsymbol{\theta})^\top \boldsymbol{\delta}$ is a first-order approximation to the updated Laplace objective. Here we recall from Equation 4 that

$$\mathcal{L}_k(\boldsymbol{\theta}) = \ell_k(\boldsymbol{\theta}) - \frac{1}{2}(\boldsymbol{\theta} - \boldsymbol{\mu}_{k-1})^T \boldsymbol{\Lambda}_{k-1}(\boldsymbol{\theta} - \boldsymbol{\mu}_{k-1}) \tag{13}$$

from which we get

$$\nabla_{\boldsymbol{\theta}} \mathcal{L}_k(\boldsymbol{\theta})^\top \boldsymbol{\delta} = \nabla_{\boldsymbol{\theta}} \ell_k(\boldsymbol{\theta})^\top \boldsymbol{\delta} - (\boldsymbol{\theta} - \boldsymbol{\mu}_{k-1})^\top \boldsymbol{\Lambda}_{k-1} \boldsymbol{\delta} \tag{14}$$

The optimization in Equation 12 is carried out by introducing a Lagrange multiplier $\eta$ to construct a Lagrangian $\tilde{\mathcal{L}}$:

$$\tilde{\mathcal{L}}(\boldsymbol{\delta}, \eta) = \mathcal{L}_k(\boldsymbol{\theta}) + \nabla_{\boldsymbol{\theta}} \ell_k(\boldsymbol{\theta})^\top \boldsymbol{\delta} - (\boldsymbol{\theta} - \boldsymbol{\mu}_{k-1})^\top \boldsymbol{\Lambda}_{k-1} \boldsymbol{\delta} + \eta(r^2 - \frac{1}{2} \boldsymbol{\delta}^\top \boldsymbol{\Lambda}_{k-1} \boldsymbol{\delta}). \tag{15}$$

We then take the derivative of $\tilde{\mathcal{L}}$ w.r.t. $\boldsymbol{\delta}$ and set it to zero:

$$\nabla_{\boldsymbol{\delta}} \tilde{\mathcal{L}}(\boldsymbol{\delta}, \eta) = \nabla_{\boldsymbol{\theta}} \ell_k(\boldsymbol{\theta}) - \boldsymbol{\Lambda}_{k-1}(\boldsymbol{\theta} - \boldsymbol{\mu}_{k-1}) - \eta \boldsymbol{\Lambda}_{k-1} \boldsymbol{\delta}' = 0. \tag{16}$$

Rearranging this equation gives

$$\boldsymbol{\delta} = \frac{1}{\eta} \left[ \boldsymbol{\Lambda}_{k-1}^{-1} \nabla_{\boldsymbol{\theta}} \ell_k(\boldsymbol{\theta}) - (\boldsymbol{\theta} - \boldsymbol{\mu}_{k-1}), \right]. \tag{17}$$

where $\eta$ itself depends on $r^2$ implicitly. Finally we define a learning rate parameter $\gamma = 1/\eta$ and arrive at the NCL learning rule:

$$\boldsymbol{\theta} \leftarrow \boldsymbol{\theta} + \gamma \left[ \boldsymbol{\Lambda}_{k-1}^{-1} \nabla_{\boldsymbol{\theta}} \ell_k(\boldsymbol{\theta}) - (\boldsymbol{\theta} - \boldsymbol{\mu}_{k-1}) \right]. \tag{18}$$

**Alternative derivation**   Here, we present an alternative derivation of the NCL learning rule. In this formulation, we seek to update the parameters of our model on task $k$ by maximizing $\mathcal{L}_k(\boldsymbol{\theta})$ subject to a constraint on the allowed change in the prior term. To find our parameter updates $\boldsymbol{\delta}$, we again solve a constrained optimization problem:

$$\boldsymbol{\delta} = \arg\min_{\boldsymbol{\delta}} \mathcal{L}_k(\boldsymbol{\theta}) + \nabla_{\boldsymbol{\theta}} \mathcal{L}_k(\boldsymbol{\theta})^\top \boldsymbol{\delta} \quad \text{such that} \quad \mathcal{C}(\boldsymbol{\delta}) \le r^2. \tag{19}$$

Here we define $\mathcal{C}(\boldsymbol{\delta})$ as the approximate change in log probability under the prior

$$\mathcal{C}(\boldsymbol{\delta}) = (\boldsymbol{\theta} + \boldsymbol{\delta} - \boldsymbol{\mu}_{k-1})^\top \boldsymbol{\Lambda}_{k-1}(\boldsymbol{\theta} + \boldsymbol{\delta} - \boldsymbol{\mu}_{k-1}) - (\boldsymbol{\theta} - \boldsymbol{\mu}_{k-1})^\top \boldsymbol{\Lambda}_{k-1}(\boldsymbol{\theta} - \boldsymbol{\mu}_{k-1}). \tag{20}$$

Following a similar derivation to above, we find the solution to this optimization problem as

$$\eta \boldsymbol{\delta} = \boldsymbol{\Lambda}_{k-1}^{-1} \nabla_{\boldsymbol{\theta}} \mathcal{L}_k(\boldsymbol{\theta}) - \eta(\boldsymbol{\theta} - \boldsymbol{\mu}_{k-1}) = \boldsymbol{\Lambda}_{k-1}^{-1} \nabla_{\boldsymbol{\theta}} \ell_k(\boldsymbol{\theta}) - (1 + \eta)(\boldsymbol{\theta} - \boldsymbol{\mu}_{k-1}) \tag{21}$$

for some Lagrange multiplier $\eta$. This gives rise to the update rule

$$\boldsymbol{\theta} \leftarrow \boldsymbol{\theta} + \gamma \left[ \boldsymbol{\Lambda}_{k-1}^{-1} \nabla_{\boldsymbol{\theta}} \ell_k(\boldsymbol{\theta}) - \lambda(\boldsymbol{\theta} - \boldsymbol{\mu}_{k-1}) \right] \tag{22}$$

for a learning rate parameter $\gamma$ and some choice of the parameter $\lambda$ that depends on both $\eta$ and $\gamma$. We recover the learning rule derived in Section 2.2 with the choice of $\lambda = 1$. In practice, $\lambda$ can also be treated as a hyperparameter to be optimized (Appendix I.1).

# B   Task details

**Split MNIST**   The split MNIST benchmark involves 5 tasks, each corresponding to the pairwise classification of two digits. The 10 digits of the MNIST dataset are randomly divided over the 5 tasks (i.e., for each random seed, this division can be different). During the incremental training protocol, these tasks are visited one after the other, followed by testing on all tasks. The original $28 \times 28$ pixel grey-scale images and the standard train/test-split are used, giving 60,000 training ($\sim$6,000 per digit) and 10,000 test images ($\sim$1,000 per digit).

**Split CIFAR-100**   The split CIFAR-100 benchmark consists of 10 tasks, with each task corresponding to a ten-way classification problem. The 100 classes of the CIFAR-100 dataset are randomly divided over the 10 tasks. Each network is trained on these tasks one after the other followed by testing on all tasks. The $32 \times 32$ pixel RGB-colour images are normalised by z-scoring each channel (using means and standard deviations calculated over the training set). We use the standard train/test-split, giving 500 training and 100 test images for each class.

**Stimulus-response tasks**   Here, we provide a brief overview of the six stimulus-response (SR) tasks. Detailed descriptions of the stimulus-response tasks used in this work can be found in the appendix of Yang et al. [51]. All tasks are characterized by a stimulus period and a response period, and some tasks include an additional delay period between the two. The duration of the stimulus and delay periods are variable across trials and drawn uniformly at random within an allowed range. During the stimulus period, the input to the network takes the form of $\boldsymbol{x} = (\cos\theta_{in}, \sin\theta_{in})$, where $\theta_{in} \in [0, 2\pi]$ is some stimulus drawn uniformly at random for each trial. An additional tonic input is provided to the network which indicates the identity of the task using a one-hot encoding. A constant input to a 'fixation channel' during the stimulus and delay periods signifies that the network output should be 0 in the response channels and 1 in a 'fixation channel'. During the response period, the fixation input is removed and the output should be 0 in the fixation channel. The target output in the response channels takes the form $\boldsymbol{y} = (\cos\theta_{out}, \sin\theta_{out})$ where $\theta_{out}$ is some target output direction described for each task below:

- **task 1 (fdgo)** During this task $\theta_{out} = \theta_{in}$ and there is no delay period.
- **task 2 (fdanti)** During this task $\theta_{out} = 2\pi - \theta_{in}$ and there is no delay period.
- **task 3 (delaygo)** During this task $\theta_{out} = \theta_{in}$ and there is a delay period separating the stimulus and response periods.
- **task 4 (delayanti)** During this task $\theta_{out} = 2\pi - \theta_{in}$ and there is a delay period separating the stimulus and response periods.
- **task 5 (dm1)** During this task, two stimuli are drawn from $[0, 2\pi]$ with different input magnitudes such that $\boldsymbol{x} = (m_1 \cos\theta_1 + m_2 \cos\theta_2, m_1 \sin\theta_1 + m_2 \sin\theta_2)$. $\theta_{out}$ is then the element in $(\theta_1, \theta_2)$ corresponding to the largest $m$.
- **task 6 (dm2)** As in 'dm1', but where the input is now provided through a separate input channel.

The loss for each task was computed as a mean squared error from the target output.

**SMNIST**   For this task set, we use the stroke MNIST dataset created by de Jong [4]. This consists of a series of digits, each of which is represented as a sequence of vectors $\{\boldsymbol{x}_t \in \mathbb{R}^4\}$. The first two columns take values in $[-1, 0, 1]$ and indicate the discretized displacement in the x and y direction at each time step. The last two columns are used for special 'end-of-line' inputs when the virtual pen is lifted from the paper for a new stroke to start, and an 'end-of-digit' input when the digit is finished. See de Jong [4] for further details about how the dataset was generated and formatted. In addition to the standard digits 0-9, we include two additional sets of digits:

- the digits 0-9 where the x and y directions have been swapped (i.e. the first two elements of $\boldsymbol{x}_t$ are swapped),
- the digits 0-9 where the x and y directions have been inverted (i.e. the first two elements of $\boldsymbol{x}_t$ are negated).

Furthermore, we omitted the initial entry of each digit corresponding to the 'start' location to increase task difficulty. We turned this dataset into a continual learning task by constructing five binary classification tasks for each set of digits: $\{[2, 3], [4, 5], [1, 7], [8, 9], [0, 6]\}$. Note that we have swapped the '1' and '6' from a standard split MNIST task to avoid including the 0 vs 1 classification task which we found to be too easy. For each trial, a digit was sampled at random from the corresponding dataset, and $x_t$ was provided as an input to the network at each time step corrupted by Gaussian noise with $\sigma = 1$. After the 'end-of-digit' input, a response period with a duration of 5 time steps followed. During this response period only, a cross-entropy loss was applied to the output units $y$ to train the network. During testing, digits were sampled from the separate test dataset and classification performance was quantified as the fraction of digits for which the correct class was assigned the highest probability in the last timestep of the response period. Task identity was provided to the network, which was used in the form of a multi-head output layer.

## C   Network architectures

**Feedforward network archictecture**   For split MNIST, all methods are compared using a fully-connected network with 2 hidden layers containing 400 units with ReLU non-linearities, followed by a softmax output layer.

For split CIFAR-100, the network consists of 5 pre-trained convolutional layers, 2 fully-connected layers with 2000 ReLU units each and a softmax output layer. The architecture of the convolutional layers and their pre-training protocol on the CIFAR-10 dataset are described in [45]. The only difference is that here we pre-train a new set of convolutional layers for each random seed, while in [45] the same set of pre-trained convolutional layers was used for all random seeds. For all compared methods, the pre-trained convolutional layers are frozen during the incremental training protocol.

The softmax output layer of the feedforward networks is treated differently depending on the continual learning setting [47]. In the task-incremental learning setting, there is a separate output layer for each task and only the output layer of the task under consideration is used at any given time (i.e., a multi-head output layer). In the domain-incremental learning setting, there is a single output layer that is shared between all tasks. In the class-incremental learning setting, there is one large output layer that spans all tasks and contains a separate output unit for each class.

**Recurrent network architecture**   The dynamics of the RNN used in Section 3.2 can be described by the following equations:

$$h_t = Hr_{t-1} + Gx_t + \xi_t = Wz_t + \xi_t \tag{23}$$

$$y_t \sim p(y_t | Cr_t) \tag{24}$$

where we define $r_t = \phi(h_t)$, $z_t = (r_{t-1}^\top, x_t^\top)^\top$, $W = (H^\top, G^\top)^\top$, and time is indexed by $t$. Here, $r \in \mathbb{R}^{N_{rec} \times 1}$ are the network activations, $x \in \mathbb{R}^{n_{in} \times 1}$ are the inputs, $y \in \mathbb{R}^{n_{out} \times 1}$ are the network outputs, and we refer to $Wz_t$ as the 'recurrent inputs' to the network. The noise model $p(y_t | Cr_t)$ may be a Gaussian distribution for a regression task or a categorical distribution for a classification task, and $\phi(h)$ is a nonlinearity that is applied to $h$ element-wise (in this work the ReLU function). The parameters of the RNN are given by $\theta = (W, C)$. The process noise $\{\xi_t\}$ are zero-mean Gaussian random variables with covariance matrices $\Sigma_t^\xi$. In this model, the log-likelihood of observing a sequence of outputs $y_1, \ldots, y_T$ given inputs $x_1, \ldots, x_T$ and $\xi_1, \ldots, \xi_T$ is given by

$$\ell(\theta) = \log p_\theta(\{y\} | \{x\}, \{\xi\}) = \log p(\{y\} | \{Cr\}), \tag{25}$$

where $p(y | Cr)$ may be a Gaussian distribution for a regression task or a categorical distribution for a classification task.

## D   KFAC approximation to the Fisher matrix

For all experiments in this work, we make a Kronecker-factored approximation to the FIM of each task $k$ in Equation 6. Concretely, we use the block-wise Kronecker-factored approximation to the FIM proposed in Section 3 of Martens and Grosse [15] for feedforward neural networks. For recurrent neural networks, we use the approximation presented in Section 3.4 of Martens et al. [14]. Both

approximations allow us to write the FIM on task $k$ as the Kronecker product $\boldsymbol{F}_k \approx \hat{\boldsymbol{A}}_k \otimes \hat{\boldsymbol{G}}_k$. For completeness, we derive the approximation for RNNs below. We refer the readers to Martens and Grosse [15] for details on derivations for feedforward networks.

**KFAC approximation for RNNs**   Recall from Appendix C that the log likelihood of observing a sequence of outputs $\boldsymbol{y}_1, \ldots, \boldsymbol{y}_T$ given inputs $\boldsymbol{x}_1, \ldots, \boldsymbol{x}_T$ and $\boldsymbol{\xi}_1, \ldots, \boldsymbol{\xi}_T$ is

$$\ell(\boldsymbol{W}, \boldsymbol{C}) = \sum_{t=1}^{T} \log p(\boldsymbol{y}_t | \boldsymbol{C}\boldsymbol{r}_t), \tag{26}$$

where $\boldsymbol{r}_t$ is completely determined by the dynamics of the network and the inputs. With a slight abuse of notation, we use $\overline{\boldsymbol{x}}$ to denote both $\partial \ell / \partial \boldsymbol{x}$ for vectors $\boldsymbol{x}$ and $\partial \ell / \partial \text{vec}(\boldsymbol{X})$ for matrices $\boldsymbol{X}$. In this section, it should be clear given the context whether $\overline{\boldsymbol{x}}$ is representing the gradient of $\mathcal{L}$ with respect to a vector or a vectorized matrix. Using these notations, we can write the gradient of $\mathcal{L}$ with respect to $\text{vec}(\boldsymbol{W})$ as :

$$\overline{\boldsymbol{w}} = \sum_{t=1}^{T} \overline{\boldsymbol{h}}_t \frac{\partial \boldsymbol{h}_t}{\partial \text{vec}(\boldsymbol{W})} = \sum_{t=1}^{T} \overline{\boldsymbol{h}}_t \boldsymbol{z}_t^{\top} = \sum_{t=1}^{T} \boldsymbol{z}_t \otimes \overline{\boldsymbol{h}}_t \tag{27}$$

which can be easily derived fom the backpropagation through time (BPTT) algorithm and the definition of a Kronecker product. Using this expression for $\overline{\boldsymbol{w}}$, we can write the FIM of $\boldsymbol{W}$ as:

$$\boldsymbol{F}_{\boldsymbol{W}} = \mathbb{E}_{\{(\boldsymbol{\xi}, \boldsymbol{x}, \boldsymbol{y})\} \sim \mathcal{M}} \left[ \overline{\boldsymbol{w}} \, \overline{\boldsymbol{w}}^{\top} \right] \tag{28}$$

$$= \mathbb{E} \left[ \left( \sum_{t=1}^{T} \boldsymbol{z}_t \otimes \overline{\boldsymbol{h}}_t \right) \left( \sum_{s=1}^{T} \boldsymbol{z}_s \otimes \overline{\boldsymbol{h}}_s \right)^{\top} \right] \tag{29}$$

$$= \sum_{t=1}^{T} \sum_{s=1}^{T} \mathbb{E} \left[ \left( \boldsymbol{z}_t \boldsymbol{z}_s^{\top} \right) \otimes \left( \overline{\boldsymbol{h}}_t \overline{\boldsymbol{h}}_s^{\top} \right) \right]. \tag{30}$$

Here the expectations are taken with respect to the model distribution. Unfortunately, computing $\boldsymbol{F}_{\boldsymbol{W}}$ can be prohibitively expensive. First, the number of computations scales quadratically with the length of the input sequence $T$. Second, for networks of dimension $n$, there are $n^4$ entries in the Fisher matrix which can therefore be too large to store in memory, let alone perform any useful computations with it. For this reason, we follow Martens et al. [14] and make the following three assumptions in order to derive a tractable Kronecker-factored approximation to the Fisher. The first assumption we make is that the input and recurrent activty $\boldsymbol{z}_t$ is uncorrelated with the adjoint activations $\overline{\boldsymbol{h}}_t$:

$$\boldsymbol{F}_{\boldsymbol{W}} \approx \sum_{t=1}^{T} \sum_{s=1}^{T} \mathbb{E} \left[ \boldsymbol{z}_t \boldsymbol{z}_s^{\top} \right] \otimes \mathbb{E}_{\{(\boldsymbol{\xi}, \boldsymbol{x}, \boldsymbol{y})\} \sim \mathcal{M}} \left[ \overline{\boldsymbol{h}}_t \overline{\boldsymbol{h}}_s^{\top} \right]. \tag{31}$$

Note that this approximation is exact when the network dynamics are linear (i.e., $\phi(\boldsymbol{x}) = \boldsymbol{x}$). The second assumption that we make is that both the forward activity $\boldsymbol{z}_t$ and adjoint activity $\overline{\boldsymbol{h}}_t$ are temporally homogeneous. That is, the statistical relationship between $\boldsymbol{z}_t$ and $\boldsymbol{z}_s$ only depends on the difference $\tau = s - t$, and similarly for that between $\overline{\boldsymbol{h}}_t$ and $\overline{\boldsymbol{h}}_s$. Defining $\mathcal{A}_\tau = \mathbb{E} \left[ \boldsymbol{z}_s \boldsymbol{z}_{s+\tau}^{\top} \right]$ and similarly $\mathcal{G}_\tau = \mathbb{E} \left[ \overline{\boldsymbol{h}}_s \overline{\boldsymbol{h}}_{s+\tau}^{\top} \right]$, we have $\mathcal{A}_{-\tau} = \mathcal{A}_\tau^{\top}$ and $\mathcal{G}_{-\tau} = \mathcal{G}_\tau$. Using these expressions, we can further approximate the Fisher as:

$$\boldsymbol{F}_{\boldsymbol{W}} \approx \sum_{\tau=-T}^{T} (T - |\tau|) \mathcal{A}_\tau \otimes \mathcal{G}_\tau. \tag{32}$$

The third and final approximation we make is that $\mathcal{A}_\tau \approx 0$ and $\mathcal{G}_\tau \approx 0$ for $\tau \neq 0$. In other words, we assume the forward activity $\boldsymbol{z}_t$ and adjoint activity $\overline{\boldsymbol{h}}_t$ are approximately indendent across time. This gives the final expression:

$$\boldsymbol{F}_{\boldsymbol{W}} \approx \mathbb{E}\left[T\right] \mathbb{E} \left[ \boldsymbol{z}\boldsymbol{z}^{\top} \right] \otimes \mathbb{E} \left[ \overline{\boldsymbol{h}} \, \overline{\boldsymbol{h}}^{\top} \right] = \hat{\boldsymbol{A}}_{\boldsymbol{W}} \otimes \hat{\boldsymbol{G}}_{\boldsymbol{W}}, \tag{33}$$

where we have also taken an expectation over the sequence length $T$ to account for variable sequence lengths in the data. Following a similar derivation, we can approximate the Fisher of $\boldsymbol{C}$ as:

$$\boldsymbol{F}_C \approx \mathbb{E}\left[T\right] \mathbb{E}\left[\boldsymbol{r}\boldsymbol{r}^\top\right] \otimes \mathbb{E}\left[\overline{\boldsymbol{y}}\,\overline{\boldsymbol{y}}^\top\right] = \hat{\boldsymbol{A}}_C \otimes \hat{\boldsymbol{G}}_C. \tag{34}$$

The quality of these assumptions and comparisons with the 'approximate Fisher matrices' used in OWM and DOWM are discussed in Appendix F.

## E   Implementation

---
**Algorithm 1:** NCL with momentum

---
1  **input:** $f$ (network), $\{\mathcal{D}_k\}_{k=1}^K$, $\alpha$, $p_w$ (prior), $B$ (batch size), $\gamma$ (learning rate), $\theta_0$, $\rho$
2  **initialize:** $\boldsymbol{A}_\theta \leftarrow p_w \boldsymbol{I}$, $\boldsymbol{G}_\theta \leftarrow p_w \boldsymbol{I}$,
3  **initialize:** $\theta_1 \leftarrow \theta_0$, **initialize:** $\boldsymbol{M}_\theta \leftarrow \text{zeros\_like}(\theta_0)$,                  // Gradient momentum
4  **for** $k = 1 \ldots K$ **do**
5    $\quad \widetilde{\boldsymbol{A}}, \widetilde{\boldsymbol{G}} \leftarrow \text{nearest\_kf\_sum}(\boldsymbol{A}_\theta \otimes \boldsymbol{G}_\theta, \alpha\boldsymbol{I} \otimes \alpha\boldsymbol{I})$                  // Appendix G
6    $\quad \boldsymbol{P}_L \leftarrow \widetilde{\boldsymbol{G}}^{-1}$
7    $\quad \boldsymbol{P}_R \leftarrow \widetilde{\boldsymbol{A}}^{-1}$
8    $\quad$ **while** *not converged* **do**
9      $\quad\quad \{\boldsymbol{x}^{(i)}, \boldsymbol{y}^{(i)}\}_{i=1}^B \sim \mathcal{D}_k$                  // Input and target output
10     $\quad\quad$ **for** $i = 1, \ldots, B$ **do**
11       $\quad\quad\quad \hat{\boldsymbol{y}}^{(i)} = f(\boldsymbol{x}^{(i)}, \theta_k)$                  // Empirical output
12     $\quad\quad \ell = \sum_i^B \log p(\boldsymbol{y}^{(i)}|\hat{\boldsymbol{y}}^{(i)})/B$                  // Loss
13
14     $\quad\quad$ % Build up momentum
15     $\quad\quad \boldsymbol{M}_\theta \leftarrow \rho\boldsymbol{M}_\theta + \nabla_\theta \ell + \boldsymbol{G}_\theta(\theta_k - \theta_{k-1})\boldsymbol{A}_\theta$
16
17     $\quad\quad$ % Update model parameters
18     $\quad\quad \theta_k \leftarrow \theta_k - \gamma p_w^2 \, \boldsymbol{P}_L \, \boldsymbol{M}_\theta \boldsymbol{P}_R$
19   $\quad$ % Update Fisher matrix components
20   $\quad$ Compute $\hat{\boldsymbol{A}}_k$ and $\hat{\boldsymbol{G}}_k$                  // Appendix D
21   $\quad \boldsymbol{A}_\theta, \boldsymbol{G}_\theta \leftarrow \text{nearest\_kf\_sum}(\boldsymbol{A}_\theta \otimes \boldsymbol{G}_\theta, \hat{\boldsymbol{A}}_k \otimes \hat{\boldsymbol{G}}_k)$                  // Appendix G

---

In this section we discuss various implementation details for NCL. Algorithm 1 provides an overview of the algorithm in the form of pseudocode. For numerical stability, we add $\alpha^2\boldsymbol{I}$ to the precision matrix $\boldsymbol{\Lambda}_{k-1}$ before computing the projection matrices $\boldsymbol{P}_L$ and $\boldsymbol{P}_R$. In general, we set the prior over the parameters $\boldsymbol{\theta}$ when learning the first task as $p(\boldsymbol{\theta}) = \mathcal{N}(\boldsymbol{0}; p_w^{-2}\boldsymbol{I})$.

**Feedforward networks**   By default, we set $p_w^{-2}$ to be approximately the number of samples that the learner sees in each task, corresponding to a unit Gaussian prior before normalizing our precision matrices by the amount of data seen in each task (here, $p_w^{-2} = 12000$ for split MNIST and $p_w^{-2} = 5000$ for split CIFAR-100). We also consider hyperparameter optimizations over $p_w^{-2}$ by trying different values on a log scale from $10^2$ to $10^{11}$ with a random seed not included during the evaluation (see Appendix I.2). We use $\alpha = 10^{-10}$ and $\lambda = 1$ for all experiments.

For all experiments with feedforward networks, we use a batch size of 256 and we train for either 2000 iterations per task (split MNIST) or 5000 iterations per task (split CIFAR-100). For NCL and OWM, we train with momentum ($\rho = 0.9$) and a learning rate of $\gamma = 0.05$. For SI, EWC and KFAC, we train using the Adam optimizer ($\beta_1 = 0.9$, $\beta_2 = 0.999$) with learning rate of $\gamma = 0.001$ (split MNIST) or $\gamma = 0.0001$ (split CIFAR-100). All models were trained on single GPUs with training times of 10-100 minutes.

**RNNs**   We again set $p_w^{-2}$ approximately equal to the number of samples that the learner sees in each task, corresponding to a unit Gaussian prior before normalizing our precision matrices by the amount of data seen in each task (here, $p_w^{-2} = 10^6$ for the stimulus-response task and $p_w^{-2} = 6000$ for SMNIST).

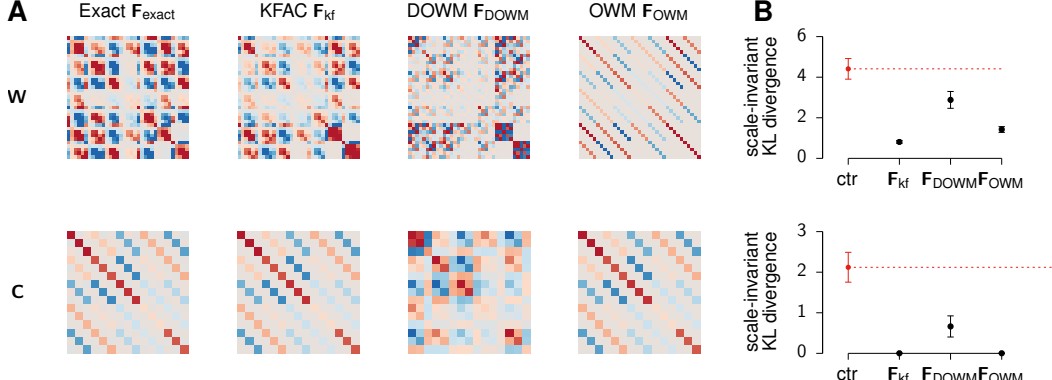

Figure 5: **Comparison of projection matrices.** In a Bayesian framework, we can formalize what is meant by directions 'important for previous tasks' as those that are strongly constrained by the prior $p(\boldsymbol{\theta}|\mathcal{D}_{1:k-1})$. To see how this compares with OWM and DOWM, we considered the Kronecker-structured precision matrices $\boldsymbol{F}_{\text{approx}}$ implied by the projection matrices $\boldsymbol{P}_R$ and $\boldsymbol{P}_L$ for each method and related them to the exact Fisher matrix $\boldsymbol{F}_{\text{exact}}$ in a linear recurrent network. **(A; top)** $\boldsymbol{F}_{\text{exact}}$ (left) for $\boldsymbol{W}$ as well as the approximations to $\boldsymbol{F}_{\text{exact}}$ provided by our Kronecker-factored approximation (KFAC; $\boldsymbol{F}_{\text{kf}}$), DOWM ($\boldsymbol{F}_{\text{DOWM}}$), and OWM ($\boldsymbol{F}_{\text{OWM}}$). **(B; top)** Scale-invariant KL-divergence (Equation 62) between $\mathcal{N}(\boldsymbol{\mu}, \boldsymbol{F}_{\text{exact}}^{-1})$ and $\mathcal{N}(\boldsymbol{\mu}, \boldsymbol{F}_{\text{approx}}^{-1})$ for each approximation. Red horizontal line indicates the mean value obtained from $\boldsymbol{F}_{\text{approx}} = \boldsymbol{R}\boldsymbol{F}_{\text{exact}}\boldsymbol{R}^\top$ where $\boldsymbol{R}$ is a random rotation matrix (averaged over 500 random samples). **(Bottom)** Same as (A–B) but for the readout matrix $\boldsymbol{C}$.

We used momentum ($\rho = 0.9$) in all our experiments involving NCL, OWM and DOWM, as is also done in Duncker et al. [7]. We found that the use of momentum greatly speeds up convergence in practice.

All models were trained on single GPUs with training times of 10-100 minutes depending on the task set and model size. We used a training batch size of 32 for the stimulus-response tasks and 256 for the SMNIST tasks. In all cases, we used a test batch size of 2048 for evaluation and for computing projection and Fisher matrices. We used a learning rate of $\gamma = 0.01$ for SMNIST and $\gamma = 0.005$ for the stimulus-response tasks across all projection-based methods. We used a learning rate of $\gamma = 0.001$ for KFAC with the Adam optimizer. All models were trained on $10^6$ data samples per task. A hyperparameter optimization over $\alpha$ for the projection-based methods and $\lambda$ for KFAC with Adam is provided in Appendix I.2.

## F  Relation to projection-based continual learning

In this section, we further elaborate on the intuition that projection-based continual learning methods such as Orthogonal Weight Modification (OWM; 26) may be viewed as variants of NCL with particular approximations to the prior Fisher matrix. These approaches are typically motivated as a way to restrict parameter changes in a neural network that is learning a new task to subspaces orthogonal to those used in previous tasks.

For example, to solve the continual learning problem in RNNs as described in Appendix C, Duncker et al. [7] proposed a projected gradient algorithm (DOWM) that restricts modifications to the recurrent/input weight matrix $\boldsymbol{W}$ on task $k+1$ to column and row spaces of $\boldsymbol{W}$ that are not heavily "used" in the first $k$ tasks. Specifically, they concatenate input and recurrent activity $\boldsymbol{z}_t$ across the first $k$ tasks into a matrix $\boldsymbol{Z}_{1:k}$. They use $\boldsymbol{Z}_{1:k}$ and $\boldsymbol{W}\boldsymbol{Z}_{1:k}$ as estimates of the row and column spaces of $\boldsymbol{W}$ that are important for the first $k$ tasks. They proceed to construct the following projection matrices:

$$\boldsymbol{P}_z^{1:k} = \boldsymbol{Z}_{1:k}(\boldsymbol{Z}_{1:k}\boldsymbol{Z}_{1:k}^\top + \alpha\boldsymbol{I})^{-1}\boldsymbol{Z}_{1:k}^\top \tag{35}$$

$$\approx k\alpha\left(\mathbb{E}\left[\boldsymbol{z}\boldsymbol{z}^\top\right] + \alpha\boldsymbol{I}\right)^{-1} \tag{36}$$

$$\boldsymbol{P}_{wz}^{1:k} = \boldsymbol{W}\boldsymbol{Z}_{1:k}(\boldsymbol{W}\boldsymbol{Z}_{1:k}\boldsymbol{Z}_{1:k}^\top\boldsymbol{W}^\top + \alpha\boldsymbol{I})^{-1}(\boldsymbol{W}\boldsymbol{Z}_{1:k})^\top \tag{37}$$

$$\approx k\alpha\left(\boldsymbol{W}\mathbb{E}\left[\boldsymbol{z}\boldsymbol{z}^\top\right]\boldsymbol{W}^\top + \alpha\boldsymbol{I}\right)^{-1}, \tag{38}$$

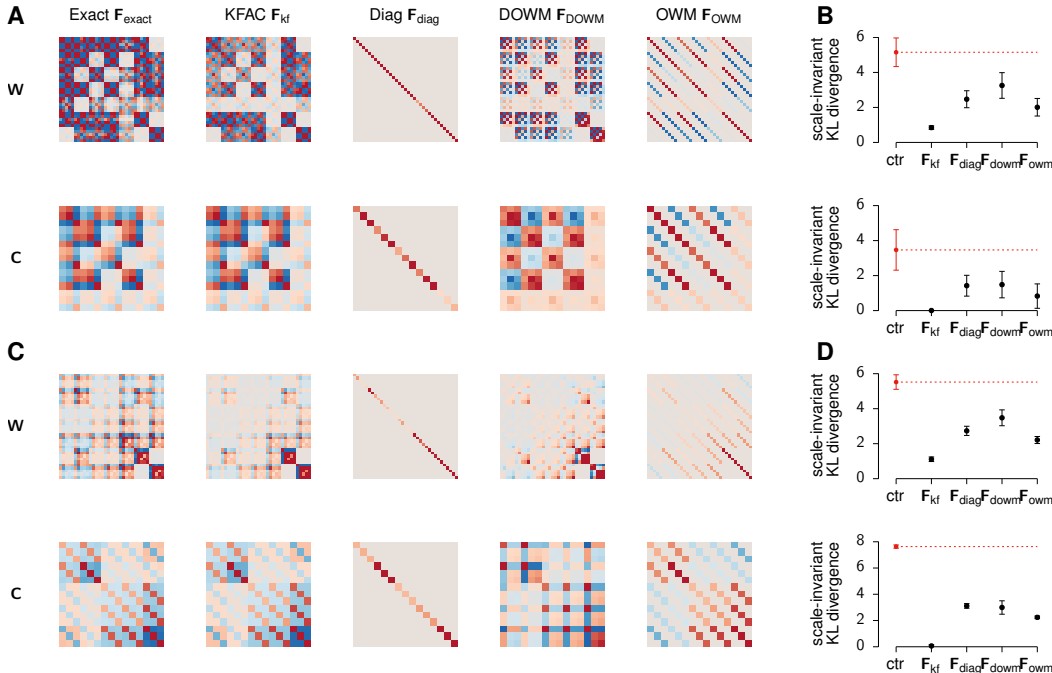

Figure 6: **Comparison of Fisher Approximations in a Linear RNN with rotated Gaussian and categorical likelihoods.** (A) Exact and approximations to the Fisher information matrix of the recurrent and input weight matrix $W$ (left) and the linear readout $C$ (bottom) of a linear recurrent neural network with Gaussian noise and non-diagonal noise covariance $\Sigma$. From the left: exact Fisher information matrix $F_{\text{exact}}$, Kronecker-Factored approximation ($F_{\text{kf}}$; KFAC), Diagonal ($F_{\text{diag}}$), DOWM ($F_{\text{DOWM}}$), and OWM ($F_{\text{OWM}}$). (B) Scale-invariant KL-divergence between $\mathcal{N}(0, F_{\text{exact}}^{-1})$ and $\mathcal{N}(0, F^{-1})$ for $F \in \{F_{\text{kf}}, F_{\text{diag}}, F_{\text{DOWM}}, F_{\text{OWM}}\}$. Red horizontal lines indicate the mean value obtained from $F_{\text{approx}} = R F_{\text{exact}} R^\top$ where $R$ is a random rotation matrix (averaged over 500 random samples). (C-D) As in (A-B), now for a categorical noise model $p(y|Cr) = \text{Cat}(\text{softmax}(Cr))$.

which are used to derive update rules for $W$ as:

$$\text{vec}(\Delta W) \propto \left( P_z^{1:k} \otimes P_{wz}^{1:k} \right) \overline{w} \tag{39}$$

$$\propto \left( \mathbb{E}\left[ z z^\top \right] + \alpha I \right)^{-1} \otimes \left( W \mathbb{E}\left[ z z^\top \right] W^\top + \alpha I \right)^{-1} \overline{w} \tag{40}$$

where $\overline{w} = \text{vec}(\nabla_W \ell_{k+1}(W, C))$. These projection matrices restrict changes in the row and column space of $W$ to be orthogonal to $Z_{1:k}$ and $W Z_{1:k}$ respectively. Similar update rules can be defined for $C$. Zeng et al. [26] propose a similar projection-based learning rule (OWM) in feedforward networks, which only restricts changes in the row-space of the weight parameters (i.e., $P_{wz} = I$).

With a scaled additive approximation to the sum of Kronecker products (see Appendix G), the NCL update rule on task $k + 1$ is given by

$$\text{vec}(\Delta W) \propto \left( \mathbb{E}\left[ z z^\top \right] + \pi \alpha I \right)^{-1} \otimes \left( \mathbb{E}\left[ \overline{h}\,\overline{h}^\top \right] + \frac{1}{\pi} \alpha I \right)^{-1} \overline{w} + (\text{vec}(W_k) - \text{vec}(W)). \tag{41}$$

We see that this NCL update rule looks similar to the OWM and DOWM update steps, and that they share the same projection matrix in the row-space $P_z$ when $\pi = 1$. The methods proposed by Duncker et al. [7] and Zeng et al. [26] can thus be seen as approximations to NCL with a Kronecker structured Fisher matrix. However, we also note that OWM and DOWM do not include the regularization term $(\text{vec}(W_k) - \text{vec}(W))$. This implies that while OWM and DOWM encourage parameter updates along flat directions of the prior, the performance of these methods may deteriorate in the limit of infinite training duration if a local minimum of task $k$ is not found in a flat subspace of previous tasks (c.f. Figure 1).

To further emphasize the relationship between OWM, DOWM and NCL, we compared the approximations to the Fisher matrix $F_{\text{approx}} = P_R^{-1} \otimes P_L^{-1}$ implied by the projection matrices of these

methods (Figure 5). Here we found that OWM and DOWM provided reasonable approximations to the true Fisher matrix with both Gaussian (Figure 5) and categorical (Figure 6) observation models. This motivates a Bayesian interpretation of these methods as using an approximate prior precision matrix to project gradients, similar to the derivation of NCL in Appendix A. Here it is also worth noting that while we use an optimal sum of Kronecker factors to update the prior precision after each task in NCL (Appendix G), OWM and DOWM simply sum their Kronecker factors. In the case of OWM, this is in fact an exact approximation to the sum of the Kronecker products since the right Kronecker factor is in this case a constant matrix $\boldsymbol{I}$. For DOWM, summing the individual Kronecker factors does not provide an optimal approximation to the sum of the Kronecker products, but our results in Appendix G suggest that it is a fairly reasonable approximation up to a scale factor which can be absorbed into the learning rate.

Another recent projection-based approach to continual learning developed by Saha et al. [40] restricts parameter updates to occur in a subspace of the full parameter space deemed important for previous tasks. This method, known as 'Gradient projection memory' (GPM), is similar to OWM but with a hard cut-off separating 'important' from 'unimportant' directions of parameter space. The important subspace is in this case determined by thresholding the singular values of the activity matrix $\boldsymbol{Z}_k$. GPM can thus be seen as a discretized version of OWM with a projection matrix constituting a binary approximation to the prior Fisher matrix.

## G  Kronecker-factored approximation to the sums of Kronecker Products

In this section, we consider three different Kronecker-factored approximations to the sum of two Kronecker products:
$$\boldsymbol{X} \otimes \boldsymbol{Y} \approx \boldsymbol{Z} = \boldsymbol{A} \otimes \boldsymbol{B} + \boldsymbol{C} \otimes \boldsymbol{D}. \tag{42}$$
In particular, we consider the special case where $\boldsymbol{A} \in \mathbb{R}^{n \times n}$, $\boldsymbol{B} \in \mathbb{R}^{m \times m}$, $\boldsymbol{C} \in \mathbb{R}^{n \times n}$, and $\boldsymbol{D} \in \mathbb{R}^{m \times m}$ are symmetric positive-definite. $\boldsymbol{Z}$ will not in general be a Kronecker product, but for computational reasons it is desirable to approximate it as one to avoid computing or storing a full-sized precision matrix.

**Scaled additive approximation**  The first approximation we consider was proposed by Martens and Grosse [15]. They propose to approximate the sum with
$$\boldsymbol{Z} \approx (\boldsymbol{A} + \pi \boldsymbol{C}) \otimes (\boldsymbol{B} + \frac{1}{\pi}\boldsymbol{D}), \tag{43}$$
where $\pi$ is a scalar parameter. Using the triangle inequality, Martens and Grosse [15] derived an upper-bound to the norm of the approximation error
$$\|\boldsymbol{Z} - (\boldsymbol{A} + \pi \boldsymbol{C}) \otimes (\boldsymbol{B} + \frac{1}{\pi}\boldsymbol{D})\| \tag{44}$$
$$= \|\frac{1}{\pi}\boldsymbol{A} \otimes \boldsymbol{D} + \pi \boldsymbol{C} \otimes \boldsymbol{B}\| \tag{45}$$
$$\leq \frac{1}{\pi}\|\boldsymbol{A} \otimes \boldsymbol{D}\| + \pi\|\boldsymbol{C} \otimes \boldsymbol{B}\| \tag{46}$$
for any norm $\| \cdot \|$. They then minimize this upper-bound with respect to $\pi$ to find the optimal $\pi$:
$$\pi = \sqrt{\frac{\|\boldsymbol{C} \otimes \boldsymbol{B}\|}{\|\boldsymbol{A} \otimes \boldsymbol{D}\|}}. \tag{47}$$
As in [31], we use a trace norm in bounding the approximation error, and noting that $\text{Tr}(\boldsymbol{X} \otimes \boldsymbol{Y}) = \text{Tr}(\boldsymbol{X})\text{Tr}(\boldsymbol{Y})$, we can compute the optimal $\pi$ as:
$$\pi = \sqrt{\frac{\text{Tr}(\boldsymbol{B})\text{Tr}(\boldsymbol{C})}{\text{Tr}(\boldsymbol{A})\text{Tr}(\boldsymbol{D})}}. \tag{48}$$

**Minimal mean-squared error**  The second approximation we consider was originally proposed by Van Loan and Pitsianis [25]. In this case, we approximate the sum of Kronecker products by

minimizing a mean squared loss:

$$\boldsymbol{X}, \boldsymbol{Y} = \underset{\boldsymbol{X}, \boldsymbol{Y}}{\arg\min} \|\boldsymbol{Z} - \boldsymbol{X} \otimes \boldsymbol{Y}\|_F^2 \tag{49}$$

$$= \underset{\boldsymbol{X}, \boldsymbol{Y}}{\arg\min} \|\mathcal{R}(\boldsymbol{A} \otimes \boldsymbol{B}) + \mathcal{R}(\boldsymbol{C} \otimes \boldsymbol{D}) - \mathcal{R}(\boldsymbol{X} \otimes \boldsymbol{Y})\|_F^2 \tag{50}$$

$$= \underset{\boldsymbol{X}, \boldsymbol{Y}}{\arg\min} \|\text{vec}(\boldsymbol{A})\text{vec}(\boldsymbol{B})^\top + \text{vec}(\boldsymbol{C})\text{vec}(\boldsymbol{D})^\top - \text{vec}(\boldsymbol{X})\text{vec}(\boldsymbol{Y})^\top\|_F^2, \tag{51}$$

where $\mathcal{R}(\boldsymbol{A} \otimes \boldsymbol{B}) = \text{vec}(\boldsymbol{A})\text{vec}(\boldsymbol{B})^\top$ is the rearrangement operator [25]. The optimization problem thus involves finding the best rank-one approximation to a rank-2 matrix. This can be solved efficiently using a singular value decomposition (SVD) without ever constructing an $n^2 \times m^2$ matrix (see Algorithm 2 for details).

---

**Algorithm 2:** Mean-squared error approximation of the sum of Kronecker products

1 **input:** $\boldsymbol{A}, \boldsymbol{B}, \boldsymbol{C}, \boldsymbol{D}$
2 $a \leftarrow \text{vec}(\boldsymbol{A}), b \leftarrow \text{vec}(\boldsymbol{B}), c \leftarrow \text{vec}(\boldsymbol{C}), d \leftarrow \text{vec}(\boldsymbol{D})$      // Vectorize $\boldsymbol{A}, \boldsymbol{B}, \boldsymbol{C}, \boldsymbol{D}$
3 $\boldsymbol{Q}, \_ \leftarrow \text{QR}([a; c])$      // Orthogonal basis for $a$ and $c$ in $\mathbb{R}^{n^2 \times 2}$
4 $\boldsymbol{H} \leftarrow (\boldsymbol{Q}^\top a)b^\top + (\boldsymbol{Q}^\top c)d^\top$
5 $\boldsymbol{U}, s, \boldsymbol{V}^\top \leftarrow \text{SVD}(\boldsymbol{H})$
6 $y \leftarrow$ first column of $\sqrt{s_1}\boldsymbol{V}$
7 $x \leftarrow$ first column of $\sqrt{s_1}\boldsymbol{Q}\boldsymbol{U}$
8 $\boldsymbol{X} \leftarrow \text{reshape}(x, (n, n)), \boldsymbol{Y} \leftarrow \text{reshape}(y, (m, m))$

---

**Minimal KL-divergence**    In this paper, we propose an alternative approximation to $\boldsymbol{Z}$ motivated by the fact that $\boldsymbol{X} \otimes \boldsymbol{Y}$ is meant to approximate the precision matrix of the approximate posterior after learning task $k$. We thus define two multivariate Gaussian distributions $q(\boldsymbol{w}) = \mathcal{N}(\boldsymbol{w}; \boldsymbol{\mu}, \boldsymbol{X} \otimes \boldsymbol{Y})$ and $p(\boldsymbol{w}) = \mathcal{N}(\boldsymbol{w}; \boldsymbol{\mu}, \boldsymbol{Z})$ (note that the mean of these distributions are found in NCL by gradient-based optimization). We are interested in finding the matrices $\boldsymbol{X}$ and $\boldsymbol{Y}$ that minimize the KL-divergence between the two distributions

$$2D_{\text{KL}}(q\|p) = \log|\boldsymbol{X} \otimes \boldsymbol{Y}| - \log|\boldsymbol{Z}| + \text{Tr}(\boldsymbol{Z}(\boldsymbol{X} \otimes \boldsymbol{Y})^{-1}) - d \tag{52}$$

$$= m\log|\boldsymbol{X}| + n\log|\boldsymbol{Y}| + \text{Tr}(\boldsymbol{A}\boldsymbol{X}^{-1} \otimes \boldsymbol{B}\boldsymbol{Y}^{-1}) + \text{Tr}(\boldsymbol{C}\boldsymbol{X}^{-1} \otimes \boldsymbol{D}\boldsymbol{Y}^{-1}) - d \tag{53}$$

$$= -m\log|\boldsymbol{X}^{-1}| - n\log|\boldsymbol{Y}^{-1}| + \text{Tr}(\boldsymbol{A}\boldsymbol{X}^{-1})\text{Tr}(\boldsymbol{B}\boldsymbol{Y}^{-1}) \tag{54}$$

$$+ \text{Tr}(\boldsymbol{C}\boldsymbol{X}^{-1})\text{Tr}(\boldsymbol{D}\boldsymbol{Y}^{-1}) - d \tag{55}$$

where $d = nm$. Differentiating with respect to $\boldsymbol{X}^{-1}$, and $\boldsymbol{Y}^{-1}$ and setting the result to zero, we get

$$0 = \frac{\partial D_{\text{KL}}(q\|p)}{\partial \boldsymbol{X}^{-1}} = \frac{1}{2}\left[-m\boldsymbol{X} + \text{Tr}(\boldsymbol{B}\boldsymbol{Y}^{-1})\boldsymbol{A} + \text{Tr}(\boldsymbol{D}\boldsymbol{Y}^{-1})\boldsymbol{C}\right] \tag{56}$$

$$0 = \frac{\partial D_{\text{KL}}(q\|p)}{\partial \boldsymbol{Y}^{-1}} = \frac{1}{2}\left[-n\boldsymbol{Y} + \text{Tr}(\boldsymbol{A}\boldsymbol{X}^{-1})\boldsymbol{B} + \text{Tr}(\boldsymbol{C}\boldsymbol{X}^{-1})\boldsymbol{D}\right]. \tag{57}$$

Rearranging these equations, we find the self-consistency equations:

$$\boldsymbol{X} = \frac{1}{m}\left[\text{Tr}(\boldsymbol{B}\boldsymbol{Y}^{-1})\boldsymbol{A} + \text{Tr}(\boldsymbol{D}\boldsymbol{Y}^{-1})\boldsymbol{C}\right] \tag{58}$$

$$\boldsymbol{Y} = \frac{1}{n}\left[\text{Tr}(\boldsymbol{A}\boldsymbol{X}^{-1})\boldsymbol{B} + \text{Tr}(\boldsymbol{D}\boldsymbol{X}^{-1})\boldsymbol{D}\right]. \tag{59}$$

This shows that the optimal $\boldsymbol{X}$ ($\boldsymbol{Y}$) is a linear combination of $\boldsymbol{A}$ and $\boldsymbol{C}$ ($\boldsymbol{B}$ and $\boldsymbol{D}$). It is unclear whether we can solve for $\boldsymbol{X}$ and $\boldsymbol{Y}$ analytically in Equation 58 and Equation 59. However, we can find $\boldsymbol{X}$ and $\boldsymbol{Y}$ numerically by iteratively applying the following update rules:

$$\boldsymbol{X}_{k+1} = (1 - \beta)\boldsymbol{X}_k + \frac{\beta}{m}\left(\text{Tr}(\boldsymbol{B}\boldsymbol{Y}_k^{-1})\boldsymbol{A} + \text{Tr}(\boldsymbol{D}\boldsymbol{Y}_k^{-1})\boldsymbol{C}\right) \tag{60}$$

$$\boldsymbol{Y}_{k+1} = (1 - \beta)\boldsymbol{Y}_k + \frac{\beta}{n}\left(\text{Tr}(\boldsymbol{A}\boldsymbol{X}_k^{-1})\boldsymbol{C} + \text{Tr}(\boldsymbol{C}\boldsymbol{X}_k^{-1})\boldsymbol{D}\right) \tag{61}$$

for initial guesses $\boldsymbol{X}_0$ and $\boldsymbol{Y}_0$. In practice, we initialize using the scaled additive approximation and find that the algorithm converges with $\beta = 0.3$ after tens of iterations.

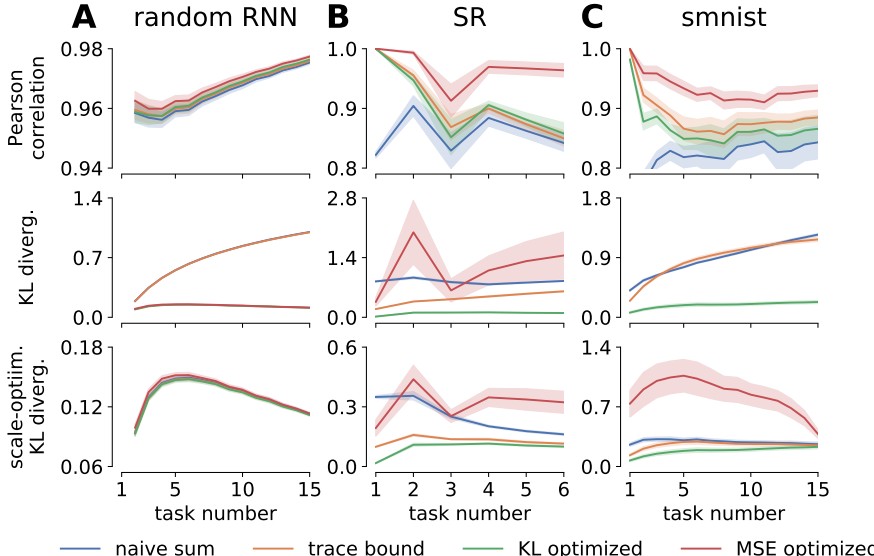

Figure 7: **Comparison of different Kronecker approximations to consecutive sums of two Kronecker products.** **(A)** Comparison of approximations for Fisher matrices computed from random RNNs with dynamics as described in Section 3.2. All similarity/distance measures are computed between the true sum $\sum_{k'}^{k} \boldsymbol{F}_{k'}$ and each iterative approximation. **(B)** As in (A) for the Fisher matrices from the stimulus-response tasks, here trained with 50 hidden units to make the computation of the true sum tractable. **(C)** As in (A) for the Fisher matrices from the SMNIST tasks. Note that the KL divergence for the MSE-minimizing approximation is not shown in panel 2 as it is an order of magnitude larger than the alternatives and thus does not fit on the axis.

**Comparisons**  To compare different approximations of the precision matrix to the posterior, we consider Kronecker structured Fisher matrices from (i) a random RNN model, (ii) the Fishers learned in the stimulus-response tasks, and (iii) the Fishers learned in the SMNIST tasks. We then iteratively update $\boldsymbol{\Lambda}_k \approx \boldsymbol{\Lambda}_{k-1} + \boldsymbol{F}_k$, approximating this sum using each of the approaches described above as well as a naive unweighted sum of the pairs of Kronecker factors. We compare these approximations using three different metrics: the correlation with the true sum of Kronecker products $\sum_{k'}^{k} \boldsymbol{F}_{k'}$ (Figure 7, top row), the KL divergence from the true sum (Figure 7, middle row), and the scale-optimized KL divergence from the true sum (Figure 7, bottom row). Here we define the scale-optimized KL divergence as

$$\mathrm{KL}_\lambda[\boldsymbol{\Lambda}_1 || \boldsymbol{\Lambda}_2] = \min_\lambda \mathrm{KL}[\lambda \boldsymbol{\Lambda}_1 || \boldsymbol{\Lambda}_2] \tag{62}$$

$$= \frac{1}{2} \left( \log \frac{|\boldsymbol{\Lambda}_1|}{|\boldsymbol{\Lambda}_2|} + d \log \frac{\mathrm{Tr}(\boldsymbol{\Lambda}_1^{-1} \boldsymbol{\Lambda}_2)}{d} \right), \tag{63}$$

where $d$ is the dimensionality of the precision matrices $\boldsymbol{\Lambda}_1$ and $\boldsymbol{\Lambda}_2$ and we take $\mathrm{KL}[\boldsymbol{\Lambda}_1, \boldsymbol{\Lambda}_2] = D_{\mathrm{KL}}(\mathcal{N}(\boldsymbol{0}, \boldsymbol{\Lambda}_1^{-1}) || \mathcal{N}(\boldsymbol{0}, \boldsymbol{\Lambda}_2^{-1}))$. This is a useful measure since a scaling of the approximate prior does not change the subspaces that are projected out in the weight projection methods but merely scales the learning rate. By contrast in NCL, having an appropriate scaling is useful for a consistent Bayesian interpretation.

We find that all the methods yield reasonable correlations and scale-optimized KL divergences between the true sum of Kronecker products and the approximate sum, although the L2-optimized approximation tends to have a slightly better correlation and slightly worse scaled KL (Figure 7, red). However, the KL-optimized Kronecker sum greatly outperforms the other methods as quantified by the regular KL divergence and is the method used in this work since it is relatively cheap to compute and only needs to be computed once per task (Figure 7, green).

## H   Natural gradient descent and the Fisher Information Matrix

When optimizing a model with stochastic gradient descent, the parameters $\boldsymbol{\theta}$ are generally changed in the direction of steepest gradient of the loss function $\mathcal{L}$:

$$\boldsymbol{g} = \nabla_{\boldsymbol{\theta}} \mathcal{L}. \tag{64}$$

This gives rise to a learning rule

$$\boldsymbol{\theta}_{i+1} = \boldsymbol{\theta}_i - \gamma \boldsymbol{g} \tag{65}$$

where $\gamma$ is a learning rate which is usually set to a small constant or updated according to some learning rate schedule. However, we note that the parameter change itself has units of $[\boldsymbol{\theta}]^{-1}$ which suggests that such a naïve optimization procedure might be pathological under some circumstances. Consider instead the more general definition of the normalized gradient $\hat{\boldsymbol{g}}$:

$$\hat{\boldsymbol{g}} = \lim_{\epsilon \to 0} \frac{1}{Z(\epsilon)} \text{argmin}_{\boldsymbol{\delta}} \mathcal{L}(\boldsymbol{\theta} + \boldsymbol{\delta}) \qquad\qquad d(\boldsymbol{\theta}, \boldsymbol{\theta} + \boldsymbol{\delta}) \le \epsilon. \tag{66}$$

Here, $\hat{\boldsymbol{g}}$ is the direction in state space which minimizes $\mathcal{L}$ given a step of size $\epsilon$ according to some distance metric $d(\cdot, \cdot)$. Canonical gradient descent is in this case recovered when $d(\cdot, \cdot)$ is Euclidean distance in parameter space

$$d(\boldsymbol{\theta}, \boldsymbol{\theta}') = ||\boldsymbol{\theta} - \boldsymbol{\theta}'||_2^2. \tag{67}$$

We now formulate $\mathcal{L}(\boldsymbol{\theta})$ as depending on a statistical model $p(\mathcal{D}|\boldsymbol{\theta})$ such that $\mathcal{L}(\boldsymbol{\theta}) = \mathcal{L}(p(\mathcal{D}|\boldsymbol{\theta}))$. This allows us to define the direction of steepest gradient in terms of the change in probability distributions

$$d(\boldsymbol{\theta}, \boldsymbol{\theta}') = \text{KL}\left[p(\mathcal{D}|\boldsymbol{\theta}')||p(\mathcal{D}|\boldsymbol{\theta})\right]. \tag{68}$$

It can be shown that the direction of steepest decent for small step sizes is in this case given by [1, 11]

$$\boldsymbol{g} \propto \boldsymbol{F}^{-1} \nabla \mathcal{L}(\boldsymbol{\theta}), \tag{69}$$

where $\boldsymbol{F}$ is the Fisher information matrix

$$\boldsymbol{F}(\boldsymbol{\theta}) = \mathbb{E}_{p(\mathcal{D}|\boldsymbol{\theta})}\left[\nabla \log p(\mathcal{D}|\boldsymbol{\theta}) \nabla \log p(\mathcal{D}|\boldsymbol{\theta})^T\right]. \tag{70}$$

We thus get an update rule of the form

$$\boldsymbol{\theta}_{i+1} = \boldsymbol{\theta}_i - \gamma \boldsymbol{F}^{-1} \nabla_{\boldsymbol{\theta}} \mathcal{L}, \tag{71}$$

which has units of $[\boldsymbol{\theta}]$ and corresponds to a step in the direction of parameter space that maximizes the decrease in $\mathcal{L}$ for an infinitesimal change in $p(\mathcal{D}|\boldsymbol{\theta})$ as measured using KL divergences. It has been shown in a large body of previous work that such natural gradient descent leads to improved performance [1, 2, 18], and the main bottleneck to its implementation is usually the increased cost of computing $\boldsymbol{F}$ or a suitable approximation to this quantity.

We note that this optimization method is very similar to that derived for NCL in Section 2.2 and Appendix A except that NCL uses the approximate Fisher for *previous* tasks instead of the Fisher information matrix of the current loss. This is important since (i) it mitigates the need for computing a fairly expensive Fisher matrix at every update step, and (ii) it ensures that parameters are updated in directions that preserve the performance on previous tasks.

## I   Further results

### I.1   Performance with different prior scalings

Here we consider the performance of KFAC and NCL for different values of $\lambda$ on the stimulus-response task set with 256 recurrent units. We start by recalling that $\lambda$ is a parameter that is used to define a modified Laplace loss function with a rescaling of the prior term (c.f. Section 2.3):

$$\mathcal{L}_k^{(\lambda)}(\boldsymbol{\theta}) = \log p(\mathcal{D}_k|\boldsymbol{\theta}) - \lambda(\boldsymbol{\theta} - \boldsymbol{\mu}_{k-1})^\top \boldsymbol{\Lambda}_{k-1}(\boldsymbol{\theta} - \boldsymbol{\mu}_{k-1}). \tag{72}$$

In this context, it is worth noting that KFAC and NCL have the same stationary points when they share the same value of $\lambda$. Despite this, the performance of NCL was robust across different values of $\lambda$ (Figure 8A), while learning was unstable and performance generally poor for KFAC with small values

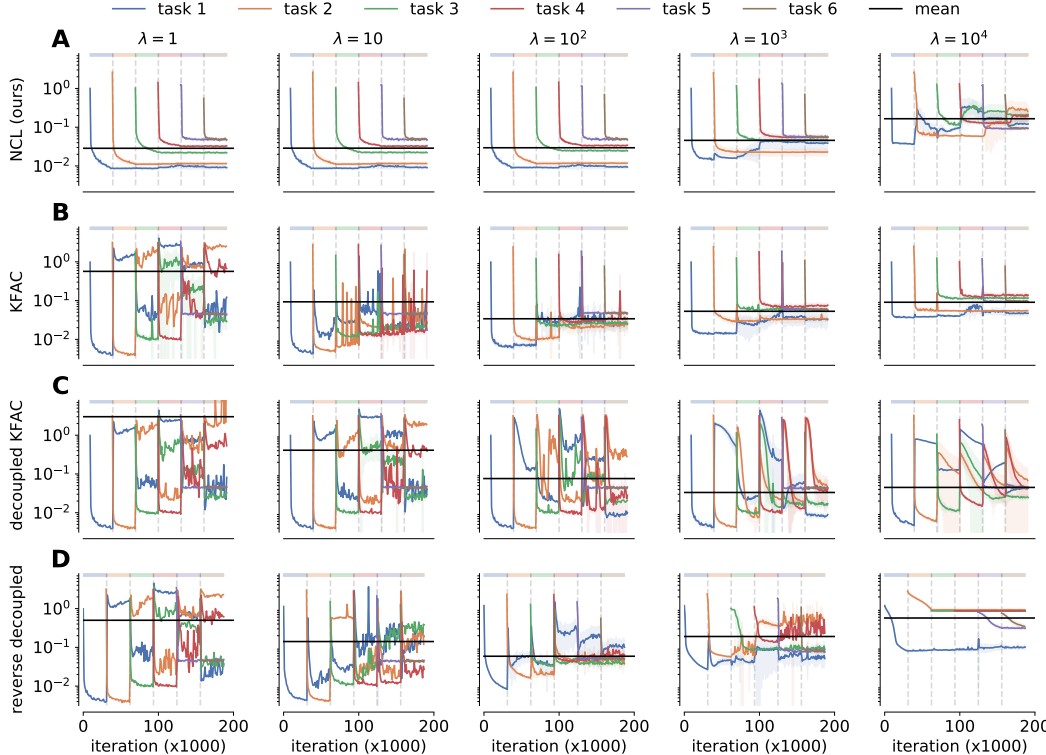

Figure 8: **Continual learning on SR tasks with different $\lambda$.** **(A)** Evolution of the loss during training for each of the six stimulus-response tasks for NCL with different values of $\lambda$. The performance of NCL is generally robust across different choices of $\lambda$ until it starts overfitting too heavily on early tasks. **(B)** As in (A), now for KFAC with Adam which performs poorly for small $\lambda$. **(C)** As in (B), now with "decoupled Adam" where we fix $\lambda_m = 1$ for the gradient estimate and vary $\lambda = \lambda_v$ for the preconditioner (see Appendix I.1 for details). Interestingly, this is sufficient to overcome the catastrophic forgetting observed for KFAC with $\lambda_m = \lambda_v = 1$. The transient forgetting observed at the beginning of a new task is likely due to the time it takes to gradually update the preconditioner for the new task as more data is observed. **(D)** As in (C), now fixing $\lambda_v = 1$ for the preconditioner and varying $\lambda = \lambda_m$ for the gradient estimate. For higher values of $\lambda_m$, this performs worse than both KFAC and decoupled KFAC.

of $\lambda \in [1, 10]$. However, as we increased $\lambda$ for KFAC, learning stabilized and catastrophic forgetting was mitigated (Figure 8B). A similar pattern was observed for the SMNIST task set (Appendix I.2).

We hypothesize that the improved performance of KFAC for high values of $\lambda$ is due in part to the gradient preconditioner of KFAC becoming increasingly similar to NCL's preconditioner $\mathbf{\Lambda}_{k-1}^{-1}$ as $\lambda$ increases (Section 2.3). To test this hypothesis, we modified the Adam optimizer [10] to use different values of $\lambda$ when computing the Adam momentum and preconditioner. Specifically, we computed the momentum and preconditioner of some scalar parameter $\theta$ as:

$$m^{(i)} \leftarrow \beta_1 m^{(i-1)} + (1 - \beta_1)\nabla_\theta \mathcal{L}^{(\lambda_m)} \tag{73}$$

$$v^{(i)} \leftarrow \beta_2 v^{(i-1)} + (1 - \beta_2)\left(\nabla_\theta \mathcal{L}^{(\lambda_v)}\right)^2 \tag{74}$$

where $\mathcal{L}^{(\lambda)}$ is defined in Equation 72 and importantly $\lambda_m$ may not be equal to $\lambda_v$. As in vanilla Adam, we used $m$ and $v$ to update the parameter $\theta$ according to the following update equations at the $i^{th}$ iteration:

$$\hat{m}^{(i)} \leftarrow m^{(i)}/(1 - \beta_1^i) \tag{75}$$

$$\hat{v}^{(i)} \leftarrow v^{(i)}/(1 - \beta_2^i) \tag{76}$$

$$\theta^{(i)} \leftarrow \theta^{(i-1)} + \gamma \hat{m}^{(i)}/(\sqrt{\hat{v}^{(i)}} + \epsilon), \tag{77}$$

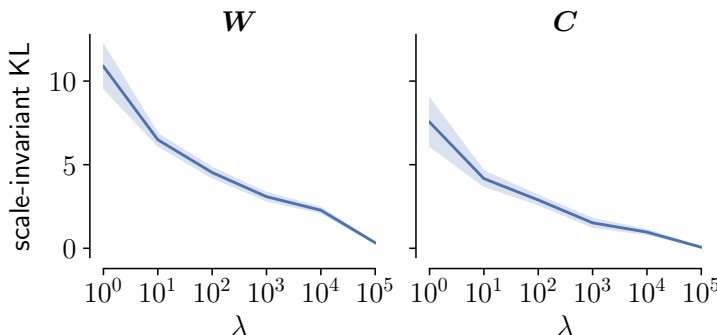

Figure 9: **Similarity of the Adam preconditioner and diagonal Fisher matrix.** Scale-invariant KL divergence (Equation 62) between the diagonal of $\mathbf{\Lambda}_{k-1}$ and the preconditioner used by Adam ($\sqrt{\mathbf{v}}$; 23) at the end of training on task $k$. Results are averaged over the five first stimulus-response tasks, and the figure indicates mean and standard error across 5 seeds for the state matrix $\mathbf{W}$ (left) and the output matrix $\mathbf{C}$ (right).

where $\gamma$ is a learning rate, and $\beta_1$, $\beta_2$, and $\epsilon$ are standard parameters of the Adam optimizer (see 10 for further details). Using this modified version of Adam, which we call "decoupled Adam", we considered two variants of KFAC: (i) "decoupled KFAC", where we fix $\lambda_m = 1$ and vary $\lambda_v$ (Figure 8C), and (ii) "reverse decoupled", where we fix $\lambda_v = 1$ and vary $\lambda_m$ (Figure 8D). We found that "decoupled KFAC" performed well for large $\lambda_v$, suggesting that it is sufficient to overcount the prior in the Adam preconditioner without changing the gradient estimate (Figure 8C). "Reverse decoupled" also partly overcame the catastrophic forgetting for high $\lambda_m$, but performance was worse than for either NCL, vanilla Adam, or decoupled Adam (Figure 8D). These results support our hypothesis that the increased performance of KFAC for high $\lambda$ is due in part to the changes in the gradient preconditioner. To further highlight how the preconditioning in Adam relates to the trust region optimization employed by NCL, we computed the scaled KL divergence between the Adam preconditioner and the diagonal of the Kronecker-factored prior precision matrix $\mathbf{\Lambda}_{k-1}$ at the end of training on task $k$. We found that the Adam preconditioner increasingly resembled $\mathbf{\Lambda}_{k-1}$, the preconditioner used by NCL, as $\lambda$ increased (Figure 9).

In summary, our results suggest that preconditioning with $\mathbf{\Lambda}_{k-1}$ in NCL may mitigate the need to overcount the prior when using weight regularization for continual learning. Additionally, such preconditioning to encourage parameter updates that retain good performance on previous tasks also appears to be a major contributing factor to the success of weight regularization with a high value of $\lambda$ when using Adam for optimization.

## I.2 Hyperparameter optimizations

**Feedforward networks** For the experiments with feedforward networks, we performed hyperparameter optimizations by searching over the following parameter ranges (all on a log-scale): $c$ in SI from $10^{-5}$ to $10^8$, $\lambda$ in EWC and KFAC from $10^{-4}$ to $10^{14}$, $\alpha$ in OWM from $10^{-12}$ to $10^6$, and $p_w^{-2}$ in NCL from $10^2$ to $10^{11}$. The hyperparameter grid searches were performed using a random seed not included during the evaluation. The selected hyperparameter values for each experiment are reported in Table 1.

|  | **Split MNIST** | | | **Split CIFAR-100** | | |
|---|---|---|---|---|---|---|
|  | **Task** | **Domain** | **Class** | **Task** | **Domain** | **Class** |
| SI ($c$) | $10^0$ | $10^5$ | $10^7$ | $10^2$ | $10^3$ | $10^6$ |
| EWC ($\lambda$) | $10^8$ | $10^9$ | $10^{13}$ | $10^7$ | $10^5$ | $10^{-3}$ |
| KFAC ($\lambda$) | $10^{10}$ | $10^5$ | $10^4$ | $10^5$ | $10^3$ | $10^{10}$ |
| OWM ($\alpha$) | $10^{-2}$ | $10^{-5}$ | $10^{-4}$ | $10^{-2}$ | $10^{-2}$ | $10^{-4}$ |
| NCL ($p_w^{-2}$) | $10^3$ | $10^7$ | $10^9$ | $10^3$ | $10^7$ | $10^8$ |

Table 1: Selected hyperparameter values for all compared methods on the experiments with feedforward networks.

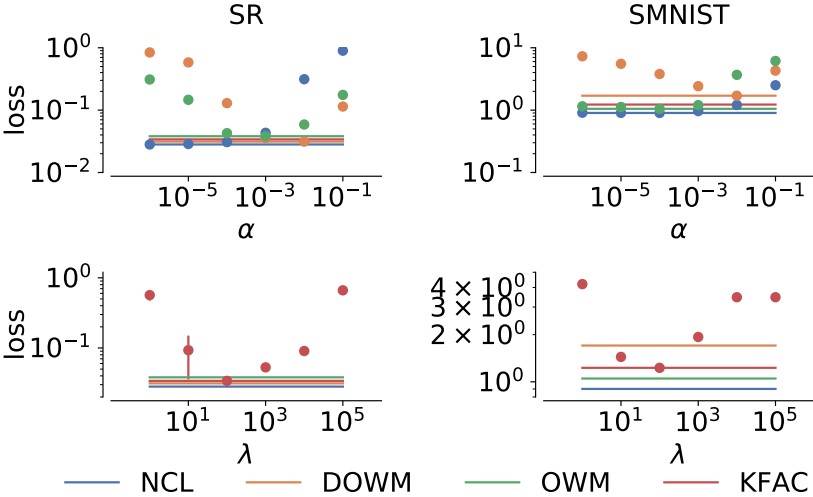

Figure 10: **Hyperparameter optimization for RNNs. (A)** Comparison of the average loss across tasks on the stimulus-response task set as a function of $\alpha$ for the projection-based methods (top panel) and as a function of $\lambda$ for KFAC (bottom panel). Circles and error bars indicate mean and s.e.m. across 5 random seeds. Horizontal lines indicate the optimal value for each method. **(B)** As in (A), now for the SMNIST task set.

**RNNs** For the experiments with RNNs, we optimized over the parameter $\alpha$ used to invert the approximate Fisher matrices in the projection-based methods (NCL, OWM and DOWM) or over the parameter $\lambda$ used to scale the importance of the prior for weight regularization (KFAC).

For KFAC, we found that the performance was very sensitive to the value of $\lambda$ across all tasks sets, and in particular that $\lambda = 1$ performed poorly. In the projection-based methods, $\alpha$ can be seen as evening out the learnings rates between directions that are otherwise constrained by the projection matrices, and indeed standard gradient descent is recovered as $\alpha \to \infty$ (on the Laplace objective for NCL and on $\ell_k$ for OWM/DOWM). We found that NCL in general outperformed the other projection-based methods with less sensitivity to the regularization parameter $\alpha$. DOWM was particularly sensitive to $\alpha$ and required a relatively high value of this parameter to balance its otherwise conservative projection matrices (Appendix F). Here it is also worth noting that there is an extensive literature on how a parameter equivalent to $\alpha$ can be dynamically adjusted when doing standard natural gradient descent using the Fisher matrix for the current loss (see 13 for an overview). While this has not been explored in the context of projection-based continual learning, it could be interesting to combine these projection based methods with Tikhonov dampening [21] in future work to automatically adjust $\alpha$.

We generally report results in the main text and appendix using the optimal hyperparameter settings for each method unless otherwise noted. However, $\alpha = 10^{-5}$ was used for both NCL and Laplace-DOWM in Figure 3C to compare the qualitative behavior of the two different Fisher approximations without the confound of a large learning rate in directions otherwise deemed "important" by the approximation.

### I.3    Numerical results of experiments with feedforward networks

To facilitate comparison to our results, here we provide a table with the numerical results (Table 2) of the experiments with feedforward networks reported in Figure 2 of the main text.

### I.4    SMNIST dynamics with DOWM and replay

In this section, we investigate the latent dynamics of a network trained by DOWM with $\alpha = 0.001$ (c.f. the analysis in Section 3.3 for NCL). Here we found that the task-associated recurrent dynamics for a given task were more stable after learning the corresponding task than in networks trained with NCL. Indeed, the DOWM networks exhibited near-zero drift for early tasks even after learning all 15

Table 2: Numerical results for the experiments with feedforward networks, corresponding to Figure 2 in the main text. Reported is the test accuracy (as %, averaged over all tasks) after training on all tasks. Each experiment was performed either 20 (split MNIST) or 10 (split CIFAR-100) times with different random seeds, and we report the mean ($\pm$ standard error) across seeds.

| Method | Split MNIST | | | Split CIFAR-100 | | |
| --- | --- | --- | --- | --- | --- | --- |
| | Task | Domain | Class | Task | Domain | Class |
| *None* | *81.58 ±1.64* | *59.47 ±1.71* | *19.88 ±0.02* | *61.43 ±0.36* | *18.42 ±0.33* | *7.71 ±0.18* |
| *Joint* | *99.69 ±0.02* | *98.69 ±0.04* | *98.32 ±0.05* | *78.78 ±0.25* | *46.85 ±0.51* | *49.78 ±0.21* |
| SI | 97.24 ±0.55 | 65.20 ±1.48 | 21.40 ±1.30 | 74.84 ±0.39 | 22.58 ±0.37 | 7.02 ±1.04 |
| EWC | 98.67 ±0.22 | 63.44 ±1.70 | 20.08 ±0.16 | 75.38 ±0.24 | 19.97 ±0.44 | 7.74 ±0.18 |
| KFAC | 99.04 ±0.10 | 67.86 ±1.33 | 19.99 ±0.04 | 76.61 ±0.23 | 26.57 ±0.66 | 7.59 ±0.17 |
| OWM | 99.36 ±0.05 | 87.46 ±0.74 | 80.73 ±1.11 | 77.07 ±0.27 | 28.51 ±0.30 | 29.23 ±0.51 |
| NCL (no opt) | 99.53 ±0.03 | 84.9 ±1.06 | 47.49 ±0.84 | 77.88 ±0.26 | 32.81 ±0.38 | 16.63 ±0.34 |
| NCL | 99.55 ±0.03 | 91.48 ±0.64 | 69.31 ±1.65 | 78.38 ±0.27 | 38.79 ±0.24 | 26.36 ±1.09 |

tasks (Figure 11). However, DOWM also learned representations that were less well-separated after the first 1-2 classification tasks (Figure 11, bottom) than those learned by NCL. This is consistent with our results in Section 3.2 where DOWM exhibited high performance on the first task even after learning all 15 tasks, but performed less well on later tasks (Figure 11). These results may be explained by the observation that DOWM tends to overestimate the number of dimensions that are important for learned tasks (Section 3.2) and thus projects out too many dimensions in the parameter updates when learning new tasks.

In the context of biological networks, it is unlikely that the brain remembers previous tasks in a way that causes it to lose the capacity to learn new tasks. However, it is also not clear how the balance between capacity and task complexity plays out in the mammalian brain, which on the one hand has many orders of magnitude more neurons than the networks analyzed here, but on the other hand also learns more behaviors that are more complex than the problems studied in this work. In networks where capacity is not a concern, it may in fact be desirable to employ a strategy similar to that of DOWM — projecting out more dimensions in the parameter updates than is strictly necessary — so as to avoid forgetting in the face of the inevitable noise and turnover of e.g. synapses and cells in biological systems.

To compare with NCL and DOWM which ensure continual learning by regularizing the parameters of the network or the changes in these parameters, we also considered a network that used replay of past data to avoid forgetting. We trained the network using a simple implementation of replay where the learner estimates the task specific loss $\ell_k(\theta)$ as in Section 2. In addition, the network gets to 'replay' a set of examples $\{\boldsymbol{x}^{(k')}, \boldsymbol{y}^{(k')}\}_{k'<k}$ from previous tasks at every iteration to estimate the expected loss on earlier tasks

$$\ell_{<k}(\boldsymbol{\theta}) = \frac{1}{k-1} \sum_{k'=1}^{k-1} \mathbb{E}\left[\sum_t \log p(\boldsymbol{y}_t^{(k')}|\boldsymbol{C}\boldsymbol{r}_t^{(k')})\right]. \tag{78}$$

The parameters are then updated as

$$\boldsymbol{\theta} \leftarrow \boldsymbol{\theta} - \gamma \left[\frac{1}{k}\nabla_\theta \ell_k(\boldsymbol{\theta}) + \frac{k-1}{k}\nabla_\theta \ell_{<k}(\boldsymbol{\theta})\right]. \tag{79}$$

Note that while we explicitly replayed examples drawn from the true data distribution for previous tasks, these examples could instead be drawn from a generative model that is learned in a continual fashion together with the discriminative model [23, 24].

In contrast to the stable task representations learned by DOWM and NCL, continual learning with replay led to task-representations that exhibited a continuous drift after the initial task acquisition (Figure 12). We can understand this by noting that parameter-based continual learning assigns a privileged status to the parameter set $\boldsymbol{\mu}_k$ used when the task is first learned, while methods using replay, generative replay, or other functional regularization methods are invariant to parameter changes that do not affect the functional mapping $f_\theta(x)$. This is interesting since a range of

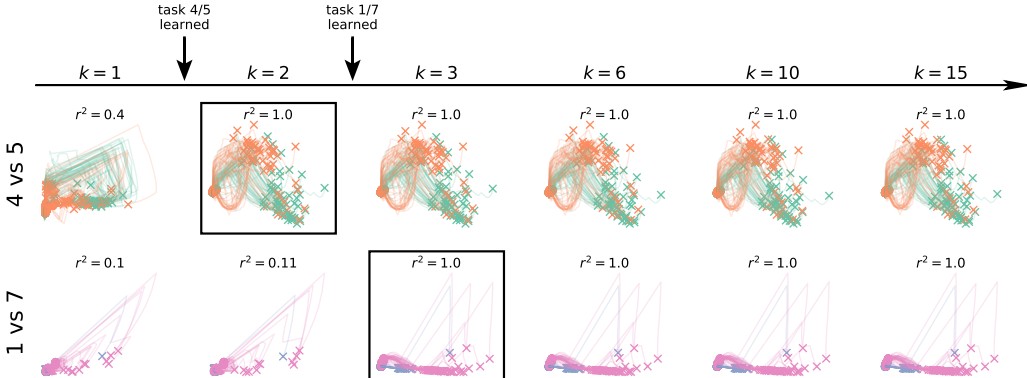

Figure 11: **Latent dynamics during SMNIST.** We considered two example tasks, 4 vs 5 (top) and 1 vs 7 (bottom). For each task, we simulated the response of a network trained by DOWM to 100 digits drawn from that task distribution at different times during learning. We then fitted a factor analysis model for each example task to the response of the network right after the correponding task had been learned (squares; $k = 2$ and $k = 3$ respectively). We used this model to project the responses at different times during learning into a common latent space for each example task. For both example tasks, the network initially exhibited variable dynamics with no clear separation of inputs and subsequently acquired stable dynamics after learning to solve the task. The $r^2$ values above each plot indicate the similarity of neural population activity with that collected immediately after learning the corresponding task, quantified across all neurons (not just the 2D projection).

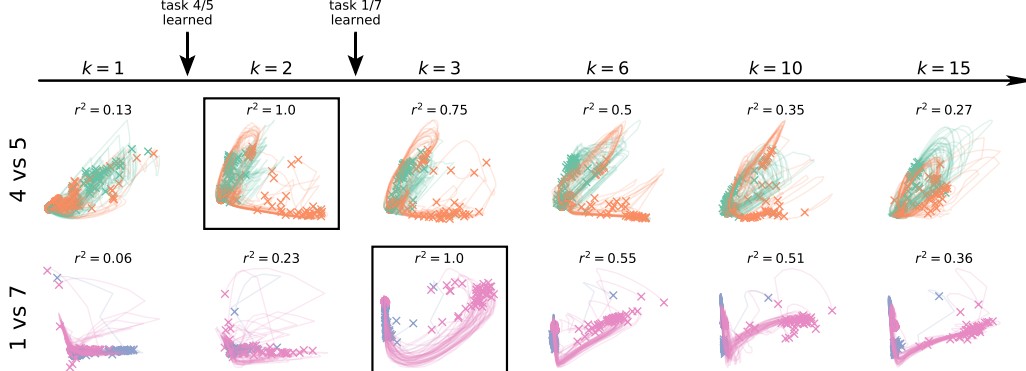

Figure 12: **Latent dynamics during SMNIST with replay.** We considered two example tasks, 4 vs 5 (top) and 1 vs 7 (bottom). For each task, we simulated the response of a network trained with replay (see main text for details) to 100 digits drawn from that task distribution at different times during learning. We then proceeded to analyze the latent space and representational stability as in Figure 11.

studies in the neuroscience literature have investigated the stability of neural representations with some indicating stable representations [3, 5, 8, 9] and others drifting representations [6, 12, 19, 20]. It is thus possible that these differences in experimental findings could reflect differences between animals, tasks and neural circuits in how biological continual learning is implemented. In particular, stable representations could result from selective stabilitization of individual synapses as in parameter regularization methods for continual learning, and drifting representations could result from regularizing a functional mapping using generative replay.

## J    NCL for variational continual learning

**Online variational inference**    In variational continual learning [16], the posterior $p(\boldsymbol{\theta}|\mathcal{D}_k, \boldsymbol{\phi}_{k-1})$ is approximated with a Gaussian variational distribution $q(\boldsymbol{\theta}_k; \boldsymbol{\phi}_k) = \mathcal{N}(\boldsymbol{\theta}_k; \boldsymbol{\mu}, \boldsymbol{\Sigma}_k)$, where $\boldsymbol{\phi}_k = (\boldsymbol{\mu}_k, \boldsymbol{\Sigma}_k)$. We then treat $\boldsymbol{\mu}_k$ and $\boldsymbol{\Sigma}_k$ as variational parameters and minimize the KL-divergence

between $q(\boldsymbol{\theta}_k; \boldsymbol{\phi}_k)$ and the approximate posterior $p(\boldsymbol{\theta}|\mathcal{D}_k, \boldsymbol{\phi}_k) \propto q(\boldsymbol{\theta}; \boldsymbol{\mu}_{k-1}, \boldsymbol{\Sigma}_{k-1})p(\mathcal{D}_k|\boldsymbol{\theta})$:

$$\mathrm{KL}\left(q(\boldsymbol{\theta}; \boldsymbol{\mu}_k, \boldsymbol{\Sigma}_k)||\frac{1}{Z_k}q(\boldsymbol{\theta}; \boldsymbol{\mu}_{k-1}, \boldsymbol{\Sigma}_{k-1})p(\mathcal{D}_k|\boldsymbol{\theta})\right) \tag{80}$$

with respect to $\boldsymbol{\mu}_k$ and $\boldsymbol{\Sigma}_k$. This is equivalent to maximizing the evidence lower-bound (ELBO):

$$\mathcal{L}(\boldsymbol{\mu}_k, \boldsymbol{\Sigma}_k) = \mathbb{E}_{q(\boldsymbol{\theta}; \boldsymbol{\mu}_k, \boldsymbol{\Sigma}_k)}\left[\log p(\mathcal{D}_k|\boldsymbol{\theta}) - \mathrm{KL}\left(q(\boldsymbol{\theta}; \boldsymbol{\mu}_k, \boldsymbol{\Sigma}_k)||q(\boldsymbol{\theta}; \boldsymbol{\mu}_{k-1}, \boldsymbol{\Sigma}_{k-1})\right)\right], \tag{81}$$

to the data log likelihood

$$\log p(\mathcal{D}_k|\boldsymbol{\phi}_{k-1}) = \log \int p(\mathcal{D}_k|\boldsymbol{\theta})q(\boldsymbol{\theta}; \boldsymbol{\phi}_{k-1})d\boldsymbol{\theta} \geq \mathcal{L} \tag{82}$$

with $q(\boldsymbol{\theta}; \boldsymbol{\phi}_{k-1})$ as the 'prior' for task $k$.

Maximizing $\mathcal{L}$ requires the computation of both the first likelihood term and the second KL term in Equation 81. While the second term can be computed analytically, the first term is intractable for general likelihoods $p(\mathcal{D}_k|\boldsymbol{\theta})$. To address this, Nguyen et al. [16] estimate this likelihood term using Monte Carlo sampling:

$$\mathbb{E}_{q(\boldsymbol{\theta}; \boldsymbol{\phi}_k)}\left[\log p(\mathcal{D}_k|\boldsymbol{\theta})\right] \approx \frac{1}{K}\sum_i \log p(\mathcal{D}_k|\boldsymbol{\theta}_i), \tag{83}$$

where $\{\boldsymbol{\theta}_i\}_{i=1}^M \sim q(\boldsymbol{\theta}; \boldsymbol{\phi}_k)$ are drawn from the variational posterior via the reparameterization trick. This allows direct optimization of the variational parameters $\boldsymbol{\mu}_k$ and $\boldsymbol{\Sigma}_k$. To make the method scale to large models with potentially millions of parameters, Nguyen et al. [16] also make a mean-field approximation to the posterior

$$q(\boldsymbol{\theta}; \boldsymbol{\phi}_k) = \mathcal{N}(\boldsymbol{\theta}; \boldsymbol{\mu}_k, \mathrm{diag}(\boldsymbol{\sigma}_k)). \tag{84}$$

**Natural variational continual learning** We now propose an alternative approach to maximizing $\mathcal{L}$ with respect to $\boldsymbol{\phi}_k = (\boldsymbol{\mu}_k, \boldsymbol{\Lambda}_k)$ within the NCL framework, where $\boldsymbol{\Lambda}_k = \boldsymbol{\Sigma}_k^{-1}$ is the precision matrix of $q$ at step $k$. We again solve a trust-region subproblem to find the optimal parameter updates for $\boldsymbol{\mu}_k$ and $\boldsymbol{\Lambda}_k$:

$$\boldsymbol{\Delta}_{\boldsymbol{\mu}_k}, \boldsymbol{\Delta}_{\boldsymbol{\Lambda}_k} = \underset{\boldsymbol{\Delta}_{\boldsymbol{\mu}_k}, \boldsymbol{\Delta}_{\boldsymbol{\Lambda}_k}}{\arg\min}\ \mathcal{L}(\boldsymbol{\mu}_k, \boldsymbol{\Lambda}_k) + \nabla_{\boldsymbol{\mu}_k}\mathcal{L}^\top \boldsymbol{\Delta}_{\boldsymbol{\mu}_k} + \nabla_{\boldsymbol{\Lambda}_k}\mathcal{L}^\top \boldsymbol{\Delta}_{\boldsymbol{\Lambda}_k} \tag{85}$$

$$\text{such that} \quad \mathcal{C}(\boldsymbol{\Delta}_{\boldsymbol{\mu}_k}, \boldsymbol{\Delta}_{\boldsymbol{\Lambda}_k}) \leq r^2, \tag{86}$$

where

$$\mathcal{C}(\boldsymbol{\Delta}_{\boldsymbol{\mu}_k}, \boldsymbol{\Delta}_{\boldsymbol{\Lambda}_k}) = \frac{1}{2}\boldsymbol{\Delta}_{\boldsymbol{\mu}_k}^\top \boldsymbol{\Lambda}_{k-1}\boldsymbol{\Delta}_{\boldsymbol{\mu}_k} + \frac{1}{4}\mathrm{vec}(\boldsymbol{\Delta}_{\boldsymbol{\Lambda}_k})^\top(\boldsymbol{\Lambda}_k^{-1}\otimes\boldsymbol{\Lambda}_k^{-1})\mathrm{vec}(\boldsymbol{\Delta}_{\boldsymbol{\Lambda}_k}). \tag{87}$$

The solution to this optimization problem is given by:

$$\boldsymbol{\Delta}_{\boldsymbol{\mu}_k} = \boldsymbol{\Lambda}_{k-1}^{-1}\nabla_{\boldsymbol{\mu}_k}\mathcal{L} \tag{88}$$

$$\boldsymbol{\Delta}_{\boldsymbol{\Lambda}_k} = 2\boldsymbol{\Lambda}_k\nabla_{\boldsymbol{\Lambda}_k}\mathcal{L}\boldsymbol{\Lambda}_k. \tag{89}$$

To compute $\nabla_{\boldsymbol{\mu}_k}\mathcal{L}$ and $\nabla_{\boldsymbol{\Lambda}_k}\mathcal{L}$, we make use of the following identities [17]:

$$\nabla_{\boldsymbol{\mu}}\mathbb{E}_{\boldsymbol{\theta}\sim\mathcal{N}(\boldsymbol{\theta}; \boldsymbol{\mu}, \boldsymbol{\Sigma})}\left[f(\boldsymbol{\theta})\right] = \mathbb{E}_{\boldsymbol{\theta}\sim\mathcal{N}(\boldsymbol{\theta}; \boldsymbol{\mu}, \boldsymbol{\Sigma})}\left[\nabla_{\boldsymbol{\theta}}f(\boldsymbol{\theta})\right] \tag{90}$$

$$\nabla_{\boldsymbol{\Sigma}}\mathbb{E}_{\boldsymbol{\theta}\sim\mathcal{N}(\boldsymbol{\theta}; \boldsymbol{\mu}, \boldsymbol{\Sigma})}\left[f(\boldsymbol{\theta})\right] = \frac{1}{2}\mathbb{E}_{\boldsymbol{\theta}\sim\mathcal{N}(\boldsymbol{\theta}; \boldsymbol{\mu}, \boldsymbol{\Sigma})}\left[\nabla_{\boldsymbol{\theta}}^2 f(\boldsymbol{\theta})\right]. \tag{91}$$

Applying these identities to compute the gradients of $\mathcal{L}$ (Equation 81), we find

$$\nabla_{\boldsymbol{\mu}_k}\mathcal{L} = \mathbb{E}_{\boldsymbol{\theta}\sim q(\boldsymbol{\theta}; \boldsymbol{\phi}_k)}\left[\nabla_{\boldsymbol{\theta}}\log p(\mathcal{D}_k|\boldsymbol{\theta}) - \boldsymbol{\Lambda}_{k-1}(\boldsymbol{\theta}_k - \boldsymbol{\mu}_{k-1})\right] \tag{92}$$

$$\nabla_{\boldsymbol{\Sigma}_k}\mathcal{L} = \frac{1}{2}\mathbb{E}_{\boldsymbol{\theta}\sim q(\boldsymbol{\theta}; \boldsymbol{\phi}_k)}\left[\nabla_{\boldsymbol{\theta}}^2\log p(\mathcal{D}_k|\boldsymbol{\theta}) - \boldsymbol{\Lambda}_{k-1} + \boldsymbol{\Lambda}_k\right]. \tag{93}$$

Using the fact that $d\boldsymbol{\Lambda}_k = -\boldsymbol{\Lambda}_k^{-1}d\boldsymbol{\Sigma}_k\boldsymbol{\Lambda}_k^{-1}$, we have

$$\nabla_{\boldsymbol{\Lambda}_k}\mathcal{L} = -\boldsymbol{\Lambda}_k^{-1}\nabla_{\boldsymbol{\Sigma}_k}\mathcal{L}\boldsymbol{\Lambda}_k^{-1} \tag{94}$$

$$= -\frac{1}{2}\boldsymbol{\Lambda}_k^{-1}\mathbb{E}_{\boldsymbol{\theta}\sim q(\boldsymbol{\theta}; \boldsymbol{\phi}_k)}\left[\nabla_{\boldsymbol{\theta}}^2\log p(\mathcal{D}_k|\boldsymbol{\theta}) - \boldsymbol{\Lambda}_{k-1} + \boldsymbol{\Lambda}_k\right]\boldsymbol{\Lambda}_k^{-1}. \tag{95}$$

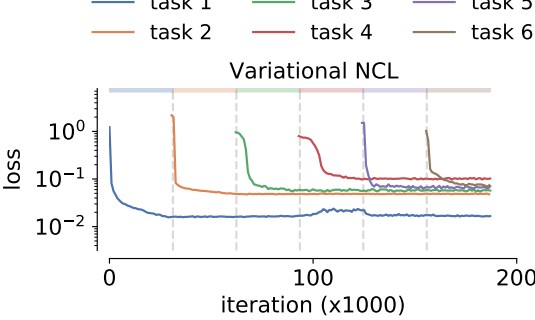

Figure 13: **Natural VCL applied to the stimulus-response task.** Evolution of the loss during training for each of the six stimulus-response tasks using NVCL.

This suggests that we can compute $\boldsymbol{\Delta}_{\boldsymbol{\mu}_k}$ and $\boldsymbol{\Delta}_{\boldsymbol{\Lambda}_k}$ as:

$$\boldsymbol{\Delta}_{\boldsymbol{\mu}_k} = \boldsymbol{\Lambda}_{k-1}^{-1}\mathbb{E}_{\boldsymbol{\theta}\sim q(\boldsymbol{\theta};\boldsymbol{\phi}_k)}\left[\nabla_{\boldsymbol{\theta}}\log p(\mathcal{D}_k|\boldsymbol{\theta}_k)\right] - (\boldsymbol{\mu}_k - \boldsymbol{\mu}_{k-1}) \tag{96}$$

$$\boldsymbol{\Delta}_{\boldsymbol{\Lambda}_k} = \boldsymbol{\Lambda}_{k-1} - \boldsymbol{\Lambda}_k - \mathbb{E}_{\boldsymbol{\theta}\sim q(\boldsymbol{\theta};\boldsymbol{\phi}_k)}\left[\nabla_{\boldsymbol{\theta}}^2\log p(\mathcal{D}_k|\boldsymbol{\theta})\right]. \tag{97}$$

This gives the following update rule at learning iteration $i$ during task $k$:

$$\boldsymbol{\mu}_k^{(i+1)} = (1-\beta)\boldsymbol{\mu}_k^{(i)} + \beta\left[\boldsymbol{\mu}_{k-1} + \boldsymbol{\Lambda}_{k-1}^{-1}\mathbb{E}_{\boldsymbol{\theta}\sim q(\boldsymbol{\theta};\boldsymbol{\phi}_k^{(i)})}\left[\nabla_{\boldsymbol{\theta}}\log p(\mathcal{D}_k|\boldsymbol{\theta})\right]\right] \tag{98}$$

$$\boldsymbol{\Lambda}_k^{(i+1)} = (1-\beta)\boldsymbol{\Lambda}_k^{(i)} + \beta\left[\boldsymbol{\Lambda}_{k-1} - \mathbb{E}_{\boldsymbol{\theta}\sim q(\boldsymbol{\theta};\boldsymbol{\phi}_k^{(i)})}\left[\nabla_{\boldsymbol{\theta}}^2\log p(\mathcal{D}_k|\boldsymbol{\theta})\right]\right], \tag{99}$$

Note that this update rule is equivalent to preconditioning the gradients $\nabla_{\boldsymbol{\mu}_k}\mathcal{L}$ and $\nabla_{\boldsymbol{\Lambda}_k}\mathcal{L}$ with $\boldsymbol{\Lambda}_{k-1}^{-1}$ and $\boldsymbol{\Lambda}_k \otimes \boldsymbol{\Lambda}_k$ respectively.

As for the online Laplace approximation (Section 2.1), one of the main difficulties of implementing the update rule described in Equation 98 and Equation 99 is that it is impractical to compute and store the Hessian of the negative log-likelihood for large models. Furthermore, we need $\boldsymbol{\Lambda}_k^{-1}$ to remain PSD which is not guaranteed as the Hessian is not necessarily PSD. In practice we therefore again approximate the Hessian with the Fisher-information matrix:

$$H_k = -\mathbb{E}\left[\nabla_{\theta}^2\log p(\boldsymbol{\theta})\right] \approx F_k = \mathbb{E}_{\hat{\mathcal{D}}_k\sim p(\mathcal{D}_k|\boldsymbol{\theta})}\left[\nabla_{\theta}\log p(\hat{\mathcal{D}}_k|\boldsymbol{\theta})\nabla_{\theta}\log p(\hat{\mathcal{D}}_k|\boldsymbol{\theta})^{\top}\right]. \tag{100}$$

As in Section 2.2 we use a Kronecker factored approximation to the FIM for computational tractability. With these approximations, we arrive at the learning rule:

$$\boldsymbol{\mu}_k^{(i+1)} = (1-\beta)\boldsymbol{\mu}_k^{(i)} + \beta\left[\boldsymbol{\mu}_{k-1} + \boldsymbol{\Lambda}_{k-1}^{-1}\mathbb{E}_{\boldsymbol{\theta}\sim q(\boldsymbol{\theta};\boldsymbol{\phi}_k^{(i)})}\left[\nabla_{\boldsymbol{\theta}}\log p(\mathcal{D}_k|\boldsymbol{\theta})\right]\right], \tag{101}$$

$$\boldsymbol{\Lambda}_k^{(i+1)} = (1-\beta)\boldsymbol{\Lambda}_k^{(i)} + \beta\left[\boldsymbol{\Lambda}_{k-1} + \mathbb{E}_{\boldsymbol{\theta}\sim q(\boldsymbol{\theta};\boldsymbol{\phi}_k^{(i)})}\left[F_k(\boldsymbol{\theta})\right]\right]. \tag{102}$$

These update rules define the 'natural variational continual learning' (NVCL) algorithm which is the variational equivalent of the Laplace algorithm derived in Section 2.2 and used in Section 3.

**Experiments** To understand how Equations 101-102 encourage continual learning, we note that the first two terms of Equation 101 urge the new parameters $\boldsymbol{\mu}_k$ to stay close to $\boldsymbol{\mu}_{k-1}$. The third term of Equation 101 improves the average performance of the learner on task $k$ by moving $\boldsymbol{\mu}_k$ along $\boldsymbol{\Lambda}_{k-1}^{-1}p(\mathcal{D}_k|\boldsymbol{\theta})$. This is a valid search direction because $\boldsymbol{\Lambda}_{k-1}^{-1} = \boldsymbol{\Sigma}_k$ is the covariance of $q(\boldsymbol{\theta};\boldsymbol{\phi}_{k-1})$ and is thus positive semi-definite (PSD). Importantly, the preconditioner $\boldsymbol{\Lambda}_{k-1}^{-1}$ ensures that $\boldsymbol{\mu}_k$ changes primarily along "flat" directions of $q(\boldsymbol{\theta};\boldsymbol{\phi}_{k-1})$. This in turn encourages $q(\boldsymbol{\theta};\boldsymbol{\phi}_k)$ to stay close to $q(\boldsymbol{\theta};\boldsymbol{\phi}_{k-1})$ in the KL sense. In Equation 102, the first two terms again encourage $\boldsymbol{\Lambda}_k$ to remain close to $\boldsymbol{\Lambda}_{k-1}$. The third term in Equation 102 updates the precision matrix of the approximate posterior with the average Fisher matrix for task $k$. This encourages the curvature of the approximate posterior to be similar to that of the loss landscape of task $k$, and thus (at least locally) parameters that have similar performance on the task will have similar probabilities under the approximate posterior.

To test the natural VCL algorithm, we applied it to the stimulus-response task set considered in Section 3.2 using an RNN with 256 units. Similar to the Laplace version of NCL, we found that

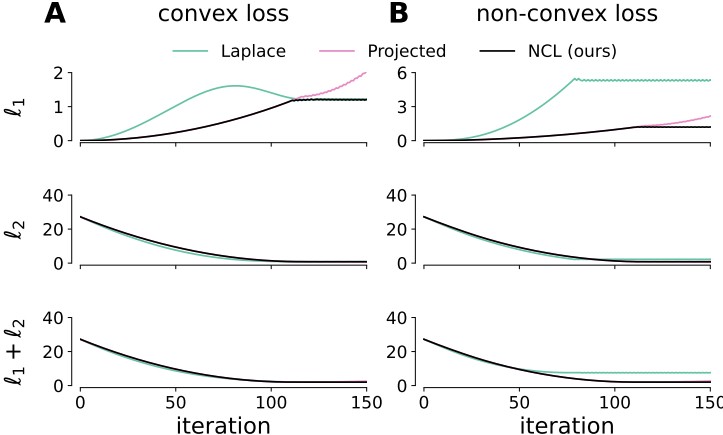

Figure 14: **Losses on toy optimization problem.** **(A)** Loss as a function of optimization step on task 1 (top), task 2 (middle) and the combined loss (bottom) on the convex toy continual learning problem for different optimization methods. **(B)** As in (A), now for the non-convex problem.

NVCL was capable of solving all six tasks without forgetting (Figure 13). While this can be seen as a proof-of-principle that our natural VCL algorithm works, we leave more extensive comparisons between the variational and Laplace algorithms for future work.

**Related work**   Previous studies have proposed the use of variants of natural gradient descent to optimize the variational continual learning objective [18, 22]. The key differences between the method proposed in this section and previous methods are two-fold: (i) we precondition the gradient updates on task $k$ with $\mathbf{\Lambda}_{k-1}^{-1}$ as opposed to $\mathbf{\Lambda}_{k}^{-1}$ as is done in prior work, and (ii) we estimate the Fisher matrix on each task by drawing samples from the model distribution as opposed to the empirical distribution as is the case in Osawa et al. [18], Tseran et al. [22]. It has previously been argued that drawing from the model distribution instead of using the 'empirical' Fisher matrix is important to retain the desirable properties of natural gradient descent [11].

# K   Details of toy example in schematic

In Figure 1A, we consider two regression tasks with losses defined as:

$$\ell_1(\boldsymbol{\theta}) = \frac{1}{2}(\boldsymbol{\theta} - \boldsymbol{\theta}_1)^T \boldsymbol{Q}_1(\boldsymbol{\theta} - \boldsymbol{\theta}_1) \tag{103}$$

$$\ell_2(\boldsymbol{\theta}) = \frac{1}{2}(\boldsymbol{\theta} - \boldsymbol{\theta}_2)^T \boldsymbol{Q}_2(\boldsymbol{\theta} - \boldsymbol{\theta}_2), \tag{104}$$

where $\boldsymbol{\theta}_1 = (3, -6)^\top$, $\boldsymbol{\theta}_2 = (3, 6)^\top$,

$$\boldsymbol{Q}_1 = \boldsymbol{R}(\phi_1) \begin{bmatrix} 1 & 0 \\ 0 & \zeta \end{bmatrix} \boldsymbol{R}(\phi_1)^T, \tag{105}$$

$$\boldsymbol{Q}_2 = \boldsymbol{R}(\phi_2) \begin{bmatrix} 2 & 0 \\ 0 & \zeta \end{bmatrix} \boldsymbol{R}(\phi_2)^T, \tag{106}$$

$$\boldsymbol{R}(\phi) = \begin{bmatrix} \cos(\phi) & -\sin(\phi) \\ \sin(\phi) & \cos(\phi) \end{bmatrix}, \tag{107}$$

and $\zeta = 5.5$. We 'train' on task 1 first by setting $\boldsymbol{\theta} = \boldsymbol{\theta}_1$. We then construct a Laplace approximation to the posterior after learning task 1 to find the posterior precision $\boldsymbol{Q}_1$ (which is in this case exact since the loss is quadratic in $\boldsymbol{\theta}$). Now we proceed to train on task 2 by maximizing the posterior (see Equation 4):

$$\mathcal{L}_2(\boldsymbol{\theta}) = \ell_2(\boldsymbol{\theta}) + \frac{1}{2}(\boldsymbol{\theta} - \boldsymbol{\theta}_1)^T \boldsymbol{Q}_1(\boldsymbol{\theta} - \boldsymbol{\theta}_1) \tag{108}$$

$$= \ell_2(\boldsymbol{\theta}) + \ell_1(\boldsymbol{\theta}) \tag{109}$$

The gradient of $\mathcal{L}_2(\boldsymbol{\theta})$ with respect to $\boldsymbol{\theta}$ is given by:

$$\nabla_{\boldsymbol{\theta}}\mathcal{L} = \boldsymbol{Q}_1(\boldsymbol{\theta} - \boldsymbol{\theta}_1) + \boldsymbol{Q}_2(\boldsymbol{\theta} - \boldsymbol{\theta}_2). \tag{110}$$

We can optimize $\ell(\boldsymbol{\theta})$ using the following three methods:

$$\text{Laplace:} \quad \Delta\boldsymbol{\theta} \propto \boldsymbol{Q}_1(\boldsymbol{\theta} - \boldsymbol{\theta}_1) + \boldsymbol{Q}_2(\boldsymbol{\theta} - \boldsymbol{\theta}_2) \tag{111}$$

$$\text{NCL:} \quad \Delta\boldsymbol{\theta} \propto (\boldsymbol{\theta} - \boldsymbol{\theta}_1) + \boldsymbol{Q}_1^{-1}\boldsymbol{Q}_2(\boldsymbol{\theta} - \boldsymbol{\theta}_2) \tag{112}$$

$$\text{Projected:} \quad \Delta\boldsymbol{\theta} \propto \boldsymbol{Q}_1^{-1}\boldsymbol{Q}_2(\boldsymbol{\theta} - \boldsymbol{\theta}_2), \tag{113}$$

where $\gamma$ is the learning rate and $\boldsymbol{Q}_1 + \boldsymbol{Q}_2$ is the Hessian of $\mathcal{L}(\boldsymbol{w})$. Note that in 'projected', we optimize on task 2 only rather than on the Laplace posterior.

In Figure 1B, we consider a slight modification to $\ell_2$ such that the loss is no longer convex:

$$\ell_2(\boldsymbol{w}) = \frac{1}{2}(\boldsymbol{\theta} - \boldsymbol{\theta}_2)^T\boldsymbol{Q}_2(\boldsymbol{\theta} - \boldsymbol{\theta}_2) + a - a\exp\left(-\frac{1}{2}(\boldsymbol{\theta} - \boldsymbol{v})^T\boldsymbol{Q}_v(\boldsymbol{\theta} - \boldsymbol{v})\right), \tag{114}$$

where we have added a Gaussian with covariance $\boldsymbol{Q}_v$ to the second loss. The NCL preconditioner from task 1 remains unchanged ($\boldsymbol{Q}_1^{-1}$) since $\ell_1$ is unchanged. Denoting $G := a\exp\left(-\frac{1}{2}(\boldsymbol{\theta} - \boldsymbol{v})^T\boldsymbol{Q}_v(\boldsymbol{\theta} - \boldsymbol{v})\right)$, we thus have the following updates when learning task 2:

$$\text{Laplace:} \quad \Delta\boldsymbol{\theta} \propto \boldsymbol{Q}_1(\boldsymbol{\theta} - \boldsymbol{\theta}_1) + \boldsymbol{Q}_2(\boldsymbol{\theta} - \boldsymbol{\theta}_2) + \boldsymbol{Q}_v(\boldsymbol{\theta} - \boldsymbol{v})G \tag{115}$$

$$\text{NCL:} \quad \Delta\boldsymbol{\theta} \propto (\boldsymbol{\theta} - \boldsymbol{\theta}_1) + \boldsymbol{Q}_1^{-1}\boldsymbol{Q}_2(\boldsymbol{\theta} - \boldsymbol{\theta}_2) + \boldsymbol{Q}_1^{-1}\boldsymbol{Q}_v(\boldsymbol{\theta} - \boldsymbol{v})G \tag{116}$$

$$\text{Projected:} \quad \Delta\boldsymbol{\theta} \propto \boldsymbol{Q}_1^{-1}\boldsymbol{Q}_2(\boldsymbol{\theta} - \boldsymbol{\theta}_2) + \boldsymbol{Q}_1^{-1}\boldsymbol{Q}_v(\boldsymbol{\theta} - \boldsymbol{v})G. \tag{117}$$

In this non-convex case, the different methods can converge to different local minima (c.f. Figure 1B).

The losses on both tasks as well as the combined loss as a function of optimization step are illustrated in Figure 14 for the convex and non-convex settings.