# OpenReview forum: "Natural continual learning: success is a journey, not (just) a destination"
_NeurIPS.cc/2021/Conference — NeurIPS 2021 Poster_

### Official Review · Reviewer_wyVR · 2021-07-07

**Rating:** 7
**Confidence:** 3

**Summary:**

The authors introduce an extension to Bayesian continual learning whereby updates in successive tasks are constrained to be close to a “trust region” of the parameters found in preceding tasks. Exploiting the Bayesian nature of the model, the trust region in question can be defined in terms of the prior precision matrix. The authors show that the method, when applied to RNNs, is effective against baselines in a) a suite of neuroscience related tasks, and b) stroke MNIST datasets.

**Limitations And Societal Impact:**

The paper is at the level of basic research, and - in my opinion - is far enough from deployment not to have to societal impact explicitly

**Main Review:**

The paper offers an elegant contribution to the continual learning literature, and I can find very little to criticise, so this review is pretty short. The formal presentation is rigorous and clear and (as far as I was able to check) appears sound. Figure 1 presents the intuition behind the work very clearly. The evaluation is thorough. The method’s relationship to the rest of the literature is well described (although I’m not sufficiently familiar with this particular corner of the continual learning field to vouch that nothing is missing). The paper is logically organised and well written, and I find myself in the unusual position of not having found a single typo. Overall I’m happy to recommend acceptance.

I have one small recommendation. It wasn’t obvious to me why the authors chose an RNN model (and only an RNN model) to evaluate their method. Relatedly, it would have been interesting to see the method applied to other standard (non dynamic) continual learning benchmarks. I assume that these are the sorts of tasks where previous methods don’t do so well. (Apologies if the authors said that somewhere, but I can’t find it looking back over the paper.) Perhaps some intuition about why the method does well in this particular setting could also be given.

**Time Spent Reviewing:**

6

---

> ### Author Response · Authors · 2021-08-10
> **Response to reviewer wyVR**
>
> > The paper offers an elegant contribution to the continual learning literature, and I can find very little to criticise, so this review is pretty short.
>
> We thank the reviewer for the kind comments and we are glad that they enjoyed our paper!
>
> ### Motivation to study RNNs
>
> > It wasn’t obvious to me why the authors chose an RNN model (and only an RNN model) to evaluate their method.
>
> As mentioned in the general comments to reviewers, we chose to focus on RNNs due to (i) their use in a closely related paper by Duncker & Driscoll et al., (ii) the general lack of methods for continual learning in RNNs, and (iii) our interest in how different continual learning algorithms affect the dynamics of the system.
>
> ### Implementation in feedforward networks
>
> > It would have been interesting to see the method applied to other standard (non dynamic) continual learning benchmarks.
>
> We have now implemented NCL for feedforward networks where it outperforms a range of previous methods including EWC and synaptic intelligence across a suite of continual learning benchmarks. We have included these results in a revised version of our paper and report a summary of the results in the general response to reviewers. We hope that this will help further convince the reviewers of the utility of NCL for continual learning.

---

> > ### Comment · Reviewer_wyVR · 2021-08-24
> > **Response to responses**
> >
> > Thanks to the authors for their response. I appreciate the extra results reported in the general response and I think these strengthen the paper. I note that the accuracy on split MNIST in the incremental class learning setting, though outperforming other gradient-based methods, is well below state-of-the-art using other (replay-based) methods (eg: MIR from Aljundi et al (NeurIPS 2019)). However, I don't see this as a weakness of this paper, which I view as a study of gradient-based methods (which can be combined with other methods such as replay). So I stand by my current evaluation.
> >
> > I have a couple of questions about the new results.
> >
> > 1) I assume the split MNIST is a 5-way split, as this is the most common one in the literature. Could you confirm?
> >
> > 2) Could you say what you mean by "simultaneous evaluation"? I assumed that meant the non-continual learning setting where all classes are learned at once. But in that case why are there three columns for this setting. What does "class incremental" mean for "simultaneous evaluation"?

---

> > > ### Author Response · Authors · 2021-08-24
> > > **Clarification on new results**
> > >
> > > Many thanks for the additional comments and clarifying questions!
> > >
> > > It is indeed the case that replay-based methods outperform NCL on the domain/class incremental split MNIST tasks (although NCL does exhibit competitive performance across domain/class incremental permuted MNIST). We agree with the reviewer that the primary focus of our submission is on parameter regularization methods and that these provide the most direct comparison for NCL. With that said, it is of course also an interesting question why there tends to be a fairly substantial performance gap between parameter regularization and replay in the case of domain/class incremental learning. This is indeed one of the reasons we find it exciting that NCL begins to at least partly bridge this gap compared to e.g. EWC and SI.
> > >
> > > To answer the two additional questions:
> > >
> > > 1. As the reviewer notes, split MNIST was implemented as a 5-way split considering pairs of consecutive numbers.
> > > 2. It is correct that 'simultaneous evaluation' corresponds to the non-continual setting where all tasks are interleaved. In this case, there are still minor differences in the network architecture and loss function between the 'task', 'domain' and 'class' incremental settings. For example, in the case of split MNIST, the task incremental setting has a multi-headed output (one for each task) and is trained on a binary loss applied to the relevant head. In contrast, the domain incremental setting has a single-headed binary output. Finally, the class incremental setting has 10 output units and applies a 10-way cross-entropy loss across all digits, but it is only presented with consecutive pairs of digits in the continual setting (simultaneous training for class incremental split MNIST corresponds to standard MNIST classification). Table 2 and the associated section of van de Ven & Tolias (https://arxiv.org/pdf/1904.07734.pdf) also provides more in-depth descriptions/discussions of these settings both in the context of split/permuted MNIST and more generally.
> > >
> > > We hope that this answers your questions and please do let us know if anything else is unclear!

---

> > > > ### Comment · Reviewer_wyVR · 2021-08-24
> > > > **Re: Clarification of new results**
> > > >
> > > > > ... there are still minor differences in the network architecture and loss function
> > > > > between the 'task', 'domain' and 'class' incremental settings
> > > >
> > > > Yes, that's right, of course. Thank you

---

### Official Review · Reviewer_L9uv · 2021-07-12

**Rating:** 6
**Confidence:** 2

**Summary:**

This paper proposes a model for continual learning. Specifically, it intends to merge parameter regularization methods and gradient projection methods. The regularization is achieved by taking a bayesian perspective over the distribution over parameters, that is approximated by a Laplace distribution in a variational fashion, with online updates. The gradient projection is addressed by trust-region optimization, resulting in an update rule that uses the precision matrix to precondition the gradient of the current task objective, plus a term that keeps the parameters close to the previous mean. Experiments are carried out for sequential problems on a stimulus-response dataset and on stroke MNIST.


**Main Review:**

The paper is overall clear and well written, and the approach seems technically sound. All the mathematical notation is correct and the authors provide intuition for the most important equations, which helps readibility.

In terms of technical contribution, the novelty of this work is somewhat limited. As acknowledged by the authors, the bayesian perspective is very common in continual learning and the variational approximations involved here are quite popular. The use of the Fisher information matrix in lieu of the hessian is also standard. To my understanding, the overall model is closely related to Online-EWC [1], where however in the overall update the precision matrix weight the gradient of the current task objective rather than the regularization term. Also, it is not very clear to me the benefit of this scheme, besides getting rid of the need to tune an hyperparameter for the regularization loss.

My main concerns lie however in the experimental section. Specifically:
  - it is peculiar that the authors limit the application of their model to RNN problems only. Indeed, there is nothing specific in the formulation that suggests that the model would behave especially well for sequential problems. Given the fact that those are not so popular in continual learning, I think the paper would benefit from the comparison on standard experiments such as the ones derived from MNIST (split, rotated, permuted) or CIFAR-10.
  - the datasets tested seem to be quite simple, and more complex learning problems may be included to make the conclusions more convincing.
  - Some other standard models such as EWC and its online version should be added to the comparisons, given they are very popular and close in spirit to the model proposed here.
  - for the main comparisons, the authors report the loss (not clear whether it is on training or test sets) as a main metric. Other metrics would be more suitable, such as accuracy or forgetting measures.

---
Overall, I think the paper is slighly below the standard for NeurIPS. The approach presented here is clear and solid, but not fundamentally novel to my understanding. However, as mentioned above, the main weaknesses lie specifically in the experimental section. I think with a revision of the experiments this submission could meet the standard for acceptance.

References:
[1] Schwarz, Jonathan, et al. "Progress & compress: A scalable framework for continual learning." International Conference on Machine Learning. PMLR, 2018.

**Time Spent Reviewing:**

4

---

> ### Author Response · Authors · 2021-08-10
> **Response to reviewer L9uv**
>
> ### General comments
>
> > The paper is overall clear and well written, and the approach seems technically sound. All the mathematical notation is correct and the authors provide intuition for the most important equations, which helps readibility.
>
> We thank the reviewer for their comments and we are delighted that they found our paper clear and technically sound.
>
> > To my understanding, the overall model is closely related to Online-EWC [1], where however in the overall update the precision matrix weight the gradient of the current task objective rather than the regularization term. Also, it is not very clear to me the benefit of this scheme, besides getting rid of the need to tune an hyperparameter for the regularization loss.
>
> We agree that our approach is closely related to online EWC with the key difference being the preconditioning with the prior Fisher matrix in NCL to stabilize the optimization path. In addition to providing a more principled Bayesian interpretation of the method (since EWC with $\lambda > 1$ does not correspond to proper Bayesian inference), we find that this preconditioning leads to performance improvements over such weight regularization approaches even after optimizing $\lambda$ for these methods (c.f. Figure 3 A-C in our initial submission and our results for feedforward networks above).
>
> In addition to proposing a new method for continual learning, we consider a major contribution of our work to be a further characterization and comparison of weight regularization and projection based algorithms for continual learning and shedding light on their respective shortcomings and interactions with commonly used pseudo-second order optimizers such as Adam.
> Indeed we show that with a high value of $\lambda$, EWC-type methods with Adam optimization can be considered an approximation to NCL (c.f. Appendix H.1).
>
> ### Implementation and results in feedforward networks
>
> > it is peculiar that the authors limit the application of their model to RNN problems only. (...) I think the paper would benefit from the comparison on standard experiments such as the ones derived from MNIST (split, rotated, permuted) or CIFAR-10.
>
> To address the questions of several reviewers, we have now implemented NCL in feedforward networks where it outperforms other continual learning algorithms on common benchmarks such as split and permuted MNIST (c.f. our general response to reviewers).
> We have included these results in a revised version of our manuscript and agree that this greatly strengthens our submission by providing further evidence for the performance of NCL as well as illustrating the generality of the method across network architectures.
> For a discussion of the focus on RNN architectures in our original submission and why we also find this to be an interesting application domain, please see our general response above.
>
> ### Difficulty of continual learning problems
>
> > the datasets tested seem to be quite simple, and more complex learning problems may be included to make the conclusions more convincing.
>
> We agree with the reviewer that it is interesting to apply NCL to more difficult and complex continual learning problems in both recurrent and feedforward networks.
> For this reason, we have now applied NCL to the 'domain-incremental' and 'class-incremental' learning paradigms of split and permuted MNIST which have previously proven challenging in continual learning (van de Ven & Tolias 2019).
> Additionally, we have incorporated NCL into a popular public continual learning library which we hope will facilitate further experiments and comparisons in the future.
>
> ### Comparison to standard methods
>
> > Some other standard models such as EWC and its online version should be added to the comparisons, given they are very popular and close in spirit to the model proposed here.
>
> To address the concerns of the reviewer, we have now implemented NCL in feedforward networks where it outperforms EWC, online EWC and synaptic intelligence on the common split MNIST and permuted MNIST benchmarks across three different continual learning scenarios (c.f. van de Ven & Tolias 2019).
>
> In our original submission, we compared directly with and outperformed the continual learning method developed by Duncker & Driscoll et al. (2020) who show that their method performs better than EWC on the same dataset.
> For our experiments in RNNs, we thus instead focused on a comparison with the method 'KFAC' proposed by Ritter et al. (2018) which is equivalent to online EWC except that it uses a Kronecker factored approximation instead of a diagonal approximation to the Fisher matrix. KFAC thus differs from NCL via the preconditioning employed by NCL and the additional hyperparameter $\lambda$ used in KFAC.
>
> ### Performance metrics
>
> > for the main comparisons, the authors report the loss (not clear whether it is on training or test sets) as a main metric. Other metrics would be more suitable, such as accuracy or forgetting measures.
>
> Throughout our submission, we report binary classification errors on a held-out test set for the SMNIST task (e.g. Figure 3C-D). We also report classification test accuracy for our new results in feedforward networks applied to image classification tasks.
>
> For the stimulus-response tasks, we report the loss used by Yang et al. (2019) on a set of trials drawn from the same task distribution as the training trials. For a more intuitive metric, we have now also included a figure in the appendix which compares the ‘success rate’ as defined by Yang et al. (2019) and Duncker & Driscoll et al. (2020). However, for most purposes, we find this to be a less meaningful metric since it relies on an arbitrary cutoff for how close the output must be to a target output for the task to be ‘successfully’ solved; that is, it is essentially a trial-by-trial discretization of the loss which we report in the main text.
>
> ### Conclusion
>
> > I think with a revision of the experiments this submission could meet the standard for acceptance.
>
> We thank the reviewer for the encouraging comment and hope that our revised and expanded experimental section will help convince them of the significance of our work and the utility of NCL for the broader continual learning community.

---

> > ### Comment · Reviewer_L9uv · 2021-08-30
> > **Response to response**
> >
> > I thank the authors for the clear and detailed response.
> >
> > I acknowledge the additional results reported on Split-MNIST and Permuted MNIST address most of my critical concerns and I'd like then to increase my score up a notch. The reason I won't increase more is because I still think this work is not fundamentally novel and presents a variation on the theme of Bayesian methods for continual learning.
> >
> > A couple questions about the new experiments:
> > - I struggle understanding how Split-MNIST is carried out in a domain incremental setting and how Permuted MNIST is carried out in task-incremental and class-incremental settings. Usually Split-MNIST is used in task or class incremental and Permuted MNIST is considered strictly domain incremental. For instance, does Permuted MNIST in class incremental settings mean that at each new task new class heads are instantiated for the new permutation? In that case, I find the results of NCL surprisingly high, which makes me think that I'm misinterpreting the experimental setting.
> > - Referring again to Permuted MNIST in class incremental settings: NCL outperforms the upper bound performance given by simultaneous training. Can the authors comment on that?
> >
> > L9uv

---

> > > ### Author Response · Authors · 2021-08-30
> > > **Further response to reviewer L9uv**
> > >
> > > > I acknowledge the additional results reported on Split-MNIST and Permuted MNIST address most of my critical concerns and I'd like then to increase my score up a notch.
> > >
> > > We thank the reviewer for their encouraging comments and for their willingness to re-evaluate our submission in light of the new results.
> > >
> > > ### Clarification of additional questions
> > >
> > > > I struggle understanding how Split-MNIST is carried out in a domain incremental setting and how Permuted MNIST is carried out in task-incremental and class-incremental settings. Usually Split-MNIST is used in task or class incremental and Permuted MNIST is considered strictly domain incremental. For instance, does Permuted MNIST in class incremental settings mean that at each new task new class heads are instantiated for the new permutation? In that case, I find the results of NCL surprisingly high, which makes me think that I'm misinterpreting the experimental setting.
> > >
> > > We thank the reviewer for these questions and the opportunity to further clarify our additional experiments (see also Figures 1-2 and Tables 2-3 of van de Ven and Tolias 2019 for more detailed explanations and illustrations of all tasks).
> > >
> > > - Split MNIST in the domain incremental setting uses a single headed output and performs a binary classification in each task. That is, the network is effectively asked to determine whether a given input digit is in the set {0,2,4,6,8} or the set {1,3,5,7,9}.
> > > - Permuted MNIST in the task incremental setting uses a single head per task, and each task involves 10-way classification. Task information is in this case available at test time via the selection of the appropriate 'head'.
> > > - Permuted MNIST in the class incremental setting involves a separate set of 10 output units for each task, and the loss is computed across all 100 output units irrespective of which task a particular input digit is drawn from.
> > > - We agree that the performance of NCL for class incremental permuted MNIST is impressively high, and it is in this case comparable to state-of-the-art replay-based methods such as DGR+distill (van de Ven and Tolias 2019). We can further dissect the high performance of NCL by considering how 'KFAC EWC' (Ritter et al. 2018) and 'OWM' (Zeng et al. 2018) perform in this setting. Here, KFAC EWC performs near chance while OWM achieves ~88% accuracy, suggesting that a regularized optimization path is important to avoid catastrophic forgetting across tasks in this setting.
> > > - Finally we note that we used the implementation and network architecture of van de Ven and Tolias (2019) for these experiments, and that they use a fairly large network (fully connected with 2 hidden layers of 1000 units). Since this means that there are many parameters per task, aggressive regularisation to avoid forgetting could potentially do well. It would therefore also be interesting to consider how different methods fare on class incremental permuted MNIST in smaller networks where appropriate allocation of 'degrees of freedom' between tasks is more important (similar to the RNN-based analyses in Figure 3 of our original submission).
> > >
> > > > Referring again to Permuted MNIST in class incremental settings: NCL outperforms the upper bound performance given by simultaneous training. Can the authors comment on that?
> > >
> > > We thank the reviewer for raising this point which does indeed seem curious at first glance, and here we provide two plausible explanations for this observation:
> > >
> > > - NCL and simultaneous training use different optimizers. In particular, we use Adam for our baseline results, while NCL uses SGD with momentum as described in our original submission. It is possible that the minor discrepancy in performance is due to this difference if Adam e.g. overfits slightly more on the training data.
> > > - Our second possible explanation is again related to overfitting. In particular, our Bayesian formulation of NCL includes implicit regularisation in the form of priors over the parameters for each task. Importantly, this includes the first task where we use a prior $\mathcal{N}(0, N^{-1})$ for all parameters. This corresponds to L2 regularization over the parameters which is propagated to other tasks as the Fisher matrix is iteratively updated (see Appendix B of our original submission for further details).
> > > - To further disambiguate these (and possibly other) explanations, we could e.g. compare the performance of NCL to simultaneous training with momentum or L2 regularisation.
> > >
> > >
> > > ### Novelty and scope
> > >
> > > > The reason I won't increase more is because I still think this work is not fundamentally novel and presents a variation on the theme of Bayesian methods for continual learning.
> > >
> > > Finally we want to address this comment in the hope of convincing the reviewer that NCL and our submission as a whole do in fact contribute new and interesting ideas and results.
> > >
> > > - First we want to discuss how our submission advances existing continual learning approaches. Notably, no previous continual learning methods have combined a Bayesian treatment of the parameters with gradient projections during optimization. However, as we illustrate in Figure 1 of our original submission, this can lead to substantially different behavior compared to using each approach on its own. Furthermore, we formulate our gradient projection as a trust region optimization with distances measured according to the prior precision. This differs from previous approaches which generally project the gradients with some variation of the covariance of the network activations (e.g. Zeng et al. 2018, Duncker & Driscoll et al. 2020, Saha et al. 2021). Indeed we suggest that one way to reinterpret these previous methods is by considering such covariance-based projection matrices as approximations to the prior precision (Figure 6), and we show that an explicit Kronecker factored Fisher approximation outperforms these more _ad hoc_ covariance matrices (Figure 3C).
> > > - We also note that a multitude of influential methods for continual learning can be considered variants of other methods. Notably, online EWC (Schwartz et al.) is a variant of EWC (Kirkpatrick et al.) with a slightly different posterior. KFAC EWC (Ritter et al.) is online EWC with a Kronecker factored instead of diagonal approximation to the Fisher matrix. Variational continual learning (Nguyen et al.) is online EWC with a variational approximation instead of a Laplace approximation. Similarly, DOWM (Duncker & Driscoll et al.) is equivalent to OWM (Zeng et al.) with a difference in their right Kronecker factor (Appendix E), and GPM (Saha et al.) resembles OWM with a hard cut-off of 'important directions in parameter space' rather than the continuum of gradient scalings employed by OWM.
> > > - Despite the similarities of such previous methods, we consider all of these papers to have contributed significantly to the CL literature and our understanding of continual learning in general. Similarly, NCL builds on existing methods and ideas, but we also combine and extend them in non-trivial ways. Together with empirical performance that exceeds other methods in its model class and additional analyses of the strengths and failure modes of different continual learning approaches, we remain of the opinion that NCL is a valuable contribution to the literature, and we hope that the reviewers will agree with this.
> > >
> > > ### References
> > >
> > > - Duncker & Driscoll et al. (2020). Organizing recurrent network dynamics by task-computation to enable continual learning. NeurIPS.
> > > - Kirkpatrick et al. (2017). Overcoming catastrophic forgetting in neural networks. Proceedings of the national academy of sciences.
> > > - Nguyen et al. (2018). Variational continual learning. ICLR.
> > > - Ritter et al. (2018). Online structured Laplace approximations for overcoming catastrophic forgetting. arXiv.
> > > - Saha et al. (2021). Gradient projection memory for continual learning. ICLR.
> > > - Schwartz et al. (2018). Progress & compress: A scalable framework for continual learning. arXiv.
> > > - van de Ven & Tolias (2019). Three scenarios for continual learning. arXiv.

---

### Official Review · Reviewer_fhnq · 2021-07-16

**Rating:** 5
**Confidence:** 4

**Summary:**

The authors propose a method called Natural Continual Learning (NCL). NCL unifies the concepts of weight regularization and gradient projection into non-interfering subspaces. Gradient projection is implemented as a trust region optimization using the Fisher information matrix. The authors also introduce a Kronecker-factor approximation to improve the scalability properties of their method. The comparison against other continual learning approaches is conducted using recurrent neural network (RNN) architectures.

**Limitations And Societal Impact:**

Yes

**Main Review:**

I found the paper to be original.

The paper is mostly clearly written (see Minor Issues later on) and well structured.

A major advantage of the paper is that it is quite easy to follow the authors' motivation and thought process that led to the development of the proposed method.

The paper's significance is reduced by the fact that the experimental evaluation was somewhat unconventional and it was conducted only using RNNs. I consider this a significant downside, given that the method is not designed exclusively with RNNs in mind. The paper would be considerably strengthened were the authors to show strong results in more standard CL benchmarks, and when using more challenging datasets than MNIST.

Specifically for subsection 3.2, I found the conclusion reached (see lines 264-266) very interesting and I think it should be validated further using other datasets.

Minor Issues:
- Lines 59-60: The authors write "or network size decreases." At first, I thought that the authors meant a decreasing network size over time, but I think they just mean a limited network capacity relative to the learning task at hand. If the latter is correct, you should probably rephrase.
- In the problem statement paragraph, the authors should probably emphasize that each task has its own distinct data distribution.
- I would also recommend, instead of using the words task and dataset interchangeably (e.g., in subsection 2.1), to instead use only the word task as is customary in CL papers.
- For the vector Δ, introduced in line 147, I would use a bold lowercase character to emphasize the fact that it is indeed a vector and not a matrix.
- Line 334: I think "is not" should be replaced with "are not."
- I recommend using the same vertical axis for Figures 3A and 3B so that they are directly comparable.

Questions:
- Why did you choose to focus only on RNNs instead of experimenting with multiple types of architectures?
- In the non-convex case of the toy problem, why did you set the Task 1 loss to be convex?
- What is the motivation for expanding the number of tasks in the Stroke MNIST benchmark, and why did you make the specific choices detailed in lines 273-278?

**Time Spent Reviewing:**

4

---

> ### Author Response · Authors · 2021-08-10
> **Response to reviewer fhnq**
>
> > I found the paper to be original. The paper is mostly clearly written (see Minor Issues later on) and well structured.
>
> We thank the reviewer for the kind comments, and for the helpful suggestions which we have endeavoured to incorporate in an updated version of our submission.
>
> > Minor Issues
>
> We have addressed the ‘minor issues’ in an updated version of the manuscript.
>
> ### Implementation and results in feedforward networks
>
> > The paper's significance is reduced by the fact that the experimental evaluation was somewhat unconventional and it was conducted only using RNNs. I consider this a significant downside, given that the method is not designed exclusively with RNNs in mind.
>
> Several reviewers have suggested that our paper would be greatly strengthened by considering more diverse problems and architectures.
> For this reason, we have now implemented NCL in feedforward networks where we show superior performance to other methods on a range of benchmarks (see the general response to reviewers for details). We also make NCL available to the commmunity by implementing it in a popular continual learning library for feedforward networks.
>
> > Why did you choose to focus only on RNNs instead of experimenting with multiple types of architectures?
>
> For a further discussion of the motivation for also investigating continual learning in recurrent networks, please see our general response to reviewers.
>
> > Specifically for subsection 3.2, I found the conclusion reached (see lines 264-266) very interesting and I think it should be validated further using other datasets.
>
> Our results in feedforward networks also provide further evidence for our claim that a good approximation to the Fisher matrix is important in challenging continual learning settings. Here we find that KFAC NCL greatly outperforms diagonal NCL in the more challenging domain- and class-incremental learning settings (van de Ven & Tolias 2019). This seems to suggest that parameter correlations are more important in these settings than in the 'task-incremental' setting.
>
> ### Toy problem convexity
>
> > In the non-convex case of the toy problem, why did you set the Task 1 loss to be convex?
>
> We retained a convex ‘task 1’ loss even for the non-convex toy problem because we were interested in the dynamics of learning on the second task rather than the goodness of the Laplace approximation itself which has been investigated in previous work (e.g. Ritter et al. 2018). Indeed with the choice of a convex task 1 loss, the Laplace approximation is exact such that we are working with the ‘true posterior’ for both weight regularization, NCL, and projected gradient descent. We considered this setting more informative since it highlights how the dynamics of learning differ on the second task and how combining regularization with gradient projections can be important even when the Bayesian posterior is exact.
>
> ### SMNIST data augmentation
>
> > What is the motivation for expanding the number of tasks in the Stroke MNIST benchmark, and why did you make the specific choices detailed in lines 273-278?
>
> As other reviewers also point out, it is particularly interesting to compare NCL to other continual learning algorithms on challenging problems. We therefore used data augmentation on the SMNIST data to increase the number of tasks and thus increase the difficulty of the problem. This further highlights the shortcomings of e.g. DOWM (Duncker & Driscoll et al.) which runs out of capacity when the number of tasks is high (c.f. Figure 3C). Such effects are harder to discern with fewer tasks. Our specific data augmentation choices were made as a simple way to generate surrogate digits to increase the number of tasks while also ensuring that none of the input dimensions were identical across multiple tasks.

---

> > ### Comment · Reviewer_fhnq · 2021-09-03
> > **Response to Rebuttal**
> >
> > I thank the authors for their extensive rebuttal. I was on the fence about whether I should increase my score to Weak Accept or not. Eventually, given the lack of experiments with more complex datasets and architectures (e.g., CIFAR-10/100 using deep CNNs), I decided to keep my original score. I would like to encourage the authors to conduct these additional experiments in order to strengthen the practical aspect of their paper, regardless of whether the paper gets accepted or not.

---

> > > ### Author Response · Authors · 2021-09-03
> > > **Further response to reviewer fhnq**
> > >
> > > We thank the reviewer for their response and acknowledge that they would find larger scale experiments particularly convincing. However, we will also make a final case for our current experiments in the hope of convincing the reviewer that our paper and method is of general interest based on our results in RNNs and feedforward networks.
> > >
> > > We note that the original review includes the comment:
> > >
> > > >The paper's significance is reduced by the fact that the experimental evaluation was somewhat unconventional and it was conducted only using RNNs. I consider this a significant downside, given that the method is not designed exclusively with RNNs in mind. The paper would be considerably strengthened were the authors to show strong results in more standard CL benchmarks, and when using more challenging datasets than MNIST.
> > >
> > > Here we will briefly argue that we have addressed each of these suggestions and thus that we have strengthened the paper considerably.
> > >
> > > With respect to the unconventional experimental evaluation, we have extended the experiments to feedforward networks and more standard CL benchmarks which we thank the reviewer for acknowledging.
> > >
> > > We also remain of the opinion that our new benchmarks are significantly more challenging than (i) our previous benchmarks and (ii) vanilla ‘task incremental’ MNIST - and indeed by some measures more challenging than larger scale CIFAR problems.
> > >
> > > To illustrate this, we find the following quote from van de Ven & Tolias (2019) particularly informative (emphasis our own; we apologize that it is slightly lengthy):
> > >
> > > > ​​For each scenario, we performed a comprehensive comparison of recently proposed methods. An important conclusion is that for the class-incremental learning scenario (...), __currently only replay-based methods are capable of producing acceptable results__. In this scenario, even for relatively simple task protocols involving the classification of MNIST-digits, __regularization-based methods such as EWC and SI completely fail__. On the split MNIST task protocol, regularization-based methods also struggle in the domain-incremental learning scenario (...). These results highlight that __for the more challenging, ethological-relevant scenarios where task identity is not provided, replay might be an unavoidable tool__.
> > >
> > > That is, we show that NCL performs remarkably well on a problem where other methods in its class have generally failed catastrophically. We consider it a strong case for NCL that it solves this otherwise open problem. Indeed, we find such a qualitative difference (good performance vs chance level performance) more convincing than performance differences of a few percent on e.g. split CIFAR where minor differences may equally well reflect improper hyperparameter optimization etc.

---

### Official Review · Reviewer_tQo4 · 2021-07-16

**Rating:** 6
**Confidence:** 4

**Summary:**

This paper proposes a novel regularization based continual learning approach, NCL, which is mainly used for recurrent neural network (RNN). Since NCL is based on Bayesian continual learning, NCL utilizes the approximate posterior of previous tasks as a prior for current task, and the Laplace approximation is used to compute the approximate posterior. Through simple toy analyses on convex and non-convex loss landscapes for NCL and other baseline methods, authors show that previous methods have a problem on the trajectories of convergence when learning a new task. Based on the results on simple toy analyses, authors proposed a novel trust region based regularization method that modulate the gradient by considering the curvature of prior distribution via its precision matrix. The proposed technique is highly similar to projection based methods, but its origin is quite different from previous methods. Experiment results show that NCL outperforms other baselines in stimulus-response and MNIST tasks.

**Main Review:**

Pros

- This paper proposed a novel trust-region based regularization methods that can outperform other baselines. The methods modifying the gradients used to update the parameters are quite novel and impressive, and the authors well analyzed the similarity between NCL and other projection based methods in related work.

- By showing simple toy analyses with other baselines, it is easy to see the problems of previous methods and exact trajectories when learning a new task.

Cons

- It would be great for carrying out some large-scale experiments on language or time-series dataset. Based on the experiment results that considered, it is hard to decide NCL is effective on general RNN architecture.

- I think the overall performance of NCL could be highly sensitive to the hyper parameter $r$ in Eq.(7). Since the $\gamma$ in Eq.(8) depends on the $r$, I think it is not easy to choose the proper value of $r$,


Question

- The mathematical form of updating weights in NCL is highly similar to gradient projection based methods except the term $(\theta - \mu_{k-1})$ in Eq.(8). Does the excellence of NCL over gradient projection based methods is from considering Hessian in Laplace approximation or the term $(\theta - \mu_{k-1})$ which tries to reduce the deviation from $\mu_{k-1}$?


Minor comments

- In my humble opinion, in line 146, there is a typo: involves "maximizing" the posterior --> involves "minimizing" the posterior.

**Time Spent Reviewing:**

48 hours

---

> ### Author Response · Authors · 2021-08-10
> **Response to reviewer tQo4**
>
> > The methods modifying the gradients used to update the parameters are quite novel and impressive, and the authors well analyzed the similarity between NCL and other projection based methods in related work.
>
> We thank the reviewer for their encouraging comments and for their suggestions for further improvement of our submission. Here we address their major concerns and comments.
>
> ### Importance of hyperparameter $r$
>
> > I think the overall performance of NCL could be highly sensitive to the hyper parameter $r$ in Eq.(7). Since the $\gamma$ in Eq.(8) depends on the $r$, I think it is not easy to choose the proper value of $r$.
>
> When formalizing NCL as a constrained optimization problem, we can find an explicit expression for the parameter update step (c.f. Eqs. 20-22 in the appendix) where the Lagrange multiplier $\eta$ becomes a function of $r$. However, $\eta$ is simply the inverse of the learning rate $\gamma$. The hyperparameter $r$ thus only enters the algorithm through the learning rate $\gamma$ (intuitively this is because $r$ determines the size of the trust region and thus the length of each parameter update step). Choosing the proper value of r is thus essentially equivalent to choosing the right learning rate. In practice, we found it simpler to work in the space of $\gamma$ directly rather than in the space of $r$ for consistency with other methods.
>
> We experimented with various learning rates and found that performance was fairly robust across learning rates (and thus the value of $r$) for the algorithms and continual learning problems we considered. That being said, using too large a learning rate will of course lead to divergence while too small a learning rate will lead to slow convergence as in most other machine learning settings. We have now added a paragraph in our revised manuscript which discusses the choice of $r$/$\gamma$ in more detail.
>
> ### Large-scale experiments
>
> > It would be great for carrying out some large-scale experiments on language or time-series dataset. Based on the experiment results that considered, it is hard to decide NCL is effective on general RNN architecture.
>
> We agree that it will be interesting to apply NCL to larger scale experiments in both RNNs and feedforward networks.
> In our revised manuscript, we have focused on experiments in feedforward networks to address the concerns of a number of reviewers.
>
> In these feedforward networks, we include results in more challenging domain- and class-incremental learning settings (van de Ven & Tolias 2019) where the difference from previous regularization-based approaches is clearer.
>
> However, it will also be interesting to compare NCL to other methods in larger scale language or time-series tasks in RNNs and image classification tasks in feedforward networks.
> We hope that our publicly available implementations of NCL in both RNNs and feedforward networks will help facilitate such experiments in future work.
>
> ### Importance of the regularization term
>
> > The mathematical form of updating weights in NCL is highly similar to gradient projection based methods except the $(\theta - \mu_{k-1})$ term in Eq.(8). Does the excellence of NCL over gradient projection based methods is from considering Hessian in Laplace approximation or the $(\theta - \mu_{k-1})$ term which tries to reduce the deviation from $\mu_{k-1}$?
>
> As the reviewer correctly notes, what sets NCL apart from other projection based methods is (i) the explicit use of the Fisher matrix for gradient projections, and (ii) the inclusion of a regularization term ($\theta-\mu_{k-1}$) in the loss (conversely what sets NCL apart from EWC-type methods is the projection step).
> We show in our original submission that NCL performs better than the method by Ritter et al. (2018) which corresponds to NCL without gradient projection, suggesting that the gradient projection step is indeed important.
> We also find that the performance of NCL deteriorates with $\lambda=0$ corresponding to removing the regularization term.
> Taken together, this suggests that both the regularization term and the preconditioning are important for the performance of NCL.
>
> For Figure 3D, we also implemented ‘Laplace-DOWM’ which includes the regularization term ($\theta-\mu_{k-1}$) but uses the DOWM projection matrices (Duncker & Driscoll et al. 2020). We found the performance of this method to be comparable to vanilla DOWM, suggesting that the projection step is more important than the regularization term in this case. This is likely due to the DOWM projection being overly conservative as we discuss in section 3.3, making the regularization term less important. However, Laplace-DOWM still underperformed compared to NCL, suggesting that the use of the prior precision matrix for the gradient projection step contributes to the superior performance of NCL compared to e.g. DOWM.
>
> ### Typo
>
> > In my humble opinion, in line 146, there is a typo: involves "maximizing" the posterior --> involves "minimizing" the posterior.
>
> Thanks for spotting this!

---

> > ### Comment · Reviewer_tQo4 · 2021-09-10
> > **Response to Rebuttal**
> >
> > After reading other reviewers' comments and the authors responses, I'd like to keep my rating 6 because the authors resolved my question before. However, it is somewhat unfortunate that experiments on large scale or vision dataset (e.g. CIFAR-100 or mini-ImageNet) was not carried out. Therefore, I decided to keep my rating instead of increasing the score to 7.

---

### Official Review · Reviewer_v6dD · 2021-07-18

**Rating:** 5
**Confidence:** 5

**Summary:**

The paper addresses the problem of catastrophic forgetting during sequential task learning with recurrent neural networks, a notoriously difficult continual learning setting.

The main theoretical contribution is a formalism, Natural Continual Learning (NCL), which generalizes several related continual learning approaches in the regularization-based family, including several recent SOTA techniques. This formalism applies to the probabilistic learning setting using neural networks, so it is not restricted to supervised learning or recurrent models.

Experimental evidence is provided to support the claim that NCL outperforms previous approaches.

Finally, interesting analogies are made to models of biological learning, which are also supported by accompanying experiments; similar analysis tools are applied to practical instantiations of the NCL framework in controlled experimental settings.


**Limitations And Societal Impact:**

The discussion of impact is not particular to this paper, and does apply to the entire field of machine learning, but is well thought out, despite not being specific to this paper.

I would suggest including a discussion of the impact of the work itself and its context rather than making too general remarks.

As it is the case with most papers and discouraged by prevailing reviewing practices, the paper expends little effort to characterize the limitations of the proposed practical algorithm, the diminished relevance of experiments presented within to setups of interest, to related approaches or the wider subfield. Truisms and 'catchphrases of the day' are offered in place of a honest discussion of what current continual learning methods have to offer today, and what is left for future work. I know this is difficult, but please make an effort along these lines!

**Main Review:**

# Originality
## Context and Related work
The paper build on a well developed line of work in continual learning research, namely the regularization-based family of approaches. Such methods rely on relatively "coarse" approximations of second-order quantities, and hence have to come up with practical workaround which mitigate the issues arising from errors in such approximation and their systematic interactions throughout learning.

***

# Quality

## Technical Soundness
* I believe the technical framework proposed for NCL provides clarity and unifies several related approaches.
* However, this it is not itself novel, since all such approaches start from a Bayesian perspective and diverge in practical implementations of diverse approximations of similar quantities.
* While I believe there is merit to a unified approach, the primary limitations of this family of approaches are efficient and effective approximations; I believe the conceptual goals of this family of continual learning approaches were made clear in the literature for some time now, even as early as the EWC paper.


## Assumptions and Limitations
* While the text correctly identifies limitations of related approaches, it does not characterize the significance of the difference between related work and practical implementations of the proposed method. Could you please rectify this or point me to the relevant paragraphs?


## Completeness of Evaluation
* I am not convinced that the presented experiments sufficiently support the claims of superiority to related methods or indeed of relevance to the wider set of continual learning approaches.
* Could you please compare like-for-like with existing approaches on standard settings beyond MNIST classification? While RNNs are of particular importance, it is not clear that the tasks considered are themselves sufficient to demonstrate the superiority of proposed approach, at least not in my view. This is because we already have many solutions to toy continual learning problems; in my view, sufficient progress means solving continual learning in more realistic problem settings.

***

# Significance

## What's unique about this work?
* I appreciate the in-depth study of continual learning with RNNs presented in this paper, the detailed experiments and carefully controlled problem setups considered. This is a valuable paper!
* However, I do not believe it presents enough evidence for the proposed practical algorithms to support claims of superior performance to related or alternative methods. Please let me know whether there are other reasons why no other continual learning approach could be successfully applied to this niche and please argue for the niche itself rather than claiming general applicability. If you mean general applicability in practice, I believe much more evidence is required. If you mean general applicability in principle, please quantify exactly what "general" means and provide an exhaustive list limitations, of both conceptual and practical nature.


# Summary
* This paper is interesting to a number of researchers working on regularization-based continual learning solutions and perhaps modelers of biological learning systems with recurrent models. However, I cannot argue for wider applicability and relevance, which is reflected in my score.
* Furthermore, while progress is being made and well reported in this paper, the significance of this progress to date since to be comparable to that of previously published papers, since the capabilities of continual learning algorithms proposed in this paper are virtually identical to previous work. I could not identify a strong argument for superior capabilities, but I am willing to be convinced otherwise.
* While the perspective is interesting and the analyses revealing, I am not convinced that they contribute enough to recommend acceptance.



**Time Spent Reviewing:**

5

---

> ### Author Response · Authors · 2021-08-10
> **Response to reviewer v6dD**
>
> > I believe the technical framework proposed for NCL provides clarity and unifies several related approaches. (...) I appreciate the in-depth study of continual learning with RNNs presented in this paper, the detailed experiments and carefully controlled problem setups considered. This is a valuable paper!
>
> We thank the reviewer for their very thorough and helpful comments, and we are glad they found our submission to be clear, detailed, and in-depth!
>
> ### Novelty and relation to prior work
>
> > This it is not itself novel, since all such approaches start from a Bayesian perspective and diverge in practical implementations of diverse approximations of similar quantities. (...) While the text correctly identifies limitations of related approaches, it does not characterize the significance of the difference between related work and practical implementations of the proposed method. Could you please rectify this or point me to the relevant paragraphs?
>
> We agree that the Bayesian formulation of continual learning is not in itself novel, as indeed it has been used extensively in the past decade (e.g. Kirkpatrick et al., Huszar et al., Ritter et al., Nguyen et al., and many others).
> However, it is worth noting that these methods do not generally correspond to proper Bayesian inference since the prior is often overcounted by a factor of several hundreds to thousands. Indeed, one of our contributions is to show that this overcounting is not necessary if the optimization path itself is regularized by altering the geometry of the loss landscape via projection with the Fisher matrix.
>
> Additionally, our use of gradient projections for continual learning is not conceptually new and has most notably been used by Zeng et al. (2019) and Duncker & Driscoll et al. (2020). However, to the best of our knowledge such gradient projections have not previously been reformulated as a trust region optimization or combined with parameter regularization. Not only does this lead to the development of a new algorithm for continual learning, but it also (in our opinion) sheds further light on parameter regularization methods and their interactions with pseudo-second order optimizers such as Adam (c.f. Appendix H.1). Furthermore, it helps us better understand the less studied gradient projection methods and provides a principled argument for including a regularization term ($\theta-\mu_{k-1}$) when using such projection-based methods.
>
> From the perspective of parameter regularization for continual learning, there is thus no novelty in the formulation of the loss but rather in introducing a new optimization algorithm that preserves performance on previous tasks throughout learning and improves performance. Conversely from the perspective of gradient projections, the novelty lies in a Bayesian formulation of the projection matrix and the inclusion of an explicit regularization term motivated by an approximation to the Bayesian posterior.
>
> ### Performance
>
> > While RNNs are of particular importance, it is not clear that the tasks considered are themselves sufficient to demonstrate the superiority of proposed approach, at least not in my view. (...) in my view, sufficient progress means solving continual learning in more realistic problem settings.
>
> We have now also implemented NCL for feedforward networks and provided evidence of good performance on more standard benchmarks as well as a PyTorch implementation of NCL in feedforward networks for further use by the community.
> For these analyses, we have focused on the 'three scenarios for continual learning' by van de Ven & Tolias (2019).
> These scenarios include the more realistic 'domain-incremental' and 'class-incremental' settings where output units are shared between tasks and the network does not receive explicit task information at test time.
>
> > I could not identify a strong argument for superior capabilities, but I am willing to be convinced otherwise.
>
> We show that NCL greatly outperforms previous parameter regularization methods across task-, domain- and class-incremental learning for both split and permuted MNIST, and that NCL also outperforms replay-based methods on the permuted MNIST benchmark across these three continual learning scenarios.
> We hope that these findings will help convince the reviewer that NCL provides a valuable continual learning framework which solves some of the shortcomings of existing methods by regularizing both the posterior loss and the optimization path itself.
>
> ### Generality
>
> > If you mean general applicability in practice, I believe much more evidence is required. If you mean general applicability in principle, please quantify exactly what "general" means and provide an exhaustive list limitations, of both conceptual and practical nature.
>
> When we speak of generality, we are speaking of the NCL algorithm rather than our particular implementation; namely the idea of combining weight regularization using an approximate Bayesian posterior with gradient projection based on the prior.
> This can be done for several standard algorithms including e.g. EWC, online-EWC and KFAC-structured continual learning.
>
> To further illustrate this generality in practice, we have now implemented NCL in feedforward networks with both diagonal and Kronecker factored Fisher approximations where we find superior performance to previous regularization-based methods including EWC and online EWC for the Kronecker factored NCL algorithm in particular.
>
> ### Limitations
>
> > I would suggest including a discussion of the impact of the work itself and its context rather than making too general remarks.
>
> We thank the reviewer for highlighting the need for further discussions of the generality and limitations of our work. We have endeavoured to improve this in an updated version of the manuscript with a greatly expanded ‘limitations’ section in the discussion.
>
> In this section, we now discuss specifics of the NCL algorithm including (i) hyperparameter optimization, (ii) the pros and cons of regularization and replay-based methods, (iii) the necessity of explicit task boundaries for computing task-specific Fisher matrices, and (iv) the difficulty in computing Kronecker factored Fisher matrices for complex network architectures. We agree that such considerations should be an important part of a machine learning paper and one that we had not been thorough enough in addressing.
>
> ### Miscellaneous
>
> > Such methods rely on relatively "coarse" approximations of second-order quantities, and hence have to come up with practical workaround which mitigate the issues arising from errors in such approximation and their systematic interactions throughout learning.
>
> Finally we want to highlight the fact that NCL was not designed solely to mitigate the coarseness of the (approximate) Laplace approximation, but also to mitigate the 'greedy' nature of standard gradient descent-based methods.
> This is what we attempted to illustrate in Figure 1b of our submission where the Laplace approximation is exact, but NCL still leads to convergence to a different minimum of the posterior loss compared to weight regularization or gradient projection alone.

---

> > ### Comment · Reviewer_v6dD · 2021-08-28
> > **Rebuttal acknowledgement**
> >
> > I would like to thank the authors for their effort to substantially improve the paper along the directions outlined by several reviewers. The feedforward model investigations bring the paper’s experimental side on a more equal footing with a large body of current CL works.
> >
> > As stated in my original review, I believe this is a valuable paper, well worth the attention of those working on regularization-based continual learning, and perhaps others interested in various competing approaches to continual learning.
> >
> > That said, I have to reiterate my concerns related to novelty and significance, as well as remaining concerns about experimental validation.
> >
> >
> > ### Experimental validation
> > - The regularization-based CL line of work is perhaps the best conceptually and theoretically motivated approach to CL. It is, however, also a good example where the details matter to a substantial degree in practice, as I outline in the original review; hence, I believe it is vital to insist on challenging current benchmarks.
> > - Issues with applying regularization-based methods for CIFAR 10/100 in CL settings have been identified as early as Zenke et al 2017, where the near perfect performance on MNIST benchmarks is misleading, and not representative of performance on a large number of other settings. For example, please consider experiments in the recent (and arguably SOTA) work of Saha et al. 2021. It is clear that the ordering of methods on MNIST benchmark rankings is not predictive of either the gaps in performance and indeed the ordering of methods evaluated on the slightly more complex CIFAR benchmarks.
> > - To get a sense of proportion to such scores, it is also important to look at what ceiling performance may look like (multitask/joint training in these cases); according to Saha et al. 2021 the difference between current SOTA CL and the ceiling can be a double-digit percentage depending on benchmark details. Thus, MNIST results can be misleading, since they imply ceiling performance for this and other similar methods.
> > - Lastly, I want to make it clear that, while the statements above are still a matter of ongoing debate and research, I believe they are representative of the majority viewpoint, e.g. see the review of Parisi et al. 2019. Indeed, a majority of published CL works do not make claims without some form of larger-scale experiments due to the now well known inadequacies of previously popular benchmarks.
> >
> >
> > ### Significance
> > - I find it difficult to argue that the proposed method advances the state of the art in a demonstrable way, for the reasons stated above, and also because recent SOTA methods are excluded by limiting the scope of the work; however, there exist several other competitive approaches to CL (e.g. memory-based, architecture-based) to which we do not get a definitive comparison, e.g. Saha et al. 2021.
> > - While the study of continual learning with RNNs is valuable, I cannot support the view that the general approach or conclusions are unique to this paper, but it is a continuation of studies such as [8, 37].
> >
> >
> >
> > ### Novelty
> > - Ultimately, I cannot argue against the conclusion that this paper introduces a variation of known techniques with comparable performance to such methods.
> >
> >
> > Lastly, I would like to make sure that the authors do not interpret the assigned score as more than an evaluation of the current manuscript for NeurIPS publication today. It doesn’t say anything about the potential of this line of work.
> >
> >
> >
> > ### References (not included in the main paper)
> > - Zenke et al 2017. Continual Learning Through Synaptic Intelligence. http://proceedings.mlr.press/v70/zenke17a/zenke17a.pdf
> > - Saha et al. 2021. Gradient Projection Memory for Continual Learning. https://openreview.net/forum?id=3AOj0RCNC2
> > - Parisi et al 2019. Continual lifelong learning with neural networks: A review. https://www.sciencedirect.com/science/article/pii/S0893608019300231

---

> > > ### Author Response · Authors · 2021-08-29
> > > **Further reply to Reviewer v6dD**
> > >
> > > We thank the reviewer for their further comments and attempt to address some of their additional concerns here.
> > >
> > > > I would like to thank the authors for their effort to substantially improve the paper along the directions outlined by several reviewers. The feedforward model investigations bring the paper’s experimental side on a more equal footing with a large body of current CL works. As stated in my original review, I believe this is a valuable paper, well worth the attention of those working on regularization-based continual learning, and perhaps others interested in various competing approaches to continual learning.
> > >
> > > We are glad the reviewer agrees that our additional experiments in feedforward networks substantially improve the paper, and we appreciate that they have decided to increase their evaluation from the original submission. Like the reviewer, we also hope and think that our paper will be of interest to a range of people in the continual learning community.
> > >
> > > > The regularization-based CL line of work is perhaps the best conceptually and theoretically motivated approach to CL. It is, however, also a good example where the details matter to a substantial degree in practice, as I outline in the original review; hence, I believe it is vital to insist on challenging current benchmarks. (...) To get a sense of proportion to such scores, it is also important to look at what ceiling performance may look like (multitask/joint training in these cases); according to Saha et al. 2021 the difference between current SOTA CL and the ceiling can be a double-digit percentage depending on benchmark details.
> > >
> > > Here we would like to reiterate the fact that the domain and class incremental learning settings which we now include are examples where regularization based approaches have fallen very short of competitive performance (see e.g. van de Ven and Tolias 2019 for an extensive discussion of this). This is indeed the very reason we used this benchmark for our new comparisons; EWC and related algorithms seem to struggle more in this setting than in e.g. multi-headed CIFAR-100.
> > >
> > > For these results, we also report baseline performance levels for simultaneous training. This can be up to 80% better than e.g. EWC and synaptic intelligence which perform at near-chance levels in some settings (e.g. 98% vs 20-30% for class-incremental permuted MNIST). Our new results in feedforward networks of course do not imply that there are no algorithms capable of outperforming NCL on these benchmarks (e.g. methods based on generative replay as discussed in our original reply). However, NCL performs better than other algorithms in its model class and it does so in a way that (in our opinion) helps illustrate the strengths and weaknesses of prior approaches.
> > >
> > > > Lastly, I want to make it clear that, while the statements above are still a matter of ongoing debate and research, I believe they are representative of the majority viewpoint, e.g. see the review of Parisi et al. 2019. Indeed, a majority of published CL works do not make claims without some form of larger-scale experiments due to the now well known inadequacies of previously popular benchmarks.
> > >
> > > While experiments on large-scale image experiments are of course interesting, we would again like to make a case that this is not the only topic of interest in the continual learning literature. Indeed, a multitude of recent papers consider settings other than CIFAR such as tasks in RNNs (e.g. Ehret et al. 2020; Duncker & Driscoll et al. 2020); tasks in the domain/class incremental learning settings (e.g. van de Ven & Tolias 2019; He et al. 2019); and unsupervised learning tasks (e.g. Rao et al. 2019). We believe that this does not make such papers any less interesting, but rather that the literature benefits from a diversity of studies, approaches and challenges.
> > >
> > > > References (not included in the main paper)
> > >
> > > As the reviewer points out, we should have discussed Synaptic Intelligence (Zenke et al. 2017) in our original submission and now include an explicit comparison to this method in the feedforward setting. Here, NCL performs better in both the easier task incremental learning setting and in the more difficult domain and class incremental learning settings where SI fails catastrophically.
> > >
> > > We have not included explicit comparisons to ‘GPM’ recently proposed by Saha et al., but instead to the closely related 'OWM' method by Zeng et al. However, we can also point the reviewer to the ‘domain incremental permuted MNIST’ setting where Saha et al. report 93% performance while NCL achieves 98% performance - a fairly substantial improvement. We are happy to include a further discussion of the similarities and differences between NCL and GPM in an updated manuscript.
> > >
> > > > While the study of continual learning with RNNs is valuable, I cannot support the view that the general approach or conclusions are unique to this paper, but it is a continuation of studies such as [8, 37]. (...) Ultimately, I cannot argue against the conclusion that this paper introduces a variation of known techniques with comparable performance to such methods
> > >
> > > We agree that our submission is closely related to [8] (Duncker & Driscoll et al. 2020), and we discuss similarities and differences between NCL and their approach (DOWM) extensively in our original submission. We also agree that [8] builds on [37] (Yang et al. 2019) as it e.g. uses the same task set. However, this does not imply that [8] was not a valuable contribution to the literature in the same way that the relationship between NCL and DOWM in our opinion does not imply that NCL is not a valuable contribution to the literature.
> > >
> > > While it is thus true that NCL is a ‘variation of known techniques’ and builds on previous work, the same can be said of e.g. online EWC (extension of EWC), KFAC EWC (extension of online EWC), DOWM (extension of OWM), VCL (extension of online EWC with a variational instead of Laplace approximation), GPM (extension of OGD and OWM) and many others. Similarly, NCL builds on and combines previous CL approaches but also introduces new ideas such as (i) combining weight regularization with gradient projection, (ii) projecting with the prior precision rather than the covariance of the activations, and (iii) an improved approach to continually updating Kronecker structured approximate Fisher matrices. Additionally, our approach leads to performance which exceeds comparable methods across both feedforward and recurrent networks (see e.g. the domain and class incremental results for a setting where previous methods do not have comparable performance). Finally, our results and analyses also aim to shed light on the relations between previous approaches as well as their respective strengths and failure modes (e.g. Figures 1 and 4 in our original submission). We hope the reviewers will agree that such contributions can form the basis of a valuable and interesting paper as has also been the case for a multitude of prior studies in both the continual learning literature and other fields.
> > >
> > > ### References
> > >
> > > - Duncker & Driscoll et al. (2020). Organizing recurrent network dynamics by task-computation to enable continual learning. NeurIPS.
> > > - Ehret et al. (2020). Continual learning in recurrent neural networks with hypernetworks. arXiv.
> > > - He et al. (2019). Task agnostic continual learning via meta learning. arXiv.
> > > - Kirkpatrick et al. (2017). Overcoming catastrophic forgetting in neural networks. Proceedings of the national academy of sciences.
> > > - Nguyen et al. (2018). Variational continual learning. ICLR.
> > > - Rao et al. (2019). Continual unsupervised representation learning. NeurIPS
> > > - Ritter et al. (2018). Online structured Laplace approximations for overcoming catastrophic forgetting. arXiv.
> > > - Saha et al. (2021). Gradient projection memory for continual learning. ICLR.
> > > - van de Ven & Tolias (2019). Three scenarios for continual learning. arXiv.
> > > - Zenke et al. (2017). Continual  learning through synaptic intelligence. ICML.

---

### Author Response · Authors · 2021-08-10
**General response to reviewers**

### Implementation and results of NCL in feedforward networks

Several of the reviewers comment on the focus on RNNs in our submission rather than the more standard continual learning setting of feedforward networks. To address these comments, we have implemented NCL for feedforward networks and applied it to the split MNIST and permuted MNIST benchmarks across the task-incremental, domain-incremental, and class-incremental settings (van de Ven & Tolias 2019). We agree that this addition of experiments with a different network architecture and more challenging tasks strengthens our submission, and we hope that it will help convince the reviewers of the utility of NCL for continual learning.

- We have now implemented NCL with both diagonal and Kronecker factored approximations to the Fisher matrix in feedforward networks in order to (i) illustrate how the idea of preconditioning with the prior Fisher can be used with different approximations, and (ii) further investigate the importance of different approximations to the Fisher matrix.
- We performed experiments in the context of the 'three scenarios of continual learning' framework proposed by van de Ven & Tolias (2019) which has previously proven challenging for regularization-based continual learning methods such as EWC and synaptic intelligence. This is particularly true for the 'domain-incremental' and 'class-incremental' settings where output units are shared between tasks and the network does not receive explicit task information at test time (see van de Ven & Tolias 2019 for further details).
- For these comparisons, we have implemented NCL in the context of a popular public continual learning library by van de Ven et al. (https://github.com/GMvandeVen/continual-learning), and we will release this implementation after the anonymous review period. We hope that this will further facilitate the use of NCL in practice as well as additional comparisons with new continual learning methods.

Our results are summarized in the following tables where EWC, online EWC, SI, KFAC EWC and OWM use task-optimized hyperparameters while NCL uses $\lambda=1$ across all tasks and $\alpha$ has been set to a small constant value of $10^{-10}$. Numbers in brackets indicate standard error across random seeds.

| Split MNIST           | Task incremental  | Domain incremental | Class incremental |
| --------------------- | ----------------- | ------------------ | ----------------- |
| None (baseline)       | 81.30% (1.70)     | 58.99% (2.41)      | 19.88% (0.02)     |
| Simultaneous training | 99.68% (0.03)     | 98.44% (0.08)      | 97.97% (0.05)     |
| EWC                   | 98.18% (0.51)     | 62.41% (2.49)      | 20.01% (0.05)     |
| Online EWC            | 98.17% (0.78)     | 62.05% (2.60)      | 19.90% (0.03)     |
| Synaptic intelligence | 98.22% (0.43)     | 61.76% (2.06)      | 19.88% (0.02)     |
| KFAC EWC             |98.24% (0.72)   |63.50% (2.92)     |19.89% (0.02)    |
| OWM                      |99.06% (0.09)   |64.36 (2.57)        | 19.93 (0.02)   |
| Diagonal NCL          | 95.03% (1.08)     | 61.65% (2.08)      | 19.85% (0.02)     |
| KFAC NCL              | **99.47% (0.04)** | **76.41% (2.31)**  | **35.78% (2.14)** |

| Permuted MNIST        | Task incremental  | Domain incremental | Class incremental |
| --------------------- | ----------------- | ------------------ | ----------------- |
| None (baseline)       | 75.24% (1.00)     | 76.93% (0.34)      | 14.43% (0.21)     |
| Simultaneous training | 97.86% (0.02)     | 97.93% (0.02)      | 97.83% (0.01)     |
| EWC                   | 94.19% (0.06)     | 93.79% (0.05)      | 23.00% (1.20)     |
| Online EWC            | 95.61% (0.02)     | 93.54% (0.22)      | 33.60% (1.58)     |
| Synaptic intelligence | 94.46% (0.10)     | 95.32% (0.07)      | 27.30% (0.71)     |
| KFAC EWC             |97.98% (0.02)   |97.76% (0.03)     |18.19% (0.19)    |
| OWM                       |97.42 (0.03)       |93.84 (0.52)        | 88.26 (0.68)      |
| Diagonal NCL          | 93.97% (0.24)     | 93.10% (0.43)      | 75.35% (1.42)     |
| KFAC NCL              | **98.19% (0.01)** | **98.13% (0.02)**  | **97.95% (0.01)** |

- Across the board, we find that NCL with a Kronecker structured Fisher matrix (KFAC NCL) outperforms all the baseline methods. KFAC NCL also improves on the performance of NCL with a diagonal Fisher (Diagonal NCL), further illustrating the importance of a good approximation to the Fisher matrix.
  Like EWC, Diagonal NCL requires further tuning of $\lambda$ to avoid forgetting, particularly on the split MNIST tasks. This is probably to compensate for the poor Fisher approximation (note that the reported results all use $\lambda=1$ for comparison with standard (KFAC) NCL).
- In task-incremental learning, NCL achieves 99.5% accuracy on split MNIST and 98.1% on permuted MNIST. In contrast, the baseline methods generally perform worse, and EWC requires very large values of $\lambda$ on the order of $10^7$ for good performance on split MNIST (van de Ven & Tolias 2019).
- In the domain-incremental setting, weight regularization methods have previously only achieved 60-65% accuracy on split MNIST and ~95% on permuted MNIST (van de Ven & Tolias 2019). NCL with a Kronecker factored Fisher matrix greatly outperforms these methods by achieving 76.4% accuracy on split MNIST and 98.1% on permuted MNIST.
- Finally in the class-incremental setting, KFAC NCL is the only method to perform above baseline on split MNIST and achieves a near-optimal performance of 98% on permuted MNIST where the weight regularization methods only reach 20-30% accuracy and OWM 88% accuracy.
- In summary, NCL greatly improves upon the state of the art for parameter regularization methods on these benchmarks, in particular for the domain- and class-incremental learning settings.
- Replay based methods such as DGR have previously been found to outperform parameter regularization methods in the domain- and class-incremental settings. Our results illustrate that this difference can be mitigated by taking into account the optimization path and correlations between parameters as in NCL. Indeed this also leads to better performance by NCL on the permuted MNIST benchmarks than the previous best reported performance of replay-based methods (see Table 4-5 of van de Ven & Tolias 2019 for results from additional methods including those using replay).
- In future work, it will be interesting to combine NCL with such replay-based methods since the combination of parameter regularization and replay has been found to improve performance in previous work (e.g. van de Ven et al. 2020 & Nguyen et al. 2017).

### Motivation for working with RNNs

While we have now implemented NCL in feedforward networks to address the comments of several reviewers, this section provides some motivation for our original choice to focus on RNNs which we have also elaborated in our revised manuscript.

- Firstly, our work builds on previous work by Duncker & Driscoll et al. (2020) which develops a projection-based method for continual learning and applies it in RNNs. Our choice to also use RNNs therefore facilitates a more direct comparison.
- Secondly, recent work by Duncker & Driscoll et al. (2020) and Ehret et al. (2020) has suggested that traditional continual learning algorithms can struggle in an RNN setting, making this an interesting testbed for the development of new methods.
- Finally, continual learning in RNNs is interesting from a biological perspective where we can compare the dynamics of networks trained with different continual learning algorithms, both to each other and to dynamics recorded in biological systems.
- To further illustrate this point which we investigated in Figure 4 of our original submission, we now also analyze the latent dynamics of continual learning with experience replay during the SMNIST task. In contrast to networks trained by NCL which exhibit stable dynamics after task learning, continual learning with replay leads to drifting dynamics when previous tasks are revisited after new tasks are learned. Such diversity is also found in studies of different biological tasks, brain regions and cell types (e.g. Clopath et al. 2017) which raises intriguing questions of how the brain implements continual learning and how this compares to artificial systems.

### References for responses to reviewers

- Clopath et al. (2017). Variance and invariance of neuronal long-term representations. Philosophical Transactions of the Royal Society B.
- Duncker & Driscoll et al. (2020). Organizing recurrent network dynamics by task-computation to enable continual learning. Advances in Neural Information Processing Systems.
- Ehret et al. (2020). Continual learning in recurrent neural networks with hypernetworks. arXiv.
- Nguyen et al. (2017). Variational continual learning. arXiv.
- Ritter et al. (2018). Online structured Laplace approximations for overcoming catastrophic forgetting. arXiv.
- van de Ven & Tolias (2019). Three scenarios for continual learning. arXiv.
- van de Ven et al. (2020). Brain-inspired replay for continual learning with artificial neural networks. Nature communications.
- Zeng et al. (2019). Continual learning of context-dependent processing in neural networks. Nature Machine Intelligence.

### Updates to our original response to reviewers
- We have now also added comparisons to OWM (Zeng et al. 2018) and EWC with a Kronecker factored Fisher approximation ('KFAC EWC'; Ritter et al. 2018) to the results in feedforward networks above. Kronecker factored NCL outperforms these methods across all the continual learning problems considered. This highlights that the performance improvement of NCL over e.g. EWC is not merely due to a better Fisher approximation, and that NCL also outperforms previous projection based methods in the feedforward setting.

---

### Author Response · Authors · 2021-09-06
**Experiments on split CIFAR**

Following the suggestions of several reviewers, we will now also apply NCL to split CIFAR and compare its performance to previous methods on this larger scale continual learning problem with a convolutional network architecture. We hope this will help strengthen the evidence that NCL provides a valuable alternative to existing methods. In the event that our submission is accepted at NeurIPS, we will endeavor to finish these experiments by the full paper submission deadline and include them in our final manuscript.

---

### Decision · Program_Chairs · 2021-09-27

**Decision:**

Accept (Poster)

**Comment:**

The paper proposes a framework that unifies regularization based techniques in CL and projection techniques. As other reviewers pointed out (e.g. v6dD) this is maybe one of the better formally motivated and better studied direction for addressing CL. And hence at a high level the final result might not seem surprising (e.g. that regularizion methods can be seen as a trust region method, and then there is a deep connection between projection methods and these regularization methods).

However I want to stress that the final algorithm and its derivation is by no means trivial. And I think the improvement (at least in the theoretical understanding) of these methods is of great values for those focusing on such techniques. So from my perspective, in terms of novelty and significance I think the manuscript has provided enough of both. And while there might be some scalability worries, I think there is a significant subset of the CL community that will find this result very useful.

The main weakness of the work is the empirical exploration. While I believe the experiments are done carefully, and provide evidence for the efficacy of the algorithm (plus I welcome the new experiments on feedforward models), the scale of the empirical section is a bit weaker compared to an average paper on this topic. However, particularly considering the new results mentioned by the authors that should be integrated in the paper, I think they might just be sufficient to reach the requirement of the conference.